# Breaking the Barrier of Hard Samples: A Data-Centric Approach to Synthetic Data for Medical Tasks

**Maynara Donato de Souza** [1]    **Cleber Zanchettin** [1]

## Abstract

Data scarcity and quality issues remain significant barriers to developing robust predictive models in medical research. Traditional reliance on real-world data often leads to biased models with poor generalizability across diverse patient populations. Synthetic data generation has emerged as a promising solution, yet challenges related to these samples' representativeness and effective utilization persist. This paper introduces Profile2Gen, a novel data-centric framework designed to guide the generation and refinement of synthetic data, focusing on addressing hard-to-learn samples in regression tasks. We conducted approximately 18,000 experiments to validate its effectiveness across six medical datasets, utilizing seven state-of-the-art generative models. Results demonstrate that refined synthetic samples can reduce predictive errors and enhance model reliability. Additionally, we generalize the DataIQ framework to support regression tasks, enabling its application in broader contexts. Statistical analyses confirm that our approach achieves equal or superior performance compared to models trained exclusively on real data.

## 1. Introduction

Numerous studies have explored various methods to generate synthetic data for augmenting datasets (Bandara et al., 2021), problems related to the representativeness and usefulness of the synthetic data persist (Northcutt et al., 2021). Despite the central role of synthetic data in these approaches, much of the existing research focuses on enhancing model performance (Semenoglou et al., 2023; Abeysinghe et al., 2023; Iglesias et al., 2023) — typically by improving the

generative models, loss functions, or generation methods — without adequately addressing how to handle synthetic data effectively. This gap highlights the need for data-centric approaches that prioritize the quality of synthetic data itself.

Hansen et al. (2023) proposed a data-centric method to generate synthetic data for a generative classification-focused setting. They showed that it is possible to improve the performance by profiling at the preprocessing and postprocessing stages. However, such approaches remain lacking in exploration concerning generative regression-focused tasks. In this context, we introduce Profile2Gen, a systematic framework designed to enhance the generation and refinement of synthetic data by incorporating profiling techniques, ensuring that hard-to-learn samples are considered from the generation process through to the synthetic samples.

Hard-to-learn samples affect model performance, either leading models to predict correctly with low confidence or misclassifying with high confidence. These samples are critical and need to be addressed, especially in medical research, which faces real data scarcity issues and requires representative synthetic samples to enhance diagnostic precision, treatment effectiveness, and patient outcomes, contributing to more efficient and personalized treatment.

To validate the effectiveness of Profile2Gen, we conducted approximately 18,000 experiments using seven state-of-the-art generative models across six distinct medical tabular datasets. The main contributions of our work are summarized as follows:

- We show that integrating Profile2Gen synthetic data can statistically significantly reduce predictive error in regression models.

- We show that models trained through Profile2Gen perform better and exhibit more consistent results, enhancing their reliability for real-world deployment.

- Our approach leads regression models trained only using synthetic data to achieve, in some cases, statistically equal or superior performance compared to models trained using only real data.

- The Proposed approach can preserve the proportion, in most parts of the experiments, of the minority groups.

---

[1]Centro de Informática, Universidade Federal de Pernambuco, Recife, Brazil. Correspondence to: Maynara Souza <mds3@cin.ufpe.br>, Cleber Zanchettin <cz@cin.ufpe.br>.

*Proceedings of the $42^{st}$ International Conference on Machine Learning*, Vancouver, Canada. PMLR 267, 2025. Copyright 2025 by the author(s).

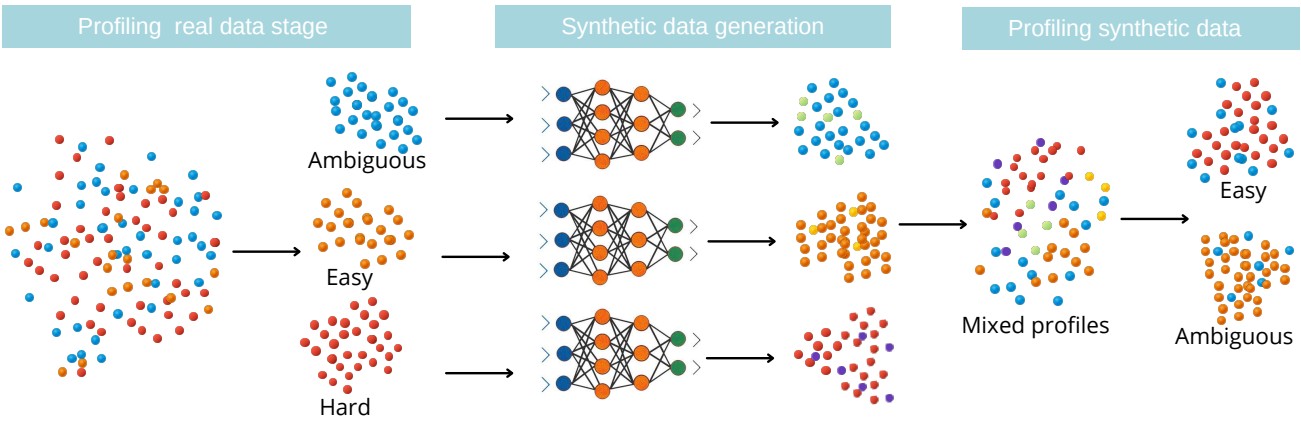

*Figure 1.* The employed methodological process involves a multi-step approach to profile and generate synthetic data. Initially, the training dataset is profiled and categorized into three sample subgroups: easy, hard, and ambiguous. Each of these subgroups is then used independently to train a generative model. The synthetic data generated from these subgroups is combined, and a new profiling is conducted. In the final stage, hard subset samples are identified and removed to refine the dataset, mitigating the risk of distribution poisoning and subsequent performance degradation.

- We propose a generalization of the DataIQ framework (Seedat et al., 2022) to support regression models. This extension addresses the original framework's limitation, as it was initially designed solely for classification tasks. This adaptation enables comprehensive experimentation within the regression context.

## 2. Related Work

We engage with AI data-centric based on characterization, with an extended discussion of related work added to the Supplementary Materials. In real-world AI applications, data scarcity can limit model performance. While improving models is a common approach, enhancing the dataset itself can also lead to better outcomes. One strategy is data characterization, which involves identifying and addressing problematic samples. Tools like DataIQ (Northcutt et al., 2021), and CleanLab (Seedat et al., 2022) help to categorize data based on confidence and uncertainty. DataIQ labels samples as easy, hard, or ambiguous, while CleanLab uses metrics such as entropy. Another approach, the Area Under the Margin (AUM) ranking (Pleiss et al., 2020), focuses on analyzing model confidence to identify difficult samples. As shown by Hansen et al. (2023), integrating these frameworks can enhance synthetic data generation, resulting in high-quality synthetic data that improves performance.

## 3. Profile2Gen framework

The proposed framework is a multi-stage process designed to guide the synthetic data generation, as illustrated in Figure 1. We adapt Hansen et al. (2023) (Hansen et al., 2023), which focuses on classification tasks, to regression tasks.

The iterative approach ensures that only high-quality samples contribute to the final synthetic dataset. The 'ambiguous' or 'hard' samples are identified based on predictive uncertainty metrics, such as low confidence scores or high errors in regression tasks (Seedat et al., 2022). These hard samples play a pivotal role in guiding refinement efforts.

A central component is DataIQReg, our version of the original DataIQ framework (Seedat et al., 2022) to handle regression tasks (Northcutt et al., 2021). It enables systematic data quality profiling and provides actionable insights to guide the generative sample process. Profile2Gen combines data profiling with generative modeling to improve synthetic dataset quality iteratively through this integration.

### 3.1. Framework Workflow

**Preprocessing:** Given a dataset $D$, a framework $F$, and a label flipping proportion $\tau$:

$$D' = \text{FlipLabels}(D, \tau)$$

where $\text{FlipLabels}(D, \tau)$ randomly alters the labels of $\tau \times |D|$ samples. Considering the scoring function $F_{\text{score}}(x)$ - associated to $F$ (See Appendix) - for each sample $x$ such:

$$x \in \text{Hard} \iff F_{\text{score}}(x) < \text{threshold}_T$$

where $\text{threshold}_T$ corresponds to a user-defined value that establishes the acceptable margin for error. The performance of $F$ is evaluated, according to the F1 score - $f$ as in Eq. 1. The best-performing $F_{opt}$ proceeds to the remaining steps.

$$r = \frac{\text{n\_correct\_hard}}{\text{n\_flipped}}, p = \frac{\text{n\_correct\_hard}}{\text{n\_hard}}, f = \frac{2(p \cdot r)}{p + r} \tag{1}$$

Here, n_flipped represents the number of samples with flipped labels, n_hard denotes the number of samples that were classified as hard by $F$, and n_correct_hard refers to the number of instances where the flipped samples classified as hard match the samples identified as hard by $F$.

**Generating Synthetic Data:** According to $F_{opt}$ scoring function (See Appendix) and threshold$_T$, it profiles the $D$ in subsets. For each subset $S \in \{\text{Easy}, \text{Hard}, \text{Ambiguous}\}$:

$$D_{\text{syn}}^{S} = \text{GenerateSyntheticData}(M, S) \tag{2}$$

where $M$ is the generative model trained on $S$.

**Postprocessing:** After combining synthetic subsets $D_{\text{syn}} = \bigcup_S D_{\text{syn}}^S$:

$$D_{\text{final}} = D_{\text{syn}} \setminus \text{HardSamples}(D_{\text{syn}}, F_{\text{opt}}) \tag{3}$$

A discussion of the optimization process, including the sensitivity of $\tau$, is provided in Section E.

## 4. Methodology

We compared Profile2Gen with traditional generative techniques and a baseline preprocessing data-centric approach (PreProcess). The generated data was assessed through multiple evaluation tasks to determine the efficacy of each method. The source code is available on GitHub[1].

### 4.1. Data Generation Methods

**Traditional Generative Techniques:** Data is generated using traditional methods, where generative models are trained without any profiling or distribution-based separation — referred to as Traditional in our analysis. More details about the baseline choice are available in Section N

**PreProcess:** The PreProcess technique serves as a baseline data-centric generative method. It involves the following steps: The dataset undergoes a data-centric profiling phase using frameworks such as Cleanlab or DataIQReg; Based on the profiling results, the data is categorized into subsets (e.g., 'easy' and 'hard'); Generative models are trained separately on each subset, taking advantage of the profile-informed grouping; The final dataset is created by combining the generated data from all subsets without further profiling or refinement. This baseline evaluates the direct impact of

data-centric preprocessing without additional postprocessing steps like those in Profile2Gen.

**Profile2Gen (Proposed Method):** Integrates both preprocessing and postprocessing steps.

### 4.2. Evaluation Tasks

**Train on Synthetic, Test on Real (TSTR):** Evaluate the contribution of synthetic samples on the model's training (Fekri et al., 2019). We compared the Root Mean Squared Error (RMSE) in each step, aiming for the lowest RMSE.

**Augmentation:** The evaluation ensures the validity of the synthetic data for expanding the training dataset.

**Data quality:** Based on three criteria to evaluate synthetic data's quality: Fidelity (accuracy), Diversity, and Generalization (Alaa et al., 2022).

### 4.3. Data-centric AI frameworks

To profile the data, we utilized CleanLab (Seedat et al., 2022) and our proposed DataIqReg, which is based on DataIQ (Northcutt et al., 2021). Further details in Section B.

### 4.4. Datasets

We used six public datasets available on the OpenML website. The datasets were selected based on availability and accessibility due to the well-documented scarcity of public, high-quality medical datasets for synthetic data research. The datasets include the Parkinson dataset (Tsanas et al., 2009) (Parkinson), which has a primary aim to predict the motor and total UPDRS scores (Tsanas & Little, 2016); The urinary dataset (Czerniak & Zarzycki, 2003) aims to predict the diagnosis of two diseases of the urinary system (Czerniak, 2015). Cholesterol dataset (Detrano et al., 1989) concerning heart disease diagnosis (OpenML, 2024d). Body Fat dataset (fat) lists estimates of the percentage of body fat (OpenML, 2024c). Plasma Levels dataset (plasma) (Nierenberg et al., 1989) studies the plasma concentrations.(OpenML, 2024b). Diabetes dataset (Torgo, 2024) aims to predict the C-peptide level (OpenML, 2024a). The Supplementary Material (Section A) contains more details.

### 4.5. Models

To generate synthetic data, we utilized seven stat-of-art in tabular synthetic data generation models: CTGAN (Xu et al., 2019a), TVAE (Xu et al., 2019a), NFlow (Kobyzev et al., 2021), Bayesian Network (Young et al., 2009), Ddpm (Ho et al., 2020), Tabformer (Padhi et al., 2021), and Great (Borisov et al., 2023). To assess the experiments, we adopted twelve tabular data predictors: XGBoost, WeightedEnsembleL2, Neural Network, CatBoost, NeuralNetFastAI, ExtraTreesMSE, LightGBMLarge, RandomForestMSE,

---

[1]https://github.com/szanara/profile2gen.git

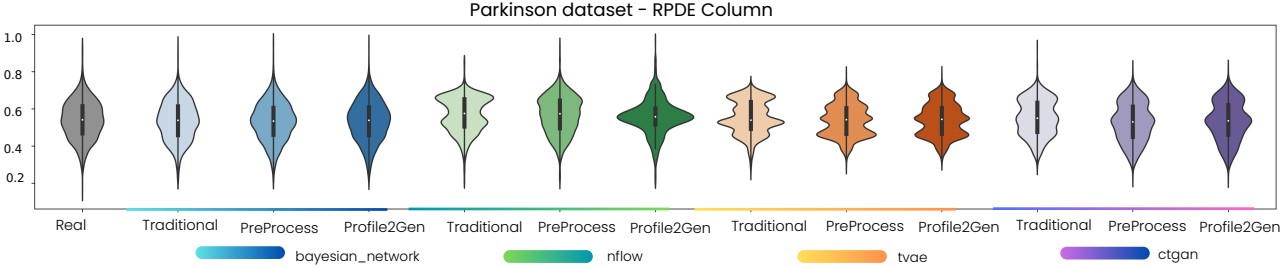

*Figure 2.* Violin plots for a randomly selected column. The first violin represents the real distribution, followed by four groups of violins, each corresponding to a generative model and separated by horizontal lines. Violins share the same color palette within each group, with each violin representing a specific technique, starting with Traditional. A caption for each group is provided below the graph. The plot illustrates the decreasing fidelity of the distributions, moving from the real data to the final technique in each group. Figure 15 provides complementary details of this analysis.

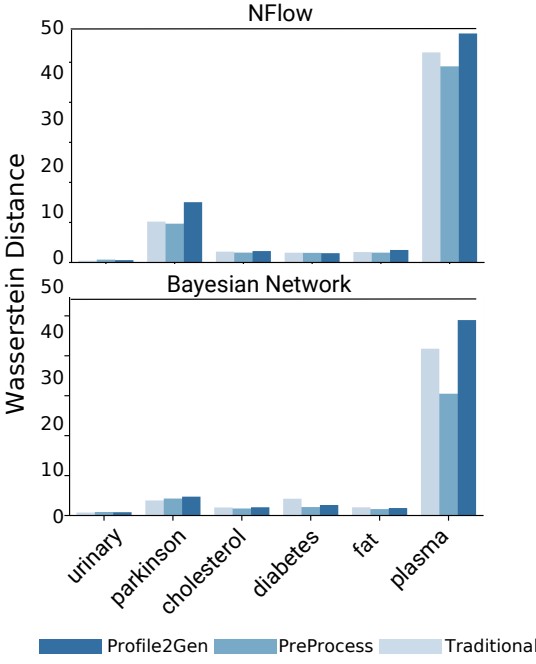

*Figure 3.* Wasserstein Distance across Data Stages for Generative Models. The grey bars represent the Wasserstein distance between the synthetic data generated in the 'Traditional' stage and the real data. The light blue bars correspond to the 'PreProcess', and the dark blue bars indicate the 'Profile2Gen' stage. The y-axis displays the Wasserstein distance values, while the x-axis corresponds to the evaluated datasets. Fig. 14 is complementary to this Figure.

LightGBM, LightGBMXT, KNeighborsUnif, and KNeighborsDist. These models are all available within the AutoGluon library (Erickson et al., 2020), a framework that automates the utilization of state-of-the-art models.

## 5. Results and Discussions

We utilized the Wasserstein distance (Panaretos & Zemel, 2019b) as a metric for measuring the similarity between synthetic and real samples (see Figure 3 for details). In addition, we analyzed the distribution of a randomly selected column from each dataset at various stages. This analysis enabled us to compare distributions and assess data quality based on key criteria such as fidelity, diversity, and generalization (Alaa et al., 2022). These comparisons are visually illustrated in Figure 2. Due to the number of experiments, we focused on commenting on highlights, and further details, including model-specific performance (see Section R) and datasets-variation impact (see Section S) are provided in the supplementary materials. To assess computational efficiency, we selected the largest dataset, Parkinson, with approximately 3,500 training samples. We applied the framework selection process (Cleanlab and DataIQ) along with profiling, using two thresholds and a label replacement ratio. These experiments required 3,510 MB of memory and approximately 3 minutes to complete. More details about scalability are discussed in Section D.

### 5.1. TSTR Results

Considering that Profile2Gen's synthetic datasets are about 10% smaller than the original (due to hard profile removal, see Figure 13), its overall performance can be considered strong. In some cases (Figure 19), TSTR even outperforms Real data, achieving up to 1.2% lower RMSE. Since matching or surpassing real data performance using only synthetic data is a significant challenge (Hao et al., 2024), we focus on comparing Profile2Gen with other approaches—particularly Traditional, a widely used baseline.

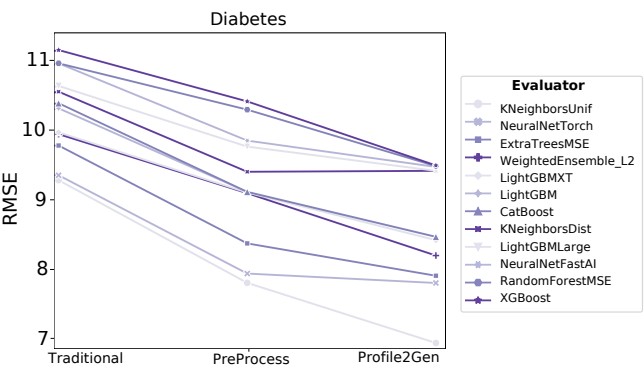

*Figure 4.* Line plots depicting RMSE across the approaches for the twelve models employed in the TSTR task. The plot illustrates the Diabetes dataset synthetically provided by the TVAE model. The plots represent the best-performing scenarios observed.

## 5.2. Augmentation Results

Profile2Gen consistently outperforms Traditional, reducing RMSE by 4.8% (Figure 5) and achieving superior performance in most scenarios with less data. Profile2Gen produces high-quality synthetic data that enables better generalization and lower RMSE. Figure 4 illustrates a common trend: Traditional exhibits high RMSE, indicating poor performance; PreProcess improves upon this with some RMSE reduction; Profile2Gen achieves the lowest RMSE, demonstrating the best overall performance. This underscores the value of data profiling and refinement, showing that a smaller, well-curated dataset can be more effective than a larger, unrefined one. Additionally, Profile2Gen enhances generalization across models and datasets (Figures 20–22), making it particularly useful when real data is scarce.

However, performance differences across approaches are minor in some cases, revealing trade-offs. For example, PreProcess sometimes achieves lower RMSE than Profile2Gen, though the latter is more consistent across models and datasets. This suggests that preprocessing alone—without additional profiling refinement—may be preferable in specific scenarios, depending on the dataset and model used.

We have a two-level evaluation: i) overall performance across experiments, grouped by dataset, model, augmentation proportions, random seeds (333, 555, 666, 888), and evaluation models; and ii) the impact of varying synthetic data proportions for each dataset and model. Proportions were randomly selected between 0.1 and 1 to avoid bias and ensure a fair evaluation of the augmentation process.

### 5.2.1. GENERAL CONSIDERATIONS

The results across various datasets demonstrate that Profile2Gen significantly improves model performance by re-

ducing RMSE values compared to the Traditional and Pre-Process techniques (see Figure 5). In the Urinary, Parkinson, Plasma, and Fat datasets, models such as Bayesian Network, NFlow, and TVAE consistently showed reduced RMSE using the Profile2Gen approach, with some models even surpassing the baseline performance of real data, particularly in the Plasma dataset (see Figure 6). However, the CTGAN model often exhibited high variability, and in the Diabetes dataset, the PreProcess stage outperformed the Profile2Gen stage, indicating some inconsistency in the effectiveness of the synthetic data across different contexts (Fig. 23 - 26).

The results suggest that the refined synthetic data enhances model performance, but its impact varies depending on the dataset and model used. However, despite showing clear benefits, it is not a one-size-fits-all solution. The variability observed across different scenarios indicates the need for ongoing optimization and possibly even custom strategies for each model-dataset combination.

### 5.2.2. PROPORTIONS ANALYSIS

Souza et al. (2023) (de Souza et al., 2023) assert that there is no established rule or recommendation regarding the proportion or quantity of synthetic data used in augmentation tasks. Our experiments align with this statement, as the results vary across models, datasets, and the amounts of data added. However, there are patterns in the performance. Adding 22.7% (0.227) of synthetic data maintains performance close to the baseline, indicating minimal impact on model accuracy. However, when the proportion of synthetic data increases to 68.2%, many models begin to show noticeable deviations from the baseline, suggesting that this is a critical point where performance can start to degrade. At 90.9% synthetic data, performance typically declines significantly, likely due to the model being overloaded with non-generalizable information, particularly in the 'Traditional' and 'PreProcess' stages. Adding 45.4% synthetic data often keeps model performance stable, though some variations may occur depending on the model and stage (Figure 23 - 26). Robust models like Bayesian Network and CTGAN generally handle this amount well, without significant increases in RMSE.

## 5.3. The Trade-off Between Generalization and Diversity

The Wasserstein distance quantifies the effort required to transform one probability distribution into another (Panaretos & Zemel, 2019a), measuring how closely synthetic data approximates real data. Lower values indicate better generalization, while higher values suggest greater diversity.

Figure 3 shows the Wasserstein distance across different approaches. As data-centric processes progress, similarity to real data generally decreases. For the urinary dataset,

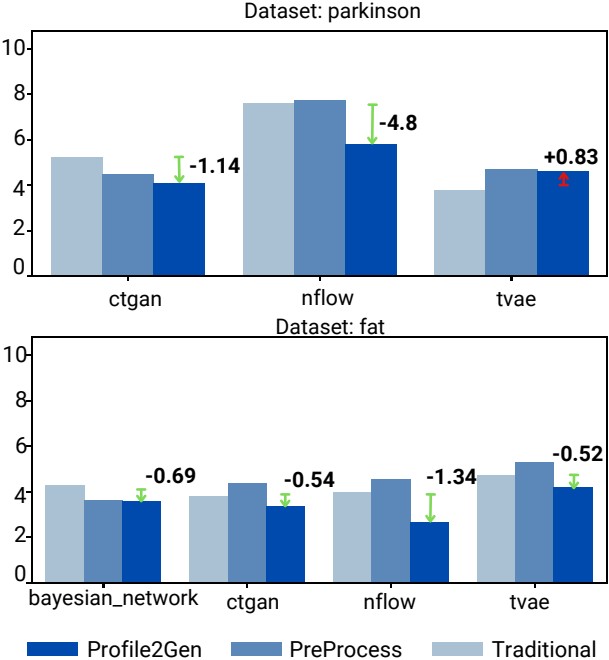

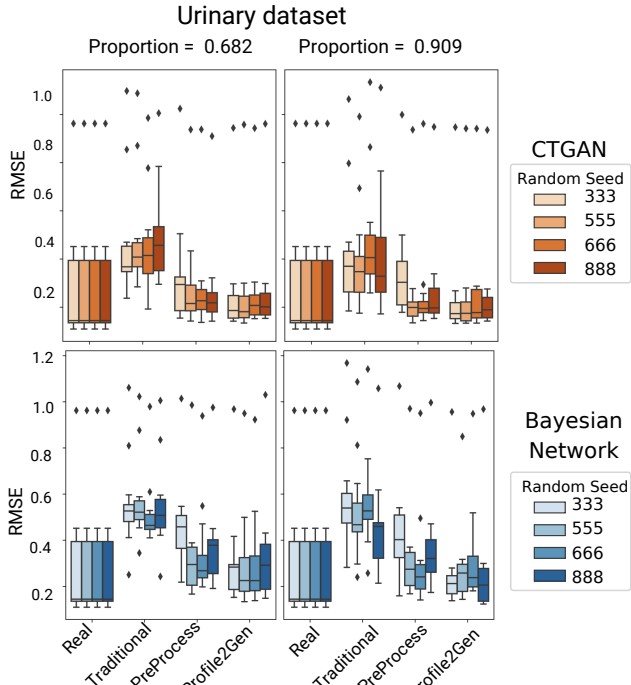

*Figure 5.* Average improvement per model and dataset using the TSTR protocol. Bars represent the average performance of generative techniques across 12 predictors, grouped by model, with each plot corresponding to a different dataset. Green arrows indicate improvement relative to traditional methods, while red arrows show no improvement. Improvement values are displayed alongside arrows with + or − signs indicating the change direction. See the Supplementary Material for the full figure.

*Figure 6.* Augmentation performance on the urinary dataset across random seeds, ensuring unbiased results. Columns compare models at the same synthetic data proportion, while rows show RMSE ranges across varying seeds. Color tones align with the legend. Real performance serves as the baseline. The Profile2Gen stage shows more consistent behavior compared to other stages. Complete results are available in the supplementary materials.

synthetic data closely mirrors the real distribution, suggesting strong generalization but limited diversity. In contrast, datasets like Plasma and Parkinson's exhibit higher distances, indicating more significant variability. Figure 2 reinforces this, showing that Profile2Gen increases variability, enhancing diversity but potentially introducing noise that reduces fidelity. The shift from Traditional to Profile2Gen highlights the generalization-diversity trade-off. Greater diversity helps prevent overfitting but may reduce the synthetic data's ability to capture the real dataset's complexity. For example, while the urinary dataset maintains consistent mean values across generative models—indicating good generalization—its lack of variability may limit its usefulness for augmentation (Gong et al., 2019). In contrast, Plasma and Parkinson's datasets show greater diversity, which enhances robustness but may hinder generalization. Balancing generalization and diversity is crucial for optimizing synthetic data. High generalization ensures fidelity to real data, making it ideal for TSTR tasks where data is scarce. Conversely, high diversity improves model robustness and supports edge-case exploration but may reduce effectiveness in tasks requiring precise real-data representation. Tailoring

synthetic data for specific tasks requires a strategic model and approach selection to achieve an optimal balance. More discussions are in Section K

### 5.4. Data Consistency, Robustness and Reliability

Using different random seeds during augmentation reduces reliance on a single sample and allows performance variability assessment. We analyze this variability through box plots (Figure 6), where narrower boxes and shorter whiskers indicate greater consistency. Our results show that removing hard samples, as proposed by Profile2Gen, significantly reduces variability and enhances consistency across models and datasets. For instance, the Bayesian Network model, which showed high variability with the Traditional approach on the Urinary dataset, became more stable with Profile2Gen—a trend also observed in Parkinson's, Plasma, Fat, and Diabetes datasets. This suggests that Profile2Gen minimizes sensitivity to randomness, improving stability and robustness (Cateni et al., 2023).

Interestingly, while Profile2Gen enhances consistency, it

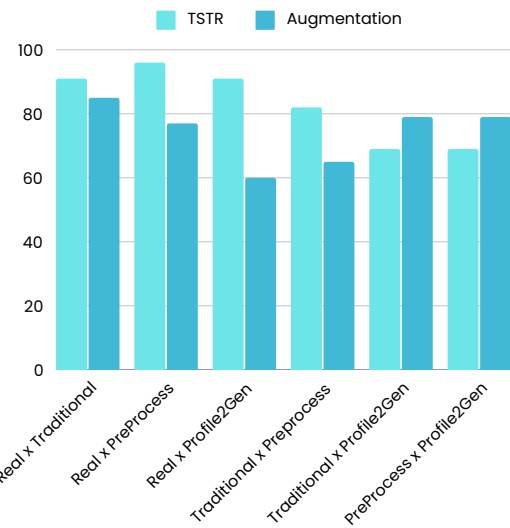

*Figure 7.* Percentage of statistically significant results (p ≤ 0.05, Wilcoxon test) for different data generation approaches. The x-axis compares pipelines (e.g., Real vs. Traditional) under two evaluation protocols: TSTR and Augmentation. 'Real' refers to the baseline performance using only real data.

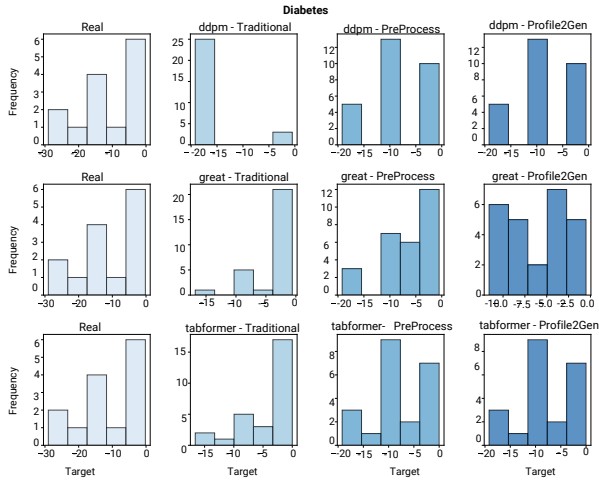

*Figure 8.* Histograms comparing the target value distributions of real and synthetic data for the Diabetes dataset. Synthetic data is generated using different models and techniques (Traditional, Preprocessing, Profile2Gen). This comparison emphasizes the effect of each model and technique on representing minority classes, particularly values in the distribution's tails.

also increases diversity (see Fig. 3). This diversity challenges models to generalize better, reducing overfitting and improving performance on unseen data (Ramanujan et al., 2023). Thus, Profile2Gen balances consistency and diversity, ensuring robust and reliable results.

The iterative profiling process removes hard samples in successive iterations and progressively refines the augmented dataset by preserving core patterns and filtering out problematic samples. Even across multiple random seeds, performance remained stable with minimal variability (see Section L), demonstrating that our approach maintains generalization without overfitting to specific data configurations. Crucially, our results show no significant performance degradation across iterations, confirming that removing hard samples prevents model collapse (Shumailov et al., 2024)—a common issue where generative models lose diversity. Instead, our method balances refinement and variability, ensuring stability and robustness in model performance.

### 5.5. Statistical Significance

In these experiments, non-parametric statistical tests were applied due to insufficient evidence of normality and small sample sizes. The Kruskal-Wallis test assessed performance differences across four groups—Real, Traditional, PreProcess, and Profile2Gen—under two evaluation protocols: TSTR and Augmentation. For post-hoc analysis, the Wilcoxon test was employed, as commonly done in similar studies. All tests were performed at a 95% confidence level ($\alpha = 0.05$). This section focuses on Profile2Gen perfor-

mance. Additional analyses are available in Section P.

#### 5.5.1. TSTR RESULTS

Using the TSTR protocol, we evaluated 30 dataset-model combinations (e.g., Bayesian Network, CTGAN). The Kruskal-Wallis test found significant differences in 96% of cases, confirming substantial performance variations. Post-hoc Wilcoxon tests (Figure 7) further identified these differences. In 91% of pairwise comparisons, Real demonstrated significant differences, often achieving comparable or better performance than Profile2Gen (see Figure 19). However, given the high statistical significance across experiments and the fact that Profile2Gen achieves similar, better, or worse performance depending on the scenario, its overall performance is comparable to Real data. This suggests that Profile2Gen can serve as a viable alternative in situations where real data is scarce, particularly when model performance remains stable across synthetic and real datasets. Traditional methods significantly differed from Real data in 96% of cases, often producing lower-quality synthetic data with higher RMSE, highlighting their unsuitability. PreProcess showed intermediate performance, with RMSE values typically between Real and Profile2Gen. While it occasionally outperformed Profile2Gen, significant differences appeared in 69% of cases, underscoring Profile2Gen's superiority. Therefore, while predictors trained on real data consistently achieve statistically significant performance, Profile2Gen offers the best performance potential. Its synthetic data is statistically more reliable than that generated

by other approaches, making it a promising alternative for predictive modeling tasks.

### 5.5.2. AUGMENTATION RESULTS

Under the augmentation protocol, we evaluated over 90 dataset-proportion-model combinations. The Kruskal-Wallis test found significant differences in 75% of the cases, indicating that at least two methods were performed distinctly. Pairwise Wilcoxon tests (Figure 7) further pinpointed these differences: **Profile2Gen vs. Real:** Statistically significant differences were observed in 60% of experiments, with Profile2Gen often showing improved performance (see Section 5.2). **Profile2Gen vs. Traditional and PreProcess:** Profile2Gen demonstrated significant differences in 79% of cases, often outperforming these approaches.**PreProcess vs. Traditional:** The PreProcess technique significantly improved over Traditional in 80% of experiments. However, its performance varied across datasets, suggesting that preprocessing alone cannot guarantee improved results.

The PreProcess technique, which involves profiling data based on quality before generation, acts as an intermediary step that reduces noise, minimizes group contamination, and facilitates the identification of behavioral patterns. This approach enhances generative model performance, aligning with findings in (Maharana et al., 2022; Dupont et al., 2022). However, due to the inherent instability in generative model training (Goodfellow, 2016), the outputs remain susceptible to quality issues, underscoring the need for profiling techniques. Traditional methods, in contrast, failed to improve predictor performance in 80% of cases, often leading to increased RMSE (see Figures 6 and 7).

### 5.6. About Fairness

In regression tasks, particularly in medical applications, capturing minority events—low-frequency occurrences in the target distribution—is crucial. We aim to preserve these rare cases. By analyzing histograms of the target variable across different synthetic datasets, we assess whether Profile2Gen retains or distorts these events.

As shown in Figure 8, Profile2Gen generally maintains low-frequency events without excessive overrepresentation, unlike the Traditional method, which significantly amplifies event frequencies. While PreProcess performs similarly, it sometimes fails to retain minority classes, particularly with models like DDPM in the diabetes dataset. The full comparison is provided in the appendices. Figure 9 demonstrates that models trained on Profile2Gen data achieve lower RMSE than those trained on Real or Traditional synthetic data, indicating improved data quality. However, when Profile2Gen fails to retain minority classes, RMSE increases, as seen in DDPM. Models like GREAT and Tab-

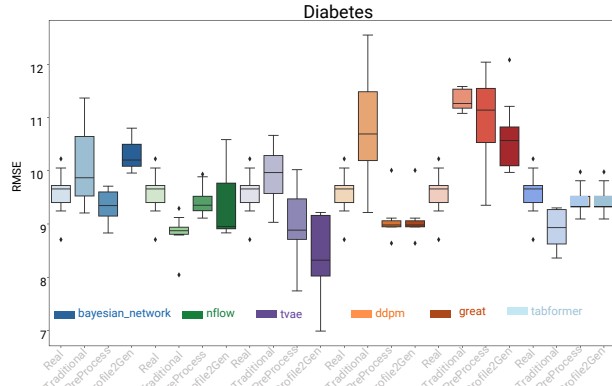

*Figure 9.* This graph shows the performance of generative models trained on synthetic data (TSTR) under three preprocessing scenarios: Traditional, PreProcess, and Profile2Gen, with Real data as the benchmark. RMSE is used as the evaluation metric, and colors represent different models. Results indicate that Profile2Gen-preprocessed models generally outperform others, including those trained on real data, with lower RMSE.

Former perform consistently well, with Profile2Gen yielding competitive results across settings. Overall, Profile2Gen enhances model performance by preserving minority groups and ensuring a more balanced distribution, often outperforming other techniques and, in some cases, even real data.

### 5.7. Ablation on Hard Samples: Are They True Outliers?

We investigated whether the removed hard samples—instances where the model had low confidence but predicted correctly or high confidence but misclassified—were statistical outliers. We selected the Cholesterol dataset for this ablation study and analyzed all generative models using the Z-score to determine whether the removed samples were statistical outliers. We initially applied a threshold of ±3, following the Empirical Rule (68-95-99.7), which considers values beyond this range as anomalies, accounting for less than 0.3% of a normal distribution (Tacq, 2010). However, since our dataset lacked strong evidence of normality, we extended the threshold to ±4 to test a more relaxed outlier detection criterion. The results remained consistent: there was no statistical evidence that the removed samples were outliers, even with the extended thresholds. Thus, our findings suggest that the removal of hard samples is unlikely to be driven by extreme values, reinforcing the idea that these samples are distinct from traditional statistical outliers [2].

---

[2]The source code is available on GitHub

## 6. Conclusion

This paper introduces Profile2Gen, a data-centric framework designed to generate and refine synthetic data, specifically targeting challenges posed by hard-to-learn samples in regression tasks. The method ensures the creation of high-quality synthetic medical datasets by profiling data during both the preprocessing and postprocessing stages. Through extensive experiments across diverse medical datasets, we show that integrating refined synthetic data significantly reduces predictive errors and enhances model reliability. Additionally, extending the DataIQ framework to support regression tasks highlights the flexibility of data-centric approaches in improving predictive modeling. Our findings emphasize the critical role of high-quality synthetic data in ensuring consistent and reliable model performance, bridging the gap between synthetic data generation and its practical application. While RMSE and Wasserstein's distance offer valuable insights into model performance and data similarity, incorporating additional evaluation metrics—such as fairness, privacy, and clinical utility—would provide a more comprehensive assessment of synthetic data quality.

## Impact Statement

This research introduces Profile2Gen, a novel data-centric framework that significantly enhances the quality and utility of synthetic data for regression tasks, particularly in the medical domain. By addressing the challenges posed by hard-to-learn samples, Profile2Gen empowers researchers and practitioners to:

1. Overcome data scarcity in medical research, access to large, high-quality datasets is often limited due to privacy concerns and ethical restrictions. Profile2Gen offers a viable solution by generating realistic synthetic data that can be used for model training, validation, and testing, thereby accelerating research progress.

2. Improve model robustness and generalization by refining synthetic data through iterative profiling; Profile2Gen ensures that models trained on this data are more robust and generalize better to real-world scenarios. This leads to more reliable and accurate predictions, which can significantly affect clinical decision-making.

3. This work contributes to the growing field of data-centric AI, emphasizing the importance of data quality and refinement in achieving optimal model performance. By providing a practical framework for synthetic data generation, Profile2Gen sets a precedent for future data-centric innovations.

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

This appendix provides additional details on the data-centric methods, experimental details, and findings.

## A. Datasets

The Parkinson dataset (Tsanas et al., 2009) consists of various biomedical voice measurements from 42 individuals with early-stage Parkinson's disease. These participants were recruited for a six-month trial of a telemonitoring device designed for remote symptom progression monitoring. The recordings were automatically captured in the patients' homes. The dataset includes subject number, age, gender, time of recruitment date, motor Unified Parkinson's Disease Rating Scale (UPDRS), total UPDRS, and 16 biomedical voice measures. The primary aim of the dataset is to predict the motor and total UPDRS scores from the 16 voice measures. In this work, we focused solely on the total UPDRS scores. The dataset is available on OpenML (Tsanas & Little, 2016).

The Urinary dataset (Czerniak & Zarzycki, 2003) aims to predict the diagnosis of two diseases of the urinary system: acute inflammation of the urinary bladder and acute nephritis. The dataset includes the following features: patient temperature, occurrence of nausea, lumbar pain, urine urgency (continuous need for urination), micturition pains, burning sensation of the urethra, itching, swelling of the urethral outlet, and the diagnosis (inflammation of the urinary bladder or nephritis of renal pelvis origin). The dataset is available on OpenML (Czerniak, 2015). We changed the target to the data column named 'V1', aiming to adapt for a regression task.

The Cholesterol dataset (Detrano et al., 1989) concerning heart disease diagnosis comprises data collected from four different locations: the Cleveland Clinic Foundation, the Hungarian Institute of Cardiology, the V.A. Medical Center, the Long Beach, and the University Hospital, Zurich. Each of these databases contains the same format and attributes. However, there are 76 raw attributes in the datasets, and only 14 attributes are commonly used in most experiments, including in this work. The dataset can be found on the OpenML website (OpenML, 2024d). The dataset is used in regression cases, and it indicates that the 'chol' column is the target. Still, we believe that the data is mislabeled since it corresponds to targets for a classification task. Due to that, we used the 'slope' as our target.

The Body Fat dataset (fat) lists estimates of the percentage of body fat determined by underwater weighing and various body circumference measurements for 252 men. The data comprises information such as age, weight, height, and neck and chest measurements. It contains fifteen columns, including the target column, corresponding to the fatness measure. No more information is available about this dataset; the web pages that describe it are no longer available; only the OpenML is available (OpenML, 2024c).

The Plasma Levels dataset (plasma) (Nierenberg et al., 1989) is a dataset raised from a study with 315 subjects. The patients who had an elective surgical procedure over three years to biopsy or remove a lesion of the lung, colon, breast, skin, ovary, or uterus that was found to be non-cancerous. Plasma concentrations of the micronutrients varied widely from subject to subject. Age, sex, and alcohol consumption have an influence, for instance. The data comprises information about the age, Sex, Smoking status, Vitamin Use, Number of calories consumed per day, and Number of alcoholic drinks consumed per week. A total of fourteen pieces of information were collected. The data target used was the Betaplasma, which corresponds to the Plasma beta-carotene levels. The data are available in (OpenML, 2024b).

The Diabetes dataset (Torgo, 2024) concerns the study of the factors affecting patterns of insulin-dependent diabetes in children. The objective is to investigate the dependence of the level of serum C-peptide on various other factors in order to understand the patterns of residual insulin secretion. The data contains information about the age and insulin deficit and aims to predict the C-peptide level. It is available on (OpenML, 2024a).

### A.1. Dataset Selection

The datasets used in this work were selected based on availability and accessibility due to the well-documented scarcity of public, high-quality medical datasets for synthetic data research. While an ideal selection process would involve curating datasets specifically tailored to diverse medical tasks, we relied on the following practical criteria:

1. Public Availability: We prioritized publicly available datasets to ensure the reproducibility and transparency of our experiments.

2. Tabular Data Format: The datasets selected were in a tabular format, aligning with the requirements of our proposed method, Profile2Gen, which is designed for tabular regression tasks.

3. Medical Relevance: While not exhaustive, the chosen datasets reflect a variety of medical domains, providing a useful starting point for evaluating the effectiveness of synthetic data generation in this field. We acknowledge that the datasets selected may not fully capture the breadth of challenges encountered in real-world medical applications. However, this limitation underscores the need for broader efforts to create and share public medical datasets to advance research in this area.

## B. Frameworks Used

Cleanlab (Northcutt et al., 2021) is an open-source tool designed to improve the quality of machine learning datasets by identifying and correcting label errors. It leverages state-of-the-art algorithms to detect mislabeled data, noisy annotations, and outliers, which can significantly degrade model performance. Cleanlab integrates seamlessly with popular machine learning frameworks, making it easy for data scientists and machine learning engineers to enhance their datasets' integrity and achieve better model accuracy and reliability. It classifies the data as 'hard' or 'easy' based on the likelihood of label errors and the confidence of predictions. This classification helps understand which data points are more challenging for the model to learn and can guide targeted data-cleaning efforts.

DataIQ (Seedat et al., 2022) is a framework that enhances data quality through various preprocessing and validation techniques. It focuses on improving the accuracy and robustness of machine learning models by ensuring the datasets used for training and testing are clean, complete, and free of inconsistencies. DataIQ offers tools for handling missing values, detecting and correcting anomalies, and balancing datasets. DataIQ helps build more reliable and generalizable machine learning models by prioritizing data quality. It classifies the data as 'hard', 'easy', and 'ambiguous' according to the models' uncertainty. The examples with low data uncertainty that the model can correctly predict with high confidence are classified as 'easy', while the ambiguous examples are those with high data uncertainty. Hence, the model cannot predict with confidence, and the examples with low data uncertainty that the model cannot predict, i.e., predicted incorrectly yet with high confidence or which have low confidence for the correct class, are the 'hard' samples.

Each framework works based on a given threshold. This means the classification is based on the threshold's confidence intervals.

## C. DataIqReg

Our tool builds upon the foundational framework of DataIQ (Seedat et al., 2022), bringing a suite of innovative improvements designed to elevate data quality and model performance. While DataIQ focuses on enhancing data accuracy and robustness through preprocessing and validation techniques, our tool refines this approach to better address regression tasks and offer deeper insights into data quality.

The framework supports two modes of use: for models from Scikit-Learn and models from PyTorch, both employing classification metrics. We made adjustments to make it more suitable for regression tasks. The changes began with the choice of the loss function. Originally, the framework used Negative Log-Likelihood Loss (NLLLoss), common in classification problems (PyTorch, 2024b).

$$\mathcal{L}_{\text{NLL}} = -\sum_{i=1}^{N} y_i \log(\hat{y}_i)$$

We replaced this with Mean Squared Error (MSE), which is widely used in regression tasks (PyTorch, 2024a).

$$\mathcal{L}_{\text{MSE}} = \frac{1}{N} \sum_{i=1}^{N} (y_i - \hat{y}_i)^2$$

where $y_i$ is the true target value and $\hat{y}_i$ is the predicted value.

Other modifications include:

1. **Targets:** In the original framework, targets were treated as 'long' for categorical data, but we adjusted them to 'float' to handle continuous targets.

2. **on_epoch_end:** Modified to save predictions instead of ground truth, reflecting the focus on prediction accuracy rather than confidence levels.

3. **gold_labels_probabilities:** Adjusted to use predicted values instead of confidence levels, emphasizing the precision of predictions in regression tasks.

4. **true_probabilities:** Renamed to true_values, representing the actual predicted values (y).

5. **confidence:** For regression, the mean of predicted values is more appropriate than the mean of confidence levels used in classification.

6. **aleatoric uncertainty:** Adapted to calculate the variance of predicted values, analogous to capturing random uncertainty in numerical predictions as in classification problems.

$$\sigma^2 = \frac{1}{N} \sum_{i=1}^{N} (\hat{y}_i - \mu)^2$$

where $\mu = \frac{1}{N} \sum_{i=1}^{N} \hat{y}_i$ is the mean of the predicted values $\hat{y}_i$.

7. **variability:** Similar to standard deviation, providing insights into the variability of prediction metrics.

8. **Correctness:** Interpreted as RMSE (Root Mean Squared Error), a key metric for evaluating regression performance.

$$\text{RMSE} = \sqrt{\frac{1}{N} \sum_{i=1}^{N} (y_i - \hat{y}_i)^2}$$

9. **entropy:** Although originally named entropy, we use R-squared (R2) calculation, which assesses the model's fit to the data and is relevant for evaluating regression model adjustment.

$$R^2 = 1 - \frac{\sum_{i=1}^{N} (y_i - \hat{y}_i)^2}{\sum_{i=1}^{N} (y_i - \bar{y})^2}$$

where $\bar{y} = \frac{1}{N} \sum_{i=1}^{N} y_i$ is the mean of the true target values.

10. **mi (mutual information):** In the context here, mutual information corresponds to the Pearson correlation coefficient, suitable for evaluating the strength and direction of linear relationships between predicted and true values.

$$r = \frac{\sum_{i=1}^{N} (y_i - \bar{y}) (\hat{y}_i - \bar{\hat{y}})}{\sqrt{\sum_{i=1}^{N} (y_i - \bar{y})^2} \sqrt{\sum_{i=1}^{N} (\hat{y}_i - \bar{\hat{y}})^2}}$$

where $\bar{y}$ and $\bar{\hat{y}}$ are the means of the true and predicted values, respectively.

These adjustments ensure that the framework effectively supports and evaluates regression models, aligning metrics and functionalities with the specific requirements of regression tasks. For a better comparison with the original version, see (Northcutt et al., 2021).

The objective of this work was not to introduce DataIQReg but rather to use it as a tool for the proposed technique. However, the DataIQReg source code will be made available, and in future work, we will present experiments and results obtained using this tool.

## D. Computational Cost

To assess computational efficiency, we selected the largest dataset, Parkinson's, which contains approximately 3,500 training samples. We applied the framework selection process (Cleanlab and DataIQ) along with profiling, using two thresholds and a label replacement ratio. These experiments required 3,510 MB of memory and took approximately 3 minutes to complete.

For larger datasets, it is important to consider that memory usage will increase proportionally, as observed in the profiling stage. However, it should be noted that the process itself does not require GPU resources. Memory remains the main concern

at this stage. We utilized an Nvidia RTX4090 to generate synthetic samples, particularly for Transformer and LLM-based models.

When working with datasets larger than 3,500 samples, memory consumption could be calculated based on this scaling factor, considering the amount of data processed. While simulating this for larger datasets may be difficult, we believe this scaling factor can provide a reasonable estimate. It is worth noting that finding large datasets, especially in healthcare, is challenging, and the idea of generating synthetic data is particularly relevant in scarcity scenarios where such large datasets are not readily available.

## E. About the Optimization Process and Sensitivity

The choice of the hard sample threshold is based on the performance of the profiling framework. In Appendix H, we explain this selection process. Specifically, the threshold is determined by identifying the best-performing framework in terms of F1 score across different levels of label flipping. Threshold Selection Process: We tested six proportions of label flipping: 0, 0.02, 0.08, 0.1, 0.20, and 0.25. For each proportion, we evaluated six different threshold values: 0.1, 0.125, 0.15, 0.175, 0.2, and 0.25. The framework that achieved the highest average F1 score across these conditions was selected as the best performer.

**Why do we not treat the threshold as a key factor in the main paper ?** Once the best framework is identified, we use the corresponding threshold that yielded the highest F1 score to profile the data. This process occurs at both the preprocessing and postprocessing stages. Since we always use the best-performing framework, we consider threshold selection a non-critical hyperparameter, as it is inherently optimized within the profiling stage. Formally, let $\mathcal{F}$ be the set of evaluated frameworks and $\mathcal{T}$ the set of tested thresholds. The goal of the process described in the paper is to find the optimal pair $(f^*, t^*)$ such that:

$$(f^*, t^*) = \arg \max_{f \in \mathcal{F}, t \in \mathcal{T}} \mathbb{E}[F1(f, t)] \tag{4}$$

where $\mathbb{E}[F1(f, t)]$ represents the average F1-score across different proportions of label flipping. Sensitivity analysis typically involves testing the robustness of a fixed parameter, but in our case, $t^*$ is not arbitrary — it is chosen as part of an optimization process that depends on $f^*$. Since we have already found the optimal pair $(f^*, t^*)$ for each dataset, testing variations in $t^*$ without re-evaluating $f^*$ would undo the optimization performed and could lead to misleading conclusions. In other words, the threshold is already thoughtfully selected along with the framework, making an additional sensitivity analysis unnecessary.

**Empirical validation of the threshold robustness:** Given that sensitivity was a concern across reviewers, we assessed how the number of hard-profiled samples changes when varying the threshold $T$. To ensure reproducibility, we set a random seed and let NumPy randomly select T from the range (0.05, 0.6) with six distinct values. Figure 10 illustrates the observed behavior using the smallest and largest datasets from our experiments.

The threshold can be understood as a flexibility level —how much confidence is required before a sample is no longer considered "hard." Lower thresholds allow greater flexibility, meaning the model tolerates lower confidence scores. Given that our predictor is a good generalizer, we observed a few samples with low confidence. As expected, increasing the threshold leads to a higher number of hard samples. However, the change follows a smooth, almost linear trend rather than abrupt shifts.

**Empirical validation of the influence of the flipping rate and threshold:** We also assessed how the number of hard-profiled samples changes when varying the threshold or the flipping rate while keeping the other parameter fixed. To ensure reproducibility, we set a random seed. We let NumPy randomly select a fixed parameter value from the range (0.05, 0.6), while the variable parameter was chosen from the same range but with six distinct values. The Figure 11 illustrates the behavior observed.

The results show that dataset characteristics significantly influence sensitivity. For the Diabetes dataset, fixing the threshold and varying the noise level leads to considerable changes in the number of hard samples, which is expected, given that label flipping in small datasets intuitively affects the data distribution more than in larger ones. However, when fixing the flipping level and varying the threshold, the sensitivity is relatively smooth for both datasets. The curves do not exhibit abrupt changes, confirming that selecting the best threshold for the highest-performing framework remains a reasonable and stable choice.

These findings reinforce our original methodology: optimizing , $T^*$ jointly with $f^*$ ensures that the threshold adapts to

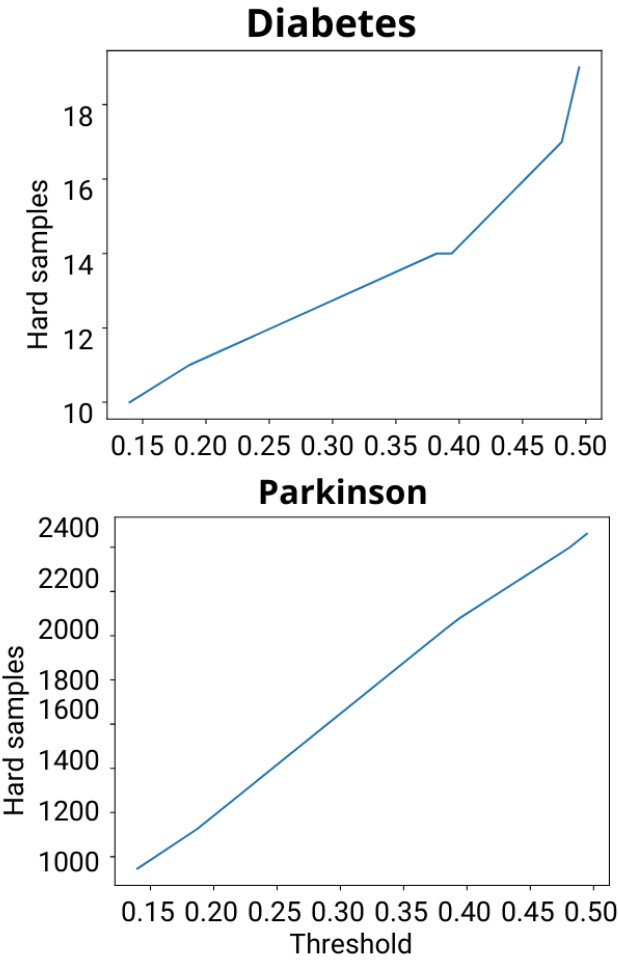

*Figure 10.* Variation of the number of hard samples according to the threshold.

dataset characteristics without introducing unnecessary complexity or computational overhead.

## F. The Generative Models

To generate synthetic data, we utilized four out of the five models employed by Hansen et al. (2023) (Hansen et al., 2023): CTGAN (Xu et al., 2019a), TVAE (Xu et al., 2019a), NFlow (Kobyzev et al., 2021), and Bayesian Network (Young et al., 2009). We decided to maintain these models due to their recognition in tabular data generation.

CTGAN (Conditional Tabular GAN) (Xu et al., 2019a) is a GAN-based model tailored for generating synthetic tabular data, tackling specific challenges inherent in tabular data, such as mixed data types, non-Gaussian distributions, multimodal distributions, sparse one-hot encoded vectors, and highly imbalanced categorical columns. CTGAN accommodates continuous and discrete columns by applying softmax to discrete variables and hyperbolic tangents to continuous variables in the generator's output. To handle non-Gaussian distributions in continuous columns and mitigate issues like the vanishing gradient problem, CTGAN utilizes mode-specific normalization. It employs kernel density estimation to determine the number of modes in a continuous column and applies a variational Gaussian mixture model (VGM) to normalize values within each identified mode.

The TVAE (Tabular Variational Autoencoder) (Xu et al., 2019a) model extends the traditional variational autoencoder (VAE) framework to generate synthetic tabular data. Proposed by the same authors as the CTGAN model, TVAE employs similar preprocessing techniques and adjusts the VAE loss function to suit tabular data generation. It utilizes the evidence of

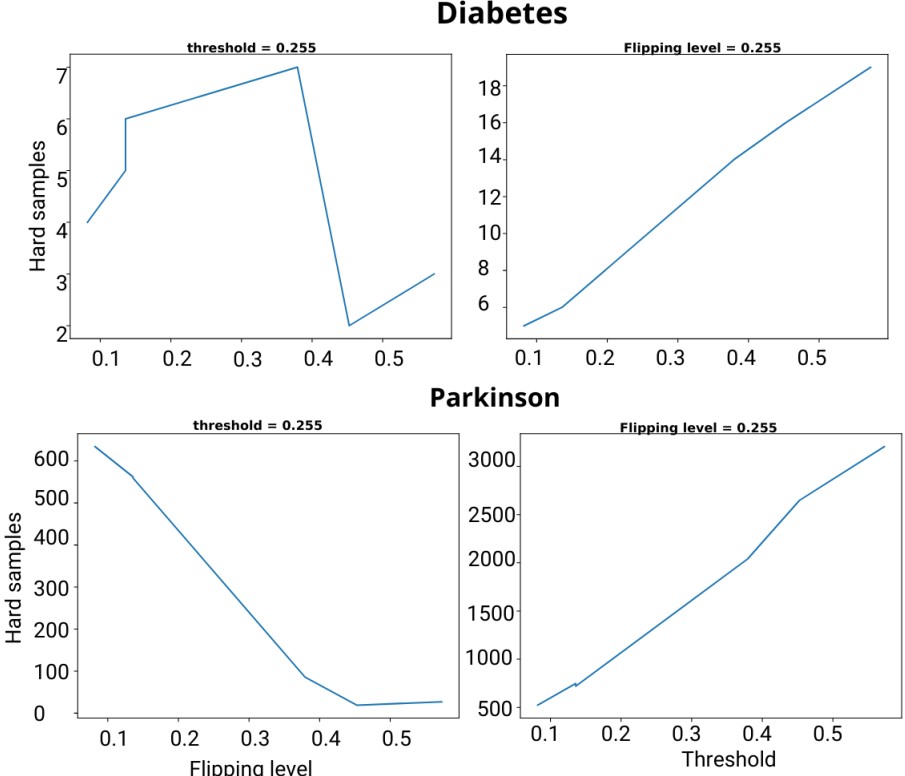

*Figure 11.* Variation of the number of hard samples according to the threshold and flipping rate.

lower-bound (ELBO) loss for training, processing the latent variable through fully connected layers with ReLU activations. Continuous variable outputs are generated using a hyperbolic tangent function and sampled from Gaussian distributions, while categorical variable outputs are generated using softmax functions.

Normalizing Flows (NFlow) (Kobyzev et al., 2021) are generative models that produce tractable distributions, enabling efficient and exact sampling and density evaluation. This model transforms a simple probability distribution into a more complex one by applying a sequence of invertible functions, allowing it to capture complex data distributions efficiently. The transformation is designed to be computationally feasible and invertible, facilitating sampling and density estimation tasks.

The Bayesian Network (Young et al., 2009) is a graphical model representing the joint probability distribution for a set of variables. Each node represents a random variable, and directed edges denote probabilistic dependencies between variables. A fundamental rule of Bayesian networks is that arrows from one node cannot form cycles leading back to the same node. Associated with each node is a conditional probability distribution, specifying the probability of each node's value given the values of its parents in the graph. Bayesian networks can be employed to create multiple synthetic datasets released by an official statistics agency while maintaining the confidentiality of the real data. This allows external analysts to explore associations between attributes of interest and other variables.

We conducted the finetuning using Optuna (Akiba et al., 2019) to minimize the training error in the Urinary dataset, the intermediary length dataset. We saved a pickle file for each model containing the optimized parameters, which are then loaded and utilized during model training. 1 shows it.

## G. Data Preparation

The research utilized four medical datasets available on the OpenML website. The data formats included either pandas DataFrames or ARFF files. The preparation process involved excluding rows with missing values, renaming the target column to 'target', and splitting the dataset into training and test sets with a 67%-33% split, a commonly used proportion in tabular dataset experiments. Additionally, we provided further details on our analyses:

| Model | Setting |
|---|---|
| CTGAN | generator_n_layers_hidden: 2
generator_n_units_hidden: 50
generator_nonlin: tanh
n_iter: 1000
generator_dropout: 0.1188534933545423
discriminator_n_layers_hidden: 3
discriminator_n_units_hidden: 100
discriminator_nonlin: leaky_relu
discriminator_n_iter: 4
discriminator_dropout: 0.01202558406064962
lr: 0.001
weight_decay: 0.001
batch_size: 100
encoder_max_clusters: 1 |
| TVAE | n_iter: 400
lr: 0.0001
decoder_n_layers_hidden: 5
weight_decay: 0.001
batch_size: 64
n_units_embedding: 300
decoder_n_units_hidden: 350
decoder_nonlin: relu
decoder_dropout: 0.02335136745950748
encoder_n_layers_hidden: 3
encoder_n_units_hidden: 100
encoder_nonlin: elu
encoder_dropout: 0.17950057532470087 |
| DDPM | lr: 0.03655328291036083
batch_size: 2918
num_timesteps: 213
n_iter: 8835 |
| NFlow | n_iter: 3200
n_layers_hidden: 7
n_units_hidden: 94
batch_size: 32
dropout: 0.04460459113703746
batch_norm: False
lr: 0.001
linear_transform_type: permutation
base_transform_type: quadratic-coupling |
| Bayesian_network | struct_learning_search_method: tree_search
struct_learning_score: k2 |

*Table 1.* Parameter values for the different models

- We examined low-probability events by plotting histograms to assess whether their distribution was preserved.

- To perform statistical validation, we applied the Shapiro-Wilk test ($\alpha = 0.05$) to check whether the datasets followed a normal distribution.

- We conducted an exploratory data analysis (EDA) to investigate missing values and NaNs, ensuring data completeness and consistency.

- We included standard statistical metrics such as mean, standard deviation, and other relevant descriptive statistics.

## H. Choosing the Framework

The main part of the paper describes that selecting the appropriate framework is a prerequisite before profiling the data. This process involves flipping the labels in the training data by a proportion $\tau$, which means randomly altering the labels of some samples. F12 illustrates this procedure.

Original                   Flipping rate = 0.5

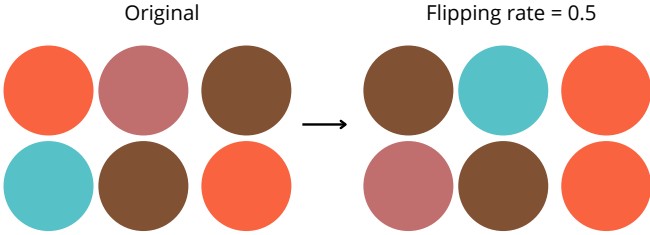

*Figure 12.* Labels flipping. Given a rate $\tau$, the samples have their labels changed randomly. In the figure, each circle has a color, and the color corresponds to its label. After the flipping, the strong pink upper circle is purple, the pastel pink is now green, the upper-left purple is strong pink, and the green is pastel pink.

After the data labels are flipped, the frameworks Cleanlab and DataIQReg are used to profile the data, focusing on identifying hard samples. The framework that achieves the highest F1 score is selected. This process is applied to each dataset, meaning that each dataset may have a different preferred framework, as each framework may be better suited to different distributions.

The F1 score calculation follows the method outlined in (Hansen et al., 2023) and involves identifying hard data and matching it with the (n_correct_hard) data with flipped targets. The recall (r), precision (p), and F1 score (f) are calculated as follows:

$$r = \frac{n\_correct\_hard}{n\_flipped} \qquad p = \frac{n\_correct\_hard}{n\_hard} \qquad f = \frac{2(p \cdot r)}{p + r}$$

where n_hard corresponds to the number of flipped-target samples according to $\tau$.

The frameworks profile the data based on specific thresholds (see their documentation for details). Our experiments tested six proportions for label flipping (0, 0.02, 0.08, 0.1, 0.25, 0.20) and six thresholds (0.1, 0.125, 0.15, 0.175, 0.2, 0.25) for both frameworks. The final decision was based on the average F1 score, with the framework showing the highest average score being chosen for data profiling. If both frameworks achieved the same average F1 score, one of them was selected randomly. The threshold used for profiling the data after the framework selection is the one that achieved the highest F1 score.

## I. Profiling the Synthetic data

As mentioned in the main paper, the framework selection process is also applied to synthetic datasets. Following this, data identified as 'hard' is removed from the synthetic dataset. The synthetic datasets produced by the 'Profile2Gen' stage contain less data than those from the 'Traditional' and 'PreProcess' stages. Figure 13 illustrates the differences in data length between the 'Original' and 'Profile2Gen' datasets. On average, the proportion of hard data across the datasets is approximately 10%.

## J. On the Generalization and Diversity

Figure 14 displays the Wasserstein distance between synthetic samples and real samples across the stages. This pattern suggests that the stages introduce more variation in the synthetic data, potentially creating a broader range of scenarios and examples.

The dataset Urinary highlights that the synthetic data for this dataset also has a high degree of consistency and uniformity, which indicates limited data diversity since the synthetic data closely replicates the real dataset without introducing variability.

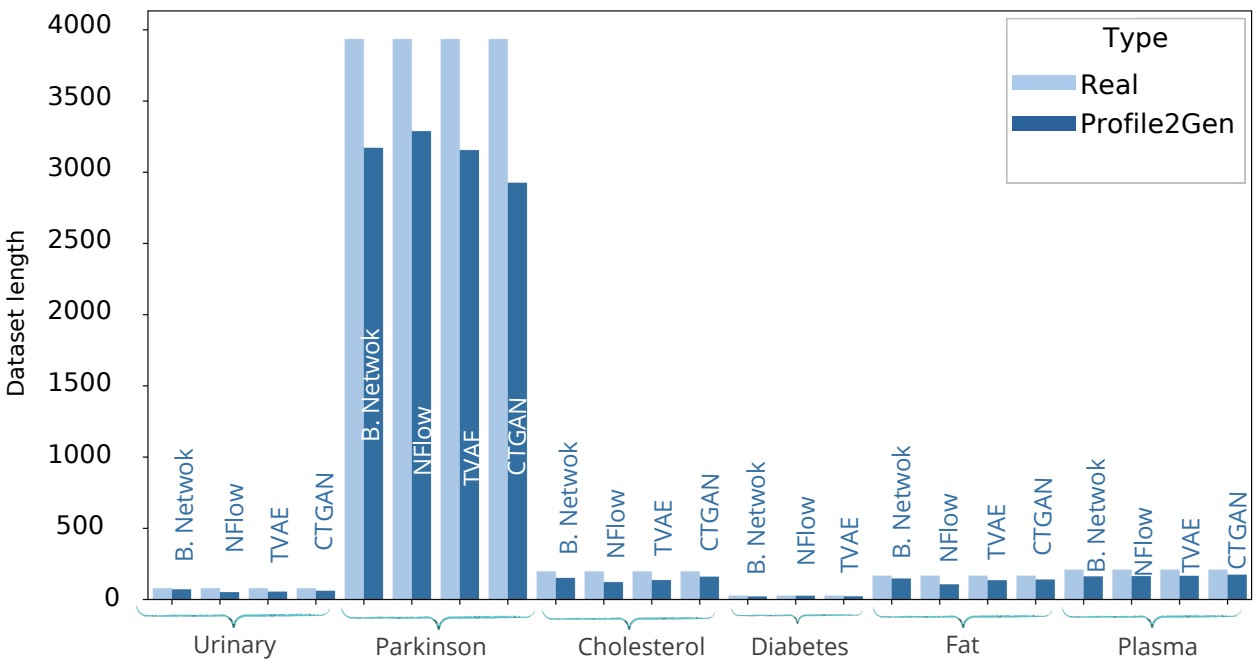

*Figure 13.* Comparison of Dataset Lengths: This figure compares the length of the original dataset with the dataset obtained in the 'Profile2Gen' stage. The x-axis represents the generative model and dataset, while the y-axis shows the number of samples.

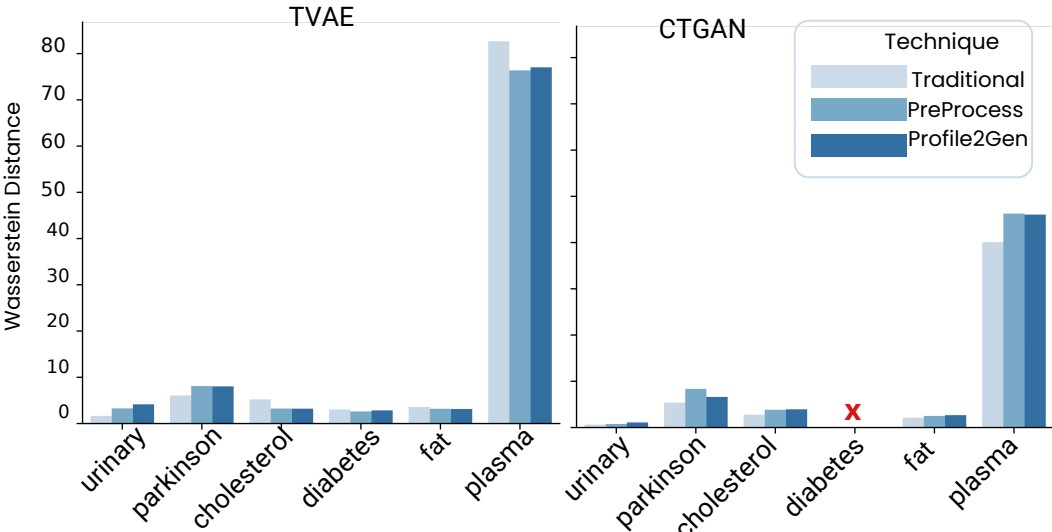

*Figure 14.* Wasserstein Distance across Data Stages for the Generative Models. The grey bars represent the Wasserstein distance between the synthetic data generated in the 'Traditional' stage and the real data. The light blue bars correspond to the 'PreProcess' stage, and the dark blue bars indicate the 'Profile2Gen' stage. The y-axis displays the Wasserstein distance values, while the x-axis corresponds to the evaluated datasets. The red X means insufficient data to train the CTGAN model for the diabetes dataset.

However, datasets like plasma and Parkinson's, which show greater distances across models and stages from the real samples, may reflect higher data diversity. The lower similarity suggests that the stages introduce more variation in the synthetic data, potentially creating a broader range of scenarios and examples.

Considering a model level, the main paper displayed that in the Bayesian network model, the Wasserstein distance between the real and synthetic data remained low across almost all the datasets, particularly in the 'Traditional' stage. This trend

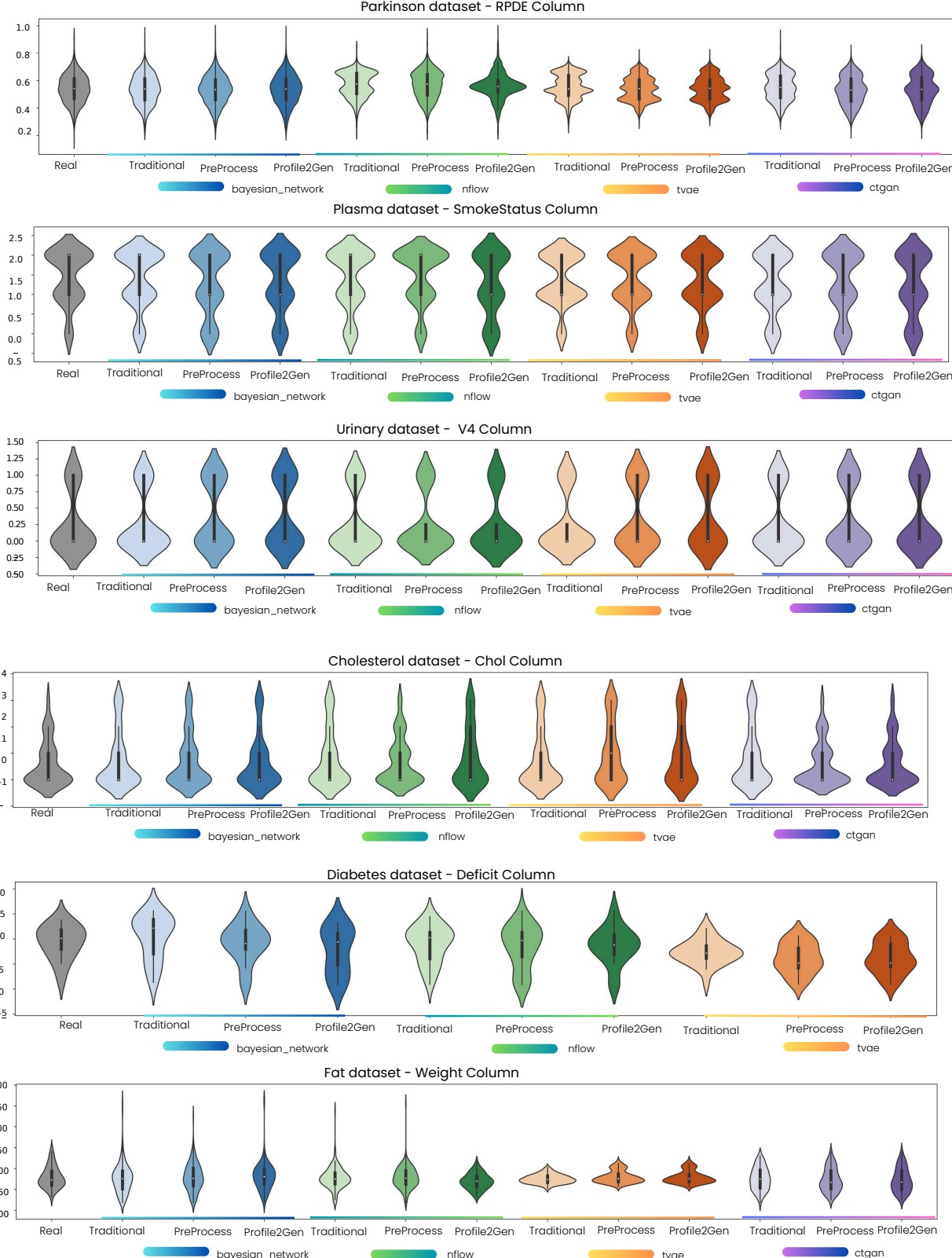

*Figure 15.* Violin plots for a randomly chosen column. Each graph corresponds to a dataset. In each dataset, the real distribution is shown (first violin), along with violin groups delineated by a horizontal line, all using the same color palette. Each group represents a generative model. A caption for the groups is provided below each graph.

underscores the model's ability to generate synthetic data closely resembling the real dataset when Traditional is applied. The model follows the general tendency - the similarity decreases toward the 'Profile2Gen' stage - except the Parkinson dataset, where there is a rise in distance was observed in the 'PreProcess' stage, followed by a subsequent decrease in the 'Profile2Gen' stage, indicating a possible recovery of data quality after initial divergence.

When considering the NFlow model, the urinary dataset demonstrated the lowest Wasserstein distance values across all stages. Despite exhibiting an unusual behavior with higher dissimilarity in the 'PreProcess' stage compared to the 'Profile2Gen' stage, this dataset is the most consistent in maintaining similarity to the real data. In contrast, the plasma dataset showed the highest distance values, indicating significant divergence from the real data.

In the case of the TVAE model, Figure 14, the Urinary dataset behaved differently from the previous models. This model is particularly distinctive since plasma, cholesterol, diabetes, and fat datasets particularly have the 'Traditional' stage showing higher distances than subsequent stages, where the PreProcess stage registered the lowest distance. However, the Parkinson and urinary datasets deviated from this pattern, with the 'Traditional' stage yielding the lowest distance, which increased progressively in the 'PreProcess' and 'Profile2Gen' stages.

The CTGAN model could not be applied to the diabetes dataset due to insufficient data, limiting its analysis. Among the datasets analyzed, the Urinary dataset showed the lowest distance across all stages, reinforcing its robustness and possible lack of diversity in the training samples. The Parkinson and plasma datasets exhibited the highest distance in the 'PreProcess' stage. As the general pattern, the cholesterol and fat datasets showed the lowest distances during the 'Traditional' stage, with subsequent stages revealing nearly identical distance values.

Figure 15 indicates that Wasserstein distance metrics and reveals that the 'Profile2Gen' stage across most models and datasets exhibits a distribution shape less similar to the real data than the initial 'Traditional' stage. Despite the observed trend of increasing dissimilarity, the Urinary dataset maintained consistent mean values across all generative models and stages, closely mirroring the real data. For the plasma dataset, the mean value remained unchanged when using the TVAE and CTGAN models, though it initially differed from the real data. In contrast to other models, the mean began similarly to the real data but shifted as the processing stages advanced. However, the minimum and maximum values exhibited convergence by the' Profile2Gen' stage, reducing the overall dissimilarity. The Parkinson dataset presented a less consistent pattern, with means generally aligning with the real data but without a clear trend in the minimum and maximum values.

Therefore, as we advance in the stages, the data becomes more diverse and less faithful to the real data. This may be more suitable for tasks such as augmentation. However, some datasets, such as urinary resistance, remain more similar to the real data and are more suitable for training tasks.

There is a trade-off between generalization and diversity. Increasing diversity is a positive point since, in training ML models, increasing the size of the dataset does not necessarily imply improving performance and data, and can cause the opposite effect if the data used for augmentation is 'more of the same' (Bailly et al., 2022). However, increasing generalization too much could make the data in a dissimilar distribution to the point that we are introducing noise when mixing it with the real data. Consequently, an opposite effect is obtained.

## K.  On the Trade-off Between Generalization and Diversity and Information Loss

Profile2Gen, which incorporates postprocessing, reduces Wasserstein's similarity between real and synthetic samples. This indicates that the generated samples are less similar to real data compared to other techniques. Here is the highlight: The similar samples, which have the same statistical characteristics, are a generalization of the real ones, while the diversity concerns about the samples that are not too similar but follow the distribution patterns (Alaa et al. (2021) (Fekri et al., 2019)). When analyzing the dissimilarity alongside the profile2gen samples and their distributions, we observe that the generated samples are not only different but also diverse while still following the patterns of the original distribution. The lower similarity indicates that Profile2Gen prioritizes more variable synthetic data, potentially creating a broader range of scenarios and examples. Consequently, postprocessing increases diversity at the cost of generalization since this set of samples is diminished in the synthetic dataset. Higher diversity is generally beneficial in tasks such as data augmentation, where increasing the dataset size does not necessarily improve model performance unless it introduces novel and meaningful variations. However, a lack of generalization may cause the synthetic data distribution to diverge too much from the real data, leading to a loss of fidelity and entering the previous trade-off discussed by Alaa et al. (2021), the diversity—fidelity trade-off.

Even with the trade-off, the critical edge-case information is not lost. It means the low-probability events, which, despite being underrepresented and important in the technique, have a synthetic representativity. Demonstrating that, overall, Profile2Gen enhances model performance by preserving minority groups, which correspond to events with low representativeness in the dataset (see Section 5.6. However, the technique's effectiveness varies depending on the generative model adopted. In cases where the technique fails to capture and represent minority data properly, a performance drop is observed. To understand why Profile2Gen better preserves these minority classes, we hypothesize that separating data groups during preprocessing allows the generative model to learn distinct distributions more effectively. In the context of Generative Adversarial Networks (GANs), let $G_\theta$ represent the generator with parameters $\theta$, and $D_\phi$ the discriminator with parameters $\phi$. The standard GAN optimization objective is given by:

$$\min_\theta \max_\phi \mathbb{E}_{x \sim p_{\text{real}}(x)} \left[ \log D_\phi(x) \right] + \mathbb{E}_{x' \sim G_\theta(z)} \left[ \log(1 - D_\phi(x')) \right]$$

where $p_{\text{real}}(x)$ is the real data distribution, and $G_\theta(z)$ generates synthetic samples from a latent noise distribution $z \sim p(z)$.

By training separate generative models for different subgroups, we effectively decompose $p_{\text{real}}(x)$ into **conditional distributions** $p(x|c)$, where $c$ represents a specific subgroup. This modifies the objective function to:

$$\min_\theta \max_\phi \sum_c \mathbb{E}_{x \sim p(x|c)} \left[ \log D_\phi(x) \right] + \mathbb{E}_{x' \sim G_\theta(z|c)} \left[ \log(1 - D_\phi(x')) \right]$$

This forces the generator to learn the distribution of each subgroup separately, reducing early collapse and improving representation for minority groups. However, synthetic data inherently introduces noise, and excessive divergence from $p(x|c)$ may result in synthetic distributions that fail to capture the underlying characteristics of minority classes. This is precisely why Profile2Gen proves more effective than PreProcessing (which lacks postprocessing). When analyzing the histograms in Figure 8 in the main paper, we observe that the PreProcessing technique alone overrepresents certain classes. However, postprocessing mitigates this overrepresentation.

We hypothesize that this correction occurs because overrepresented samples are categorized as hard samples during postprocessing. This reclassification aligns the synthetic data distribution more closely with the real data while preserving meaningful diversity. Since this adjustment is based on model confidence scores, it ensures that the selected synthetic samples remain representative of the original dataset. Consequently, postprocessing refines the synthetic data, ensuring it retains key structural properties while benefiting from the broader variations introduced by generative models.

## L. Profiling Iteratively

This study extensively evaluated data augmentation methods by iterating over the provided data from the 'Profile2Gen' stage. The Diabetes dataset was selected as a representative case due to its relatively smaller size, allowing for many experimental executions. This choice is instrumental in understanding the behavior of the models under various augmentation conditions, especially given the dataset's constraints.

We employed three generative models, tested them across four different proportions of added data, and varied the random seeds three times for each proportion to ensure the robustness of our findings. Each dataset underwent multiple iterations, with the evaluation process thoroughly applied at each step to assess performance consistently.

Despite using four different augmentation proportions, only two of them completed the process. Due to the data removal in each iteration, there was insufficient data to augment for proportions of 50% or more. As shown in Figure 17 d, each model used a varying number of iterations, reflecting the different quantity of data removed for each model. Figure 18 illustrates the quantity of data removed per iteration. It is important to note that the 'Profile2Gen' stage also involved data removal, depicted in Figure 13.

Figure 17 illustrates the iterative process employed in this evaluation. It is crucial to note that each of the twelve predictor models was utilized during this assessment. This approach enabled us to observe the performance variations and identify any potential indications of model collapse.

We conducted these experiments to investigate whether, similar to the findings in (Shumailov et al., 2024), the models exhibit a learning plateau and increased error when using each mode of profiled data or if the opposite occurs. This could indicate either a model collapse or improved data refinement.

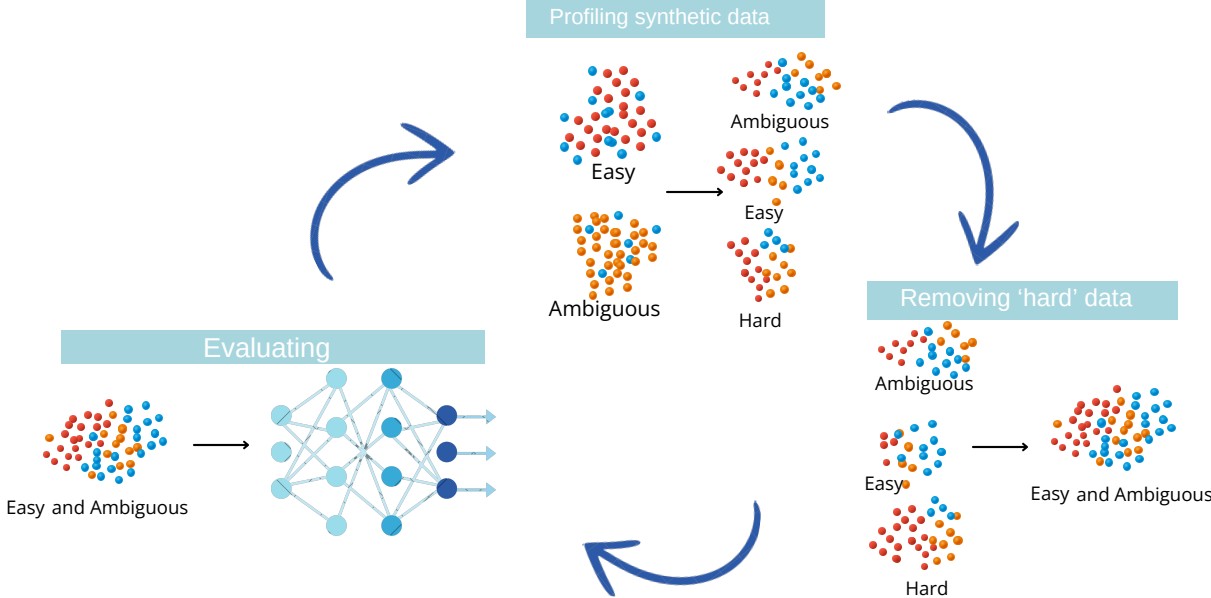

*Figure 16.* Profiling Until Breaks: The profiling process for the synthetic data from the 'Profile2Gen' stage continues until an error occurs in the framework used. In each iteration, synthetic data is combined with real data, and an augmentation evaluation process is conducted to assess performance.

Upon analyzing the results, it is evident that the model's performance demonstrated stability throughout the iterations, with no significant degradation in performance, even as the quantity of added data increased and the data used were more diverse. The continuous and consistent performance across different random seeds and proportions of augmented data suggests that the models did not experience collapse. The robust nature of these models under varying conditions reinforces the reliability of the generative approaches applied, even when challenged by a smaller dataset like Diabetes.

Furthermore, the RMSE trends in the figures indicate a balanced and controlled behavior across iterations, signifying that the models maintained their predictive power without overfitting or underfitting, which could be symptoms of collapse. The gradual and consistent changes in performance metrics across iterations and proportions confirm that the models adapted well to the augmented data. This further supports the argument that the models did not collapse under the experimental conditions tested.

The evidence suggests that the models were resilient and capable of handling the augmented data effectively without experiencing any collapse. This robustness is a promising indication that by profiling the synthetic data, we are increasing the robustness and reliability of the models, enhancing their applicability and value in predictive modeling.

## M. TSTR

Figure 20 shows that for the Urinary dataset, when analyzing the performance of the Bayesian Network model, the 'Traditional' stage shows a high RMSE value, indicating poor initial performance. In the 'PreProcess' stage, there is a reduction in RMSE, resulting in a performance improvement. Profile2Genly, in the 'Profile2Gen' stage, the model exhibits the lowest RMSE, standing out with the best overall performance among all stages. Considering the generative model NFlow, the RMSE is high in the 'Traditional' stage, with a reduction during the 'PreProcess' stage. In the 'Profile2Gen' stage, the model achieves the lowest RMSE among all stages. Both the TVAE and CTGAN models started with high RMSE values in the 'Traditional' stage and suffered an increase in 'PreProcess', but the 'Profile2Gen' remains the lowest RMSE. This pattern of the models is consistent across the Parkinson and Fat datasets.

Figure 21 shows that in the Cholesterol dataset, the performances obtained with the Bayesian Network model show a high RMSE value in the 'Traditional' stage, indicating poor initial performance, which progressively decreases in subsequent stages, with the 'Profile2Gen' stage achieving the best performance. However, this progressive error reduction does not occur for the other models; the 'Traditional' stage still has the highest RMSE, while the 'PreProcess' stage achieves the lowest

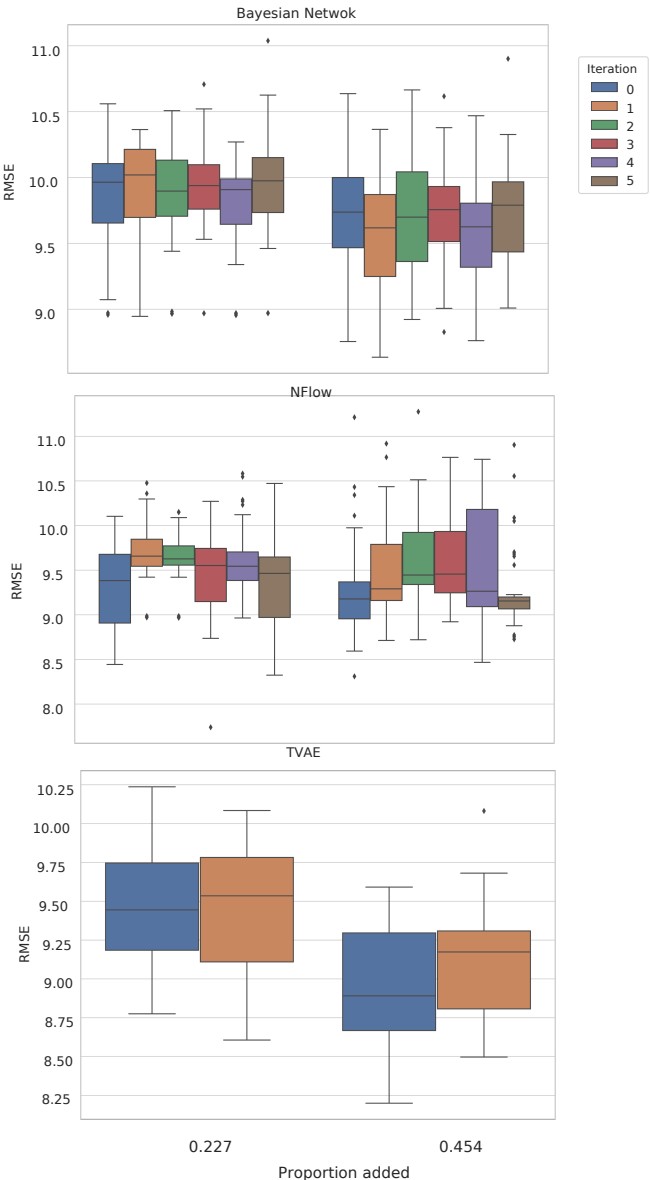

*Figure 17.* Performance of Twelve Models in Augmentation Tasks: Evaluation of the twelve models across each iteration of the profiling-hard-removal process during data augmentation.

RMSE, which slightly increases in this dataset's 'Profile2Gen' stage. Although the increase compared to the 'PreProcess' stage, the 'Profile2Gen' stage still performs better than the 'Traditional' stage, indicating that using the proposed data-centric AI approach remains the best method for generating synthetic data.

For the Diabetes dataset, considering the Bayesian Network model, the best performance is observed in the 'PreProcess' stage, surpassing the real data's performance. In this model, the 'Traditional' stage shows the highest performance variation, but the 'Profile2Gen' stage ends with the highest average RMSE. For the NFlow model, the 'Traditional' stage has the lowest RMSE and the best overall performance, achieving an error lower than the real data. The other two stages also show errors lower than the real data, but the 'Profile2Gen' stage performs better, with an average close to the 'Traditional' stage but with higher variability. For the TVAE model, the 'Traditional' stage has a performance inferior to the real data, while the 'PreProcess' and 'Profile2Gen' stages have a performance with RMSE lower than the real data, with the 'Profile2Gen' stage achieving the lowest RMSE.

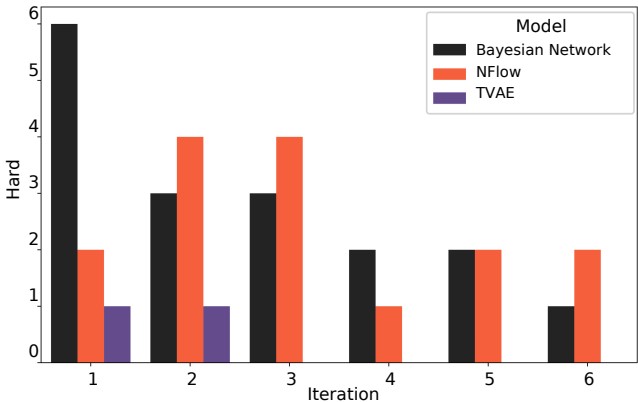

*Figure 18.* Number of hard data removed in each iteration.

Profile2Genly, Figure 22 shows that for the Plasma dataset, the 'Profile2Gen' stage performs better than the real data only for the CTGAN model. For the other models, the overall performance of the 'Profile2Gen' stage was equal to or lower than the 'Traditional' stage, with the 'PreProcess' stage showing the lowest RMSE of all, though not lower than the real data.

Overall, the 'Profile2Gen' stage presents the lowest RMSE compared to traditional techniques despite having less data (see supplementary material). Achieving the best performance compared to the baseline's performance training using only synthetic data is still a challenge. This approach could not achieve it, but it presents an evolution.

## N. On the Baselines Choice

We appreciate your feedback regarding the justification for choosing baseline models. The baseline models we selected adhere to the following criteria:

- 1. They represent traditional techniques primarily used for generating synthetic data, allowing us to demonstrate the impact of preprocessing and postprocessing phases.

- 2. We specifically chose a preprocessing technique to highlight that relying solely on it may not be sufficient for optimal performance.

The choice of baseline models (Traditional and PreProcess) was motivated by the need to compare our framework (Profile2Gen) with common approaches and a data-centric baseline approach. The Traditional method represents the generation of synthetic data without any type of profiling. The PreProcess method represents a data-centric approach that uses profiling in the preprocessing step but not in the postprocessing. These comparisons allow us to isolate the impact of the different steps of Profile2Gen and demonstrate the effectiveness of our approach. Specifically regarding the preprocessing step, we intentionally selected a profiling framework optimized for each dataset. This framework is chosen through an optimization process to identify the most suitable preprocessing method for each specific dataset. We did not compare our approach to traditional methods like simple feature selection, as we believe these methods do not adequately capture the complexity and improvements provided by our methodology. Using our profiling framework as a baseline, we demonstrated the added value of combining it with a postprocessing phase, which significantly enhances performance. However, we acknowledge that the main text should include a more detailed justification for the choice of baseline models. Due to space limitations, we plan to expand on this discussion in the appendix, with an explicit mention in the main paper. This will allow us to keep the main text focused on the methodology and results while thoroughly explaining our baseline model selection process.

## O. Augmenting

We conducted an augmentation experiment using four proportions of synthetic data, with four different random seeds for each proportion, across six datasets and four generative models. Due to the high volume of experiments, the results for the Cholesterol dataset are not yet available but will be updated as soon as they are completed.

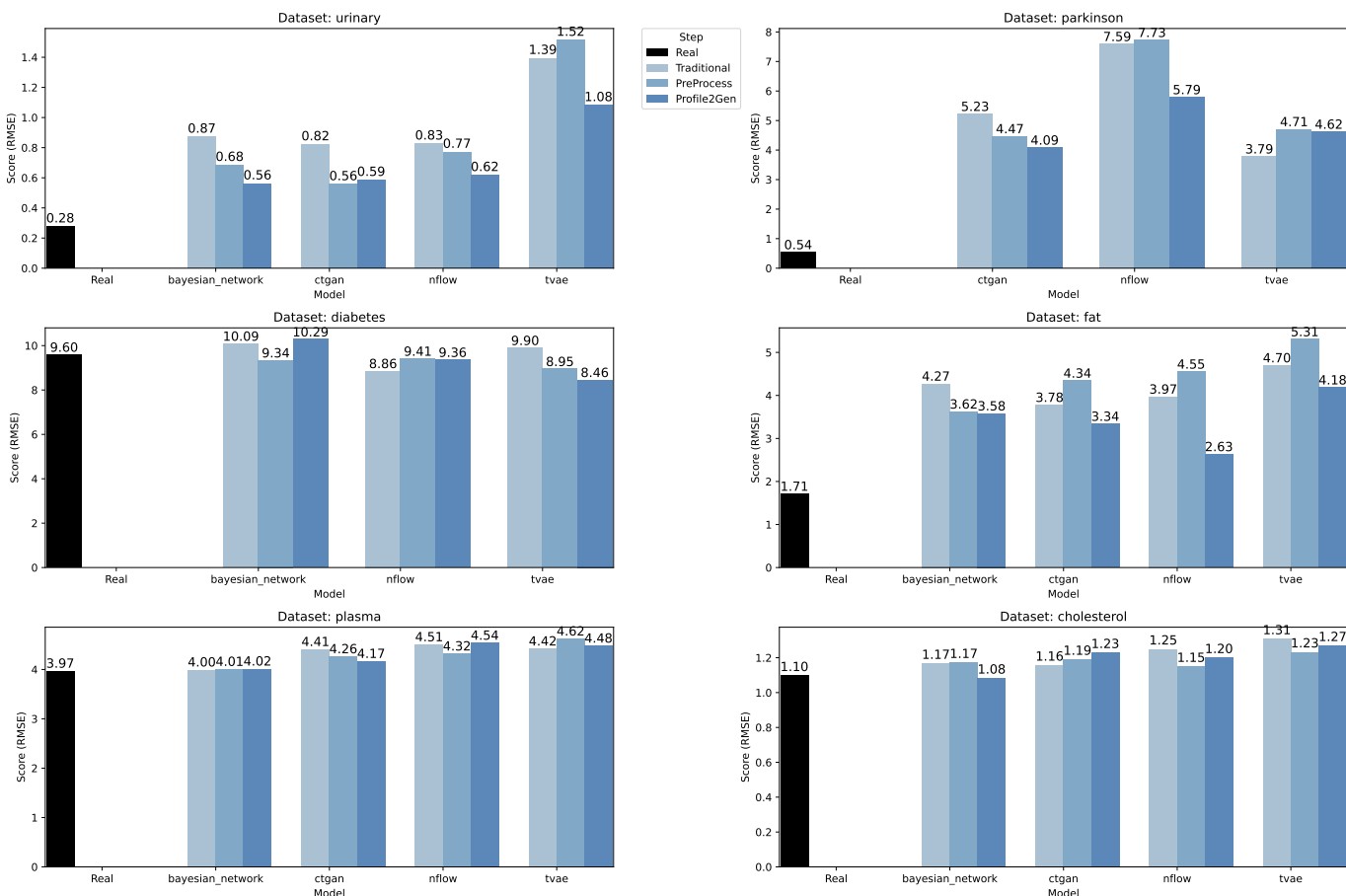

*Figure 19.* Average improvement per model and dataset using the TSTR protocol. Each bar represents the average performance of a generative technique, evaluated across 12 predictors. Bar groups correspond to the generative models, while each plot represents a different dataset. This plot is one version of the figure 5, including the real (TRTR) performance. Each bar represents the RMSE according to the generative model. The number on top of the bars represents the RMSE.

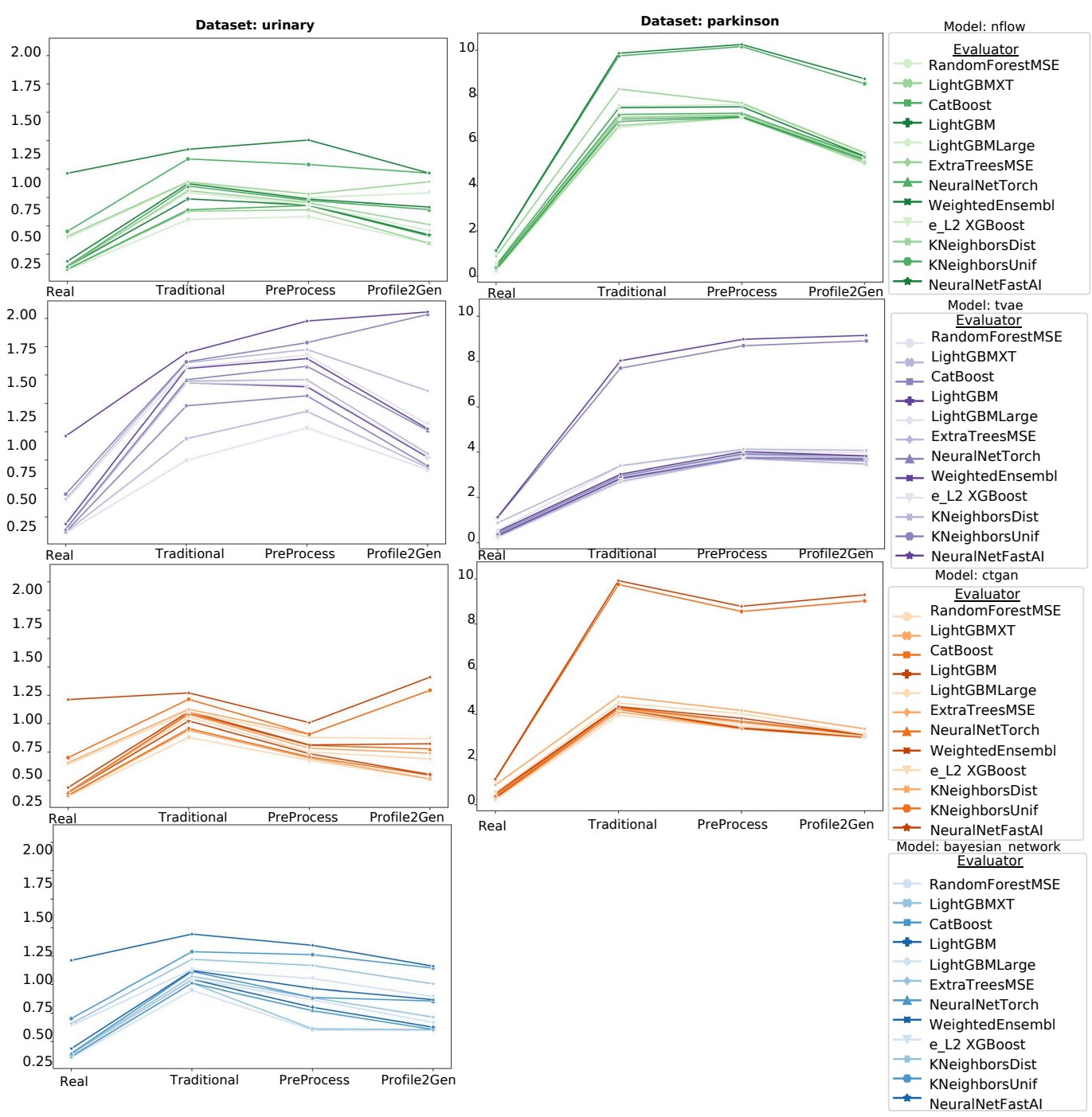

*Figure 20.* Line plots depicting RMSE across the data stages for the twelve models employed in the TSTR task.

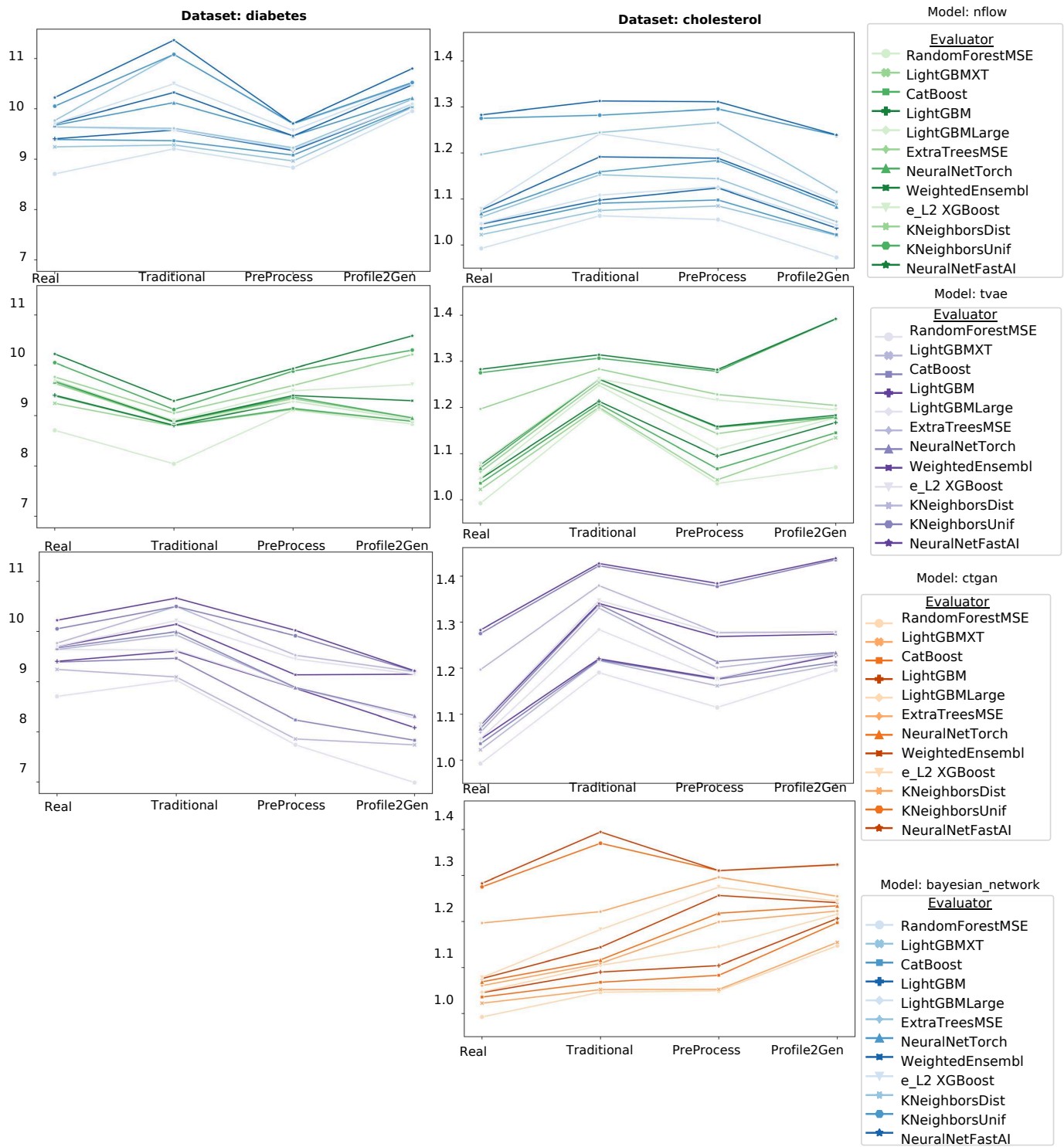

*Figure 21.* Line plots depicting RMSE across the data stages for the twelve models employed in the TSTR task.

We analyzed the results at various levels of granularity. Specifically, we evaluated the overall results across all experiments by dataset and model, including all augmentation proportions, random seeds, and evaluation models.

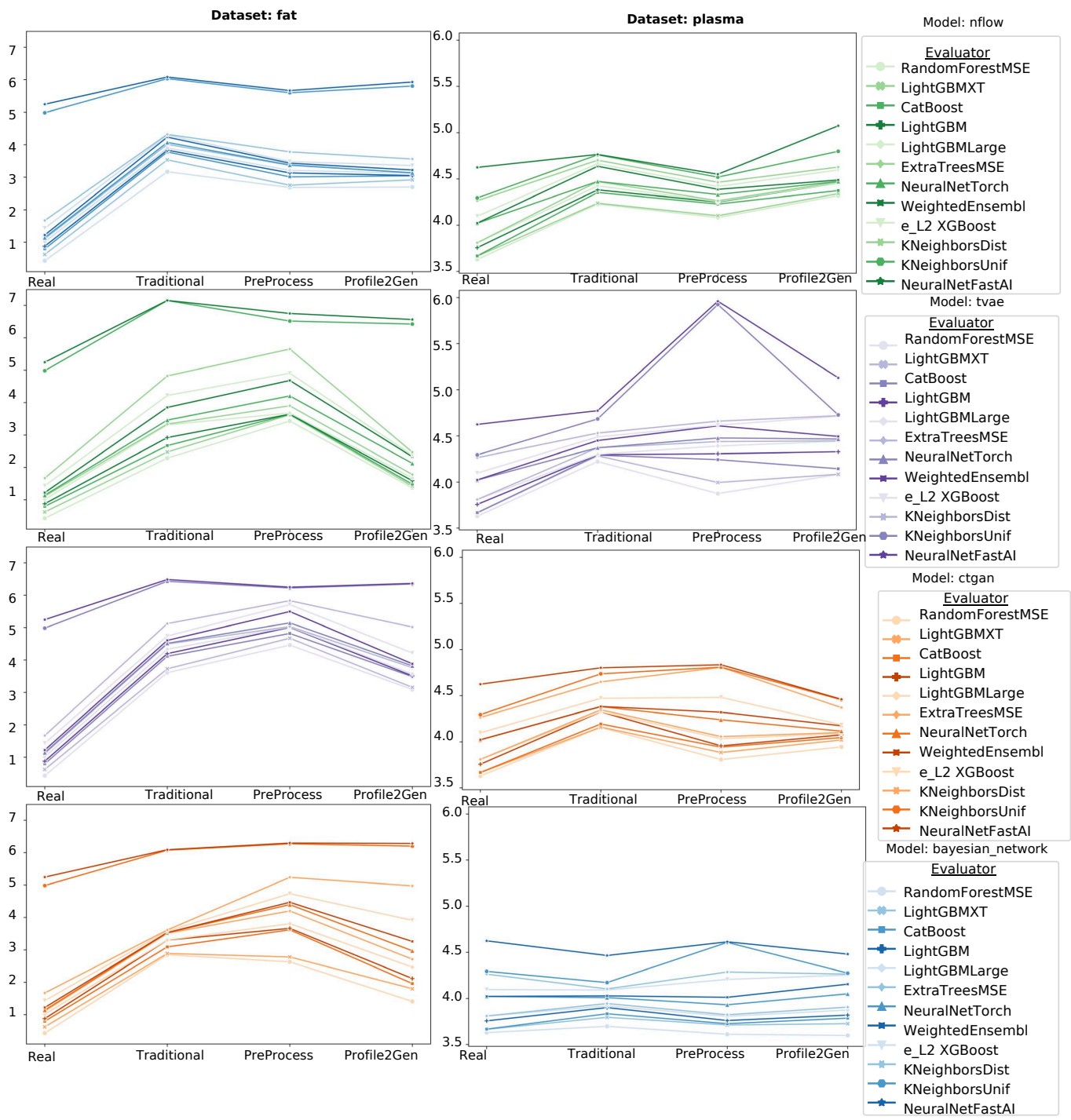

*Figure 22.* Line plots depicting RMSE across the data stages for the twelve models employed in the TSTR task.

## O.1. General

Figure 23 illustrates the results for the Urinary dataset. The 'Traditional' stage exhibited high RMSE values for the Bayesian Network model, indicating that traditional data generation methods negatively affected the model's performance. The 'PreProcess' stage significantly reduced RMSE, while the 'Profile2Gen' stage achieved the lowest RMSE values. For the

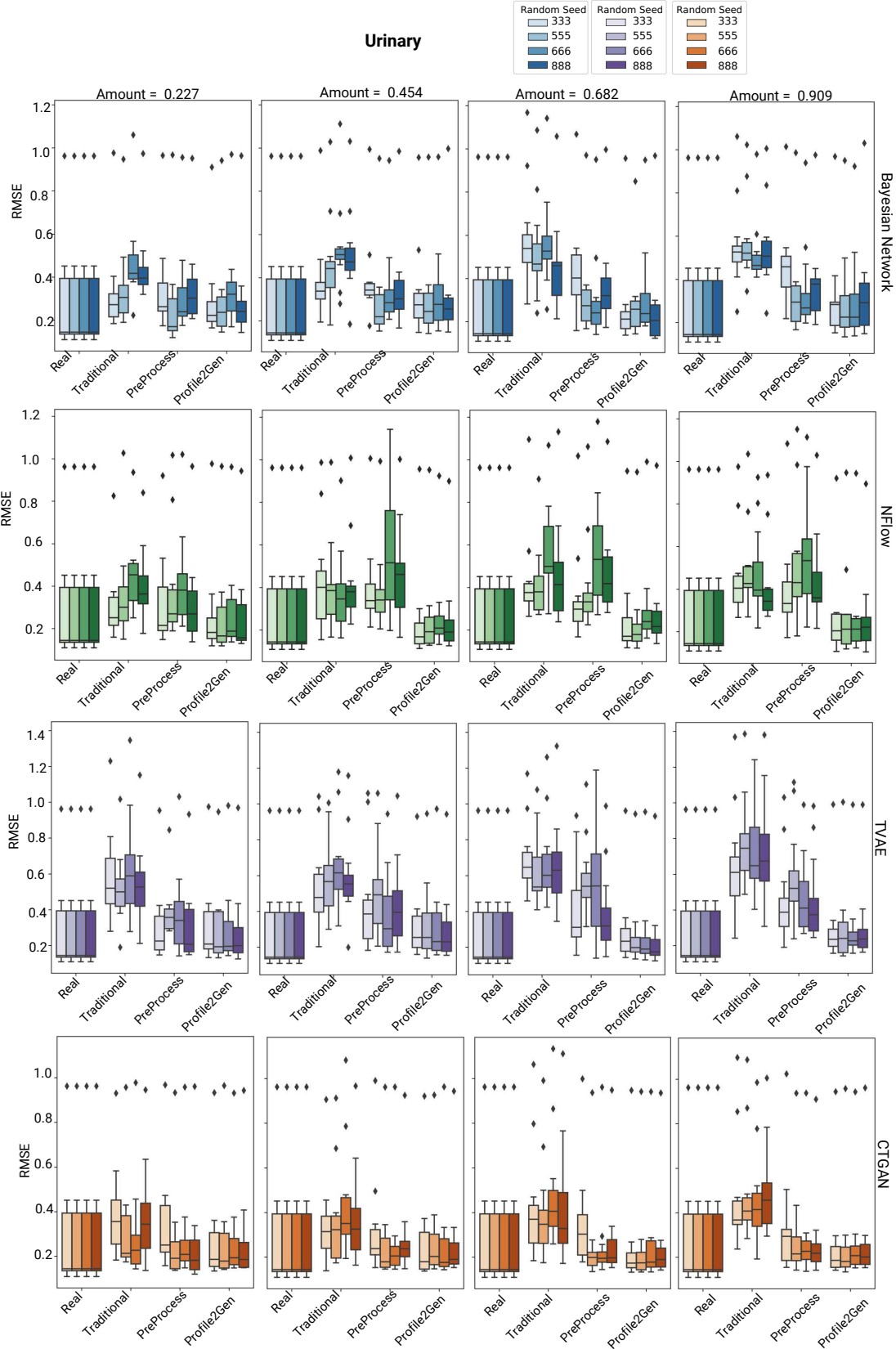

*Figure 23.* Augmentation performance across random seeds.

NFlow model, the 'Traditional' stage also resulted in relatively high RMSE values, with improvements in the 'PreProcess' stage and further reductions in the 'Profile2Gen' stage, though not as pronounced as in the Bayesian Network model. The TVAE and CTGAN models displayed high RMSE values in the 'Traditional' stage, improved in the 'PreProcess' stage, but with high variability. The 'Profile2Gen' stage showed reduced RMSE, although CTGAN exhibited elevated variability.

The results for the Parkinson dataset (see Figure 24) mirrored those of the Urinary dataset. The Bayesian Network model demonstrated high RMSE values in the 'Traditional' stage, which decreased in the 'PreProcess' stage, and the 'Profile2Gen' stage achieved the lowest RMSE values. The NFlow model showed a similar pattern, with high RMSE values in the 'Traditional' stage and improvements in the 'PreProcess' stage, with minimal difference between the 'PreProcess' and 'Profile2Gen' stages. The TVAE and CTGAN models followed this pattern, with high RMSE values in the 'Traditional' stage and improvements in the 'PreProcess' stage. The 'Profile2Gen' stage saw further reductions in RMSE, though CTGAN showed high variability.

Figure 25 presents the results for the Plasma dataset. The Bayesian Network model exhibited a pattern consistent with previous results, with higher RMSE values in the 'Traditional' stage, reductions in the 'PreProcess' stage, and the lowest RMSE values in the 'Profile2Gen' stage. For the NFlow model, the 'Traditional' stage had relatively high RMSE values, with improvements in the 'PreProcess' stage and a similar reduction in the 'Profile2Gen' stage. The TVAE and CTGAN models showed high RMSE values in the 'Traditional' stage, improved in the 'PreProcess' stage, and further reduced in the 'Profile2Gen' stage, although CTGAN maintained high variability. Notably, in this dataset, the average performance of the NFlow, TVAE, and CTGAN models in the 'Profile2Gen' stage exceeded the baseline (Real).

The results for the Fat dataset followed a similar pattern (see Figure 26). The Bayesian Network model exhibited high RMSE values in the 'Traditional' stage, followed by improvements in the 'PreProcess' stage. The 'Profile2Gen' stage achieved the lowest RMSE values, highlighting the effectiveness of the proposed approach. The NFlow model also started with high RMSE values in the 'Traditional' stage, which were further improved in the 'PreProcess' stage. Compared to both stages, the 'Profile2Gen' stage reduced RMSE, achieving performance similar to the baseline. The same occurred with the CTGAN model. For the TVAE model, the 'Traditional' stage began with high RMSE, which increased in the 'PreProcess' stage. However, the 'Profile2Gen' stage reduced RMSE, bringing the performance closer to the 'Traditional' stage.

Profile2Genly, Figure 27 shows that the Diabetes dataset, the Bayesian Network model, followed the same pattern observed previously, with high RMSE values in the 'Traditional' stage and improvements in the 'PreProcess' stage. The 'Profile2Gen' stage increased RMSE values, leaving the best performance for the 'PreProcess' stage, similar to the 'Traditional' performance. The NFlow model repeated the behavior of the previous model. For the TVAE model, as in the other datasets, RMSE values started high in the 'Traditional' stage. The 'PreProcess' stage resulted in improvements, and the 'Profile2Gen' stage further reduced RMSE, leading to the best performance, surpassing the baseline.

## O.2. Proportions

We also analyzed the impact of synthetic data proportions across different datasets and generative models, and the behavior of these models when varying the proportions and random seeds. Figures 27, 23, 24, 25, 26provide valuable insights into the optimal proportions of synthetic data to add.

Adding 22.7% (0.227) of synthetic data generally does not cause drastic changes in performance, remaining close to the baseline of real data. When 68.2% (0.682) of synthetic data is added, it tends to be a critical point where many models show a more significant deviation from the baseline, indicating that increasing this quantity can lead to a more pronounced degradation in performance. The addition of 90.9% (0.909) synthetic data typically results in lower performance, suggesting that a high proportion of synthetic data may overload the model with non-generalizable information, especially in the 'Traditional' and 'PreProcess' stages. Adding 45.4% synthetic data, in many cases, maintains model performance relatively stable, with no significant drops compared to the real data baseline. However, this quantity may introduce variations depending on the model and processing stage. Models that are presented to be robust, such as Bayesian Network and CTGAN, tend to handle this quantity well without increasing RMSE.

Compared to 22.7%, adding 45.4% synthetic data may lead to a slight degradation in performance in some models and datasets, but it is generally better than adding 68.2% or 90.9%.In many cases, 45.4% appears to be a balance point where the benefits of adding synthetic data can still be realized without overwhelming the models with information that could lead to inadequate generalization. The impact of 45.4% synthetic data is more controlled when the processing stage is 'Profile2Gen,' indicating that data from this stage helps mitigate any potential performance deterioration. This quantity

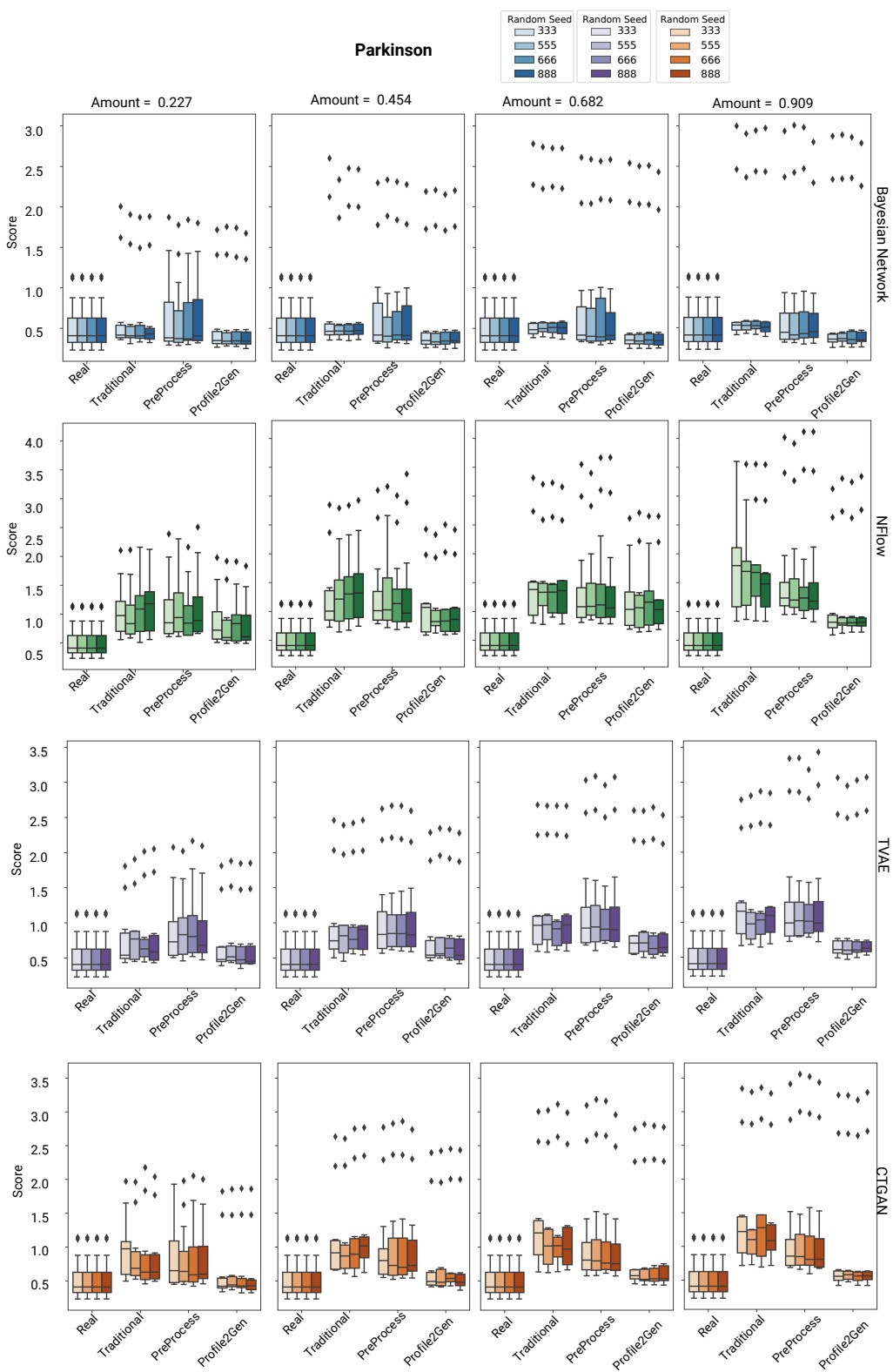

*Figure 24.* Augmentation performance across random seeds.

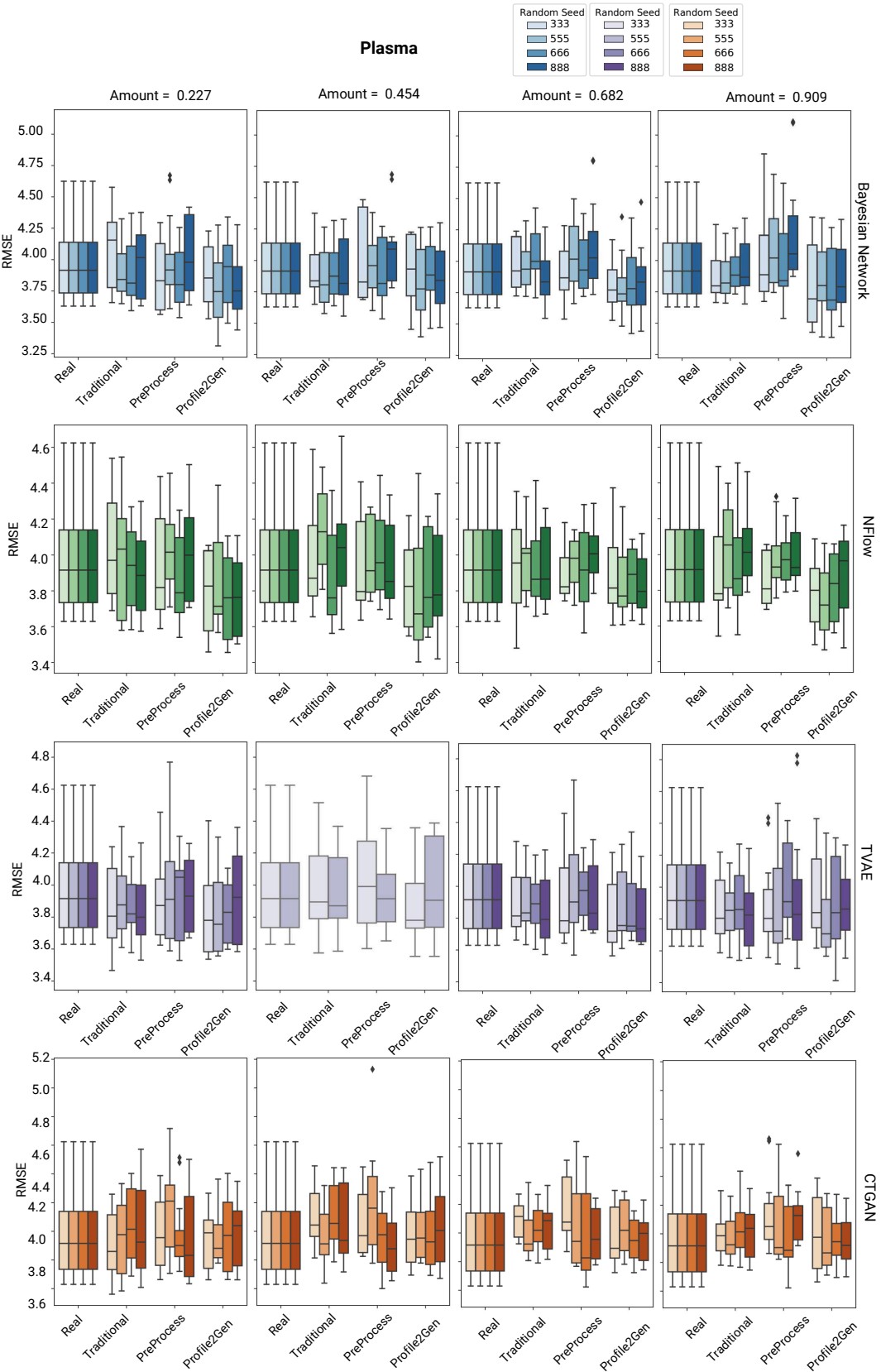

*Figure 25.* Augmentation performance across random seeds.

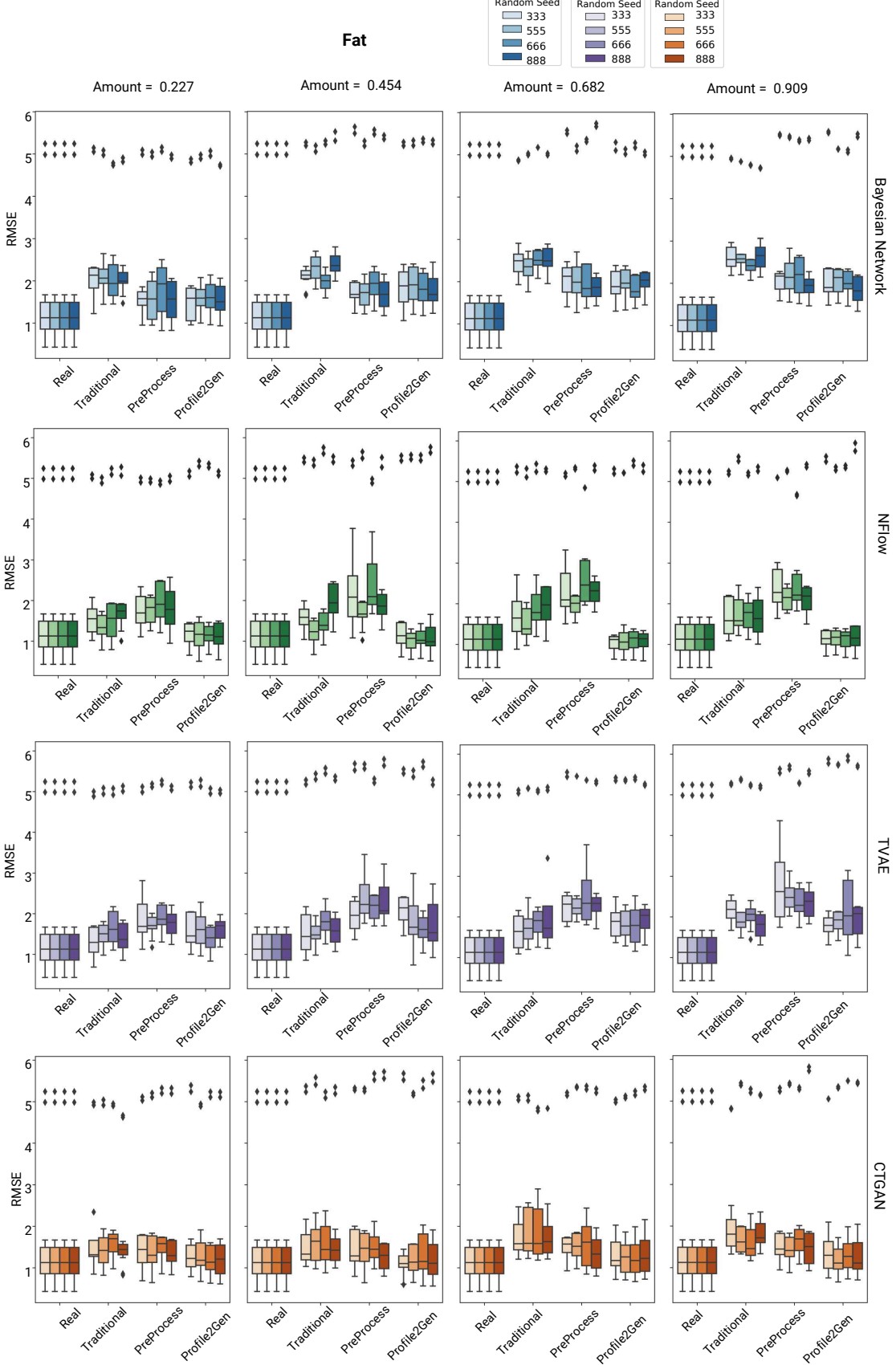

*Figure 26.* Augmentation performance across random seeds.

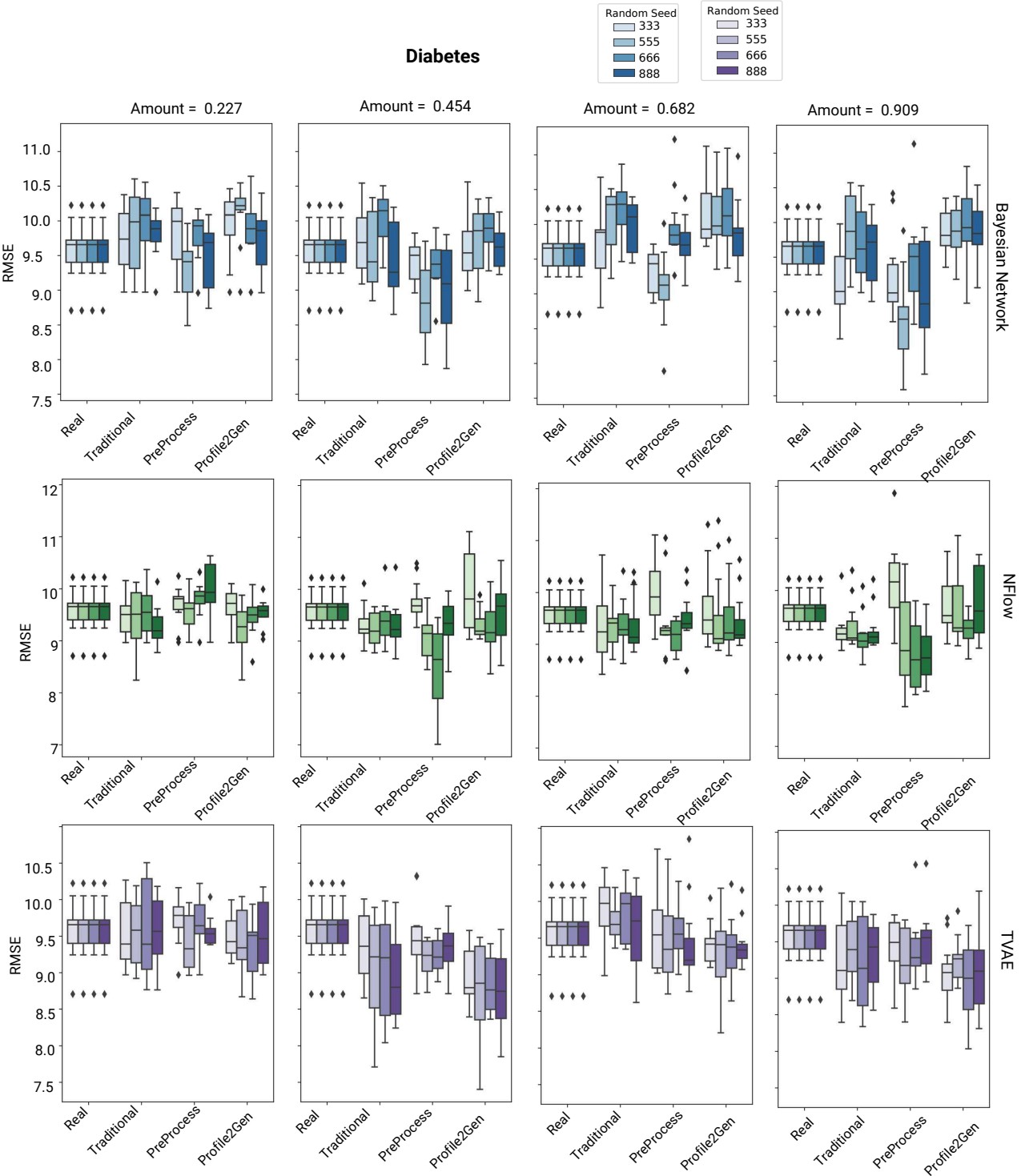

*Figure 27.* Augmentation performance across random seeds.

may show more variations depending on the model's sensitivity to the new information introduced in the' Traditional' and' PreProcess' stages. Figure 15 gives more insights.

## O.3. Evaluation Models

Figures 31, 30 , 29 and 28 demonstrate that models respond differently to synthetic data. For example, while LightGBM Large is often robust across various contexts, it exhibited moderate error and high variability in these experiments. This suggests that LightGBM Large may be less suited to certain types of synthetic data or may require additional tuning for optimal performance.

The NeuralNet Fast model, particularly in the Plasma dataset, reached the highest RMSE values for some generative models, indicating that neural networks can struggle with synthetic data under certain conditions. This could be due to issues like overfitting or a high sensitivity to the quality of the generated data.

Similarly, the K-Neighbors model showed higher error rates in the Fat dataset, highlighting the potential limitations of instance-based learning algorithms when dealing with synthetic datasets. These datasets might lack the granularity necessary for effective nearest-neighbor searches, leading to suboptimal performance.

# P. Statistical Significance

### P.1. TSTR

The Wilcoxon signed-rank tests revealed significant differences between the Real data and the synthetic data generated by all models across most datasets ($p < 0.05$). This was expected, indicating that the synthetic data, while attempting to mimic the real data, does not perfectly replicate its distribution. However, the comparisons between the intermediate and final versions of the synthetic data (Preprocess and Profile2Gen) provided valuable insights. For instance, in the plasma dataset with the TVAE model, all comparisons between synthetic, PreProcess, and Profile2Gen were non-significant ($p > 0.05$), suggesting that the model generated statistically similar data across these stages. This contrasts with the urinary dataset, where the Bayesian Network produced synthetic data that was significantly different from the real data in all comparisons. These findings highlight the varying performance of different generative models and the importance of evaluating the impact of refinement steps in the synthetic data generation process. Further analysis, including effect size calculations and focusing on the specific metric used for comparison, will provide a more complete understanding of these results.

**Dataset: urinary, Model: bayesian_network**  Wilcoxon Results: - Real vs Traditional: p-value = 0.0029 (Significant)

- Real vs PreProcess: p-value = 0.0029 (Significant)

- Real vs Profile2Gen: p-value = 0.0059 (Significant)

- Traditional vs PreProcess: p-value = 0.0029 (Significant)

- Traditional vs Profile2Gen: p-value = 0.0029 (Significant)

- PreProcess vs Profile2Gen: p-value = 0.0029 (Significant)

**Dataset: urinary, Model: NFlow**  Wilcoxon Results: - Real vs Traditional: p-value = 0.0029 (Significant)

- Real vs PreProcess: p-value = 0.0029 (Significant)

- Real vs Profile2Gen: p-value = 0.0029 (Significant)

- Traditional vs PreProcess: p-value = 0.2051 (Not Significant

- Traditional vs Profile2Gen: p-value = 0.0059 (Significant)

- PreProcess vs Profile2Gen: p-value . = 0.0410 (Significant)

**Dataset: urinary, Model: tvae**  Wilcoxon Results: - Real vs Traditional: p-value = 0.0029 (Significant)

- Real vs PreProcess: p-value = 0.0029 (Significant)

- Real vs Profile2Gen: p-value = 0.0029 (Significant)

- Traditional vs PreProcess: p-value = 0.0293 (Significant)

- Traditional vs Profile2Gen: p-value = 0.0967 (Not Significant)

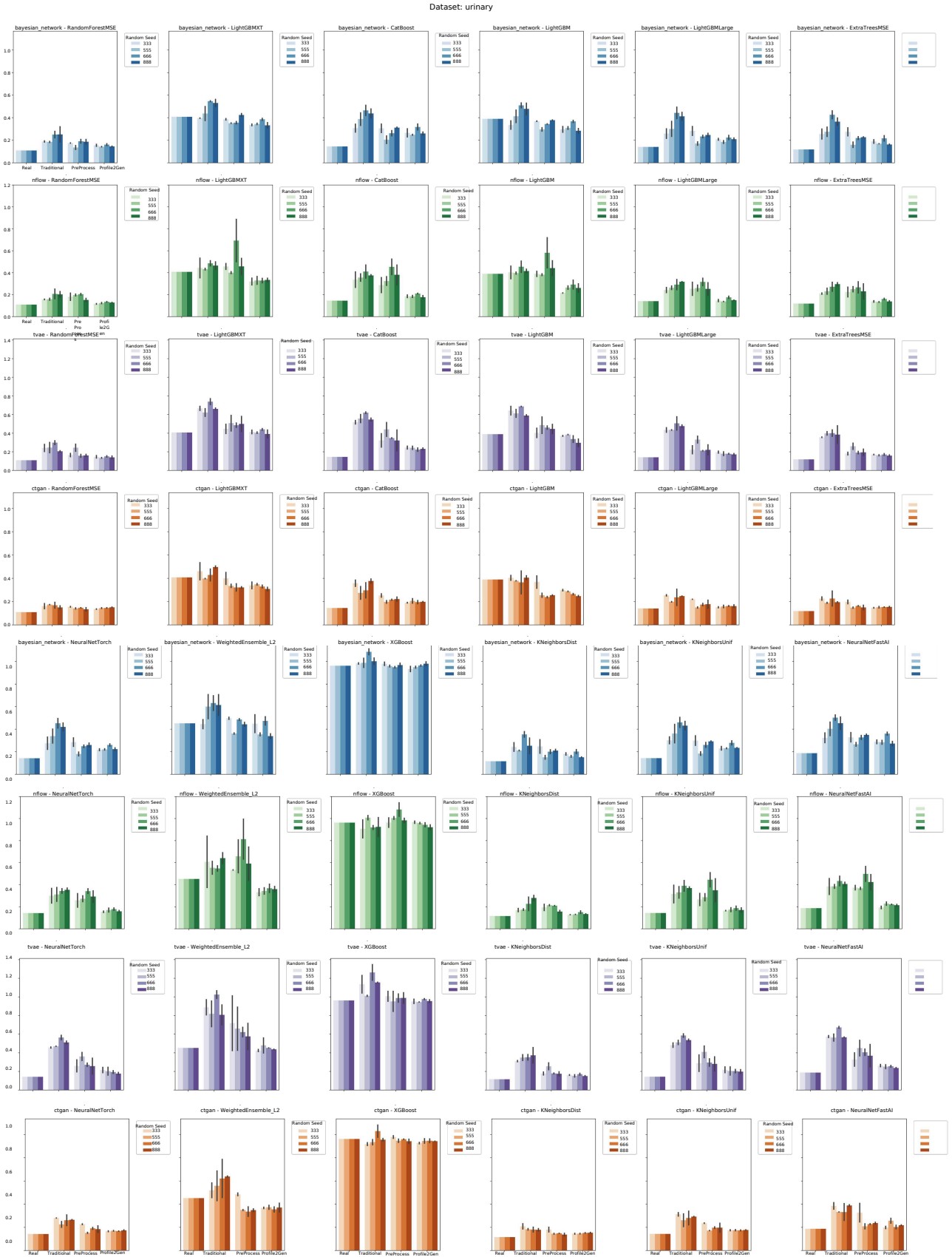

*Figure 28.* Performance of Augmentation Methods at the Evaluator Level.

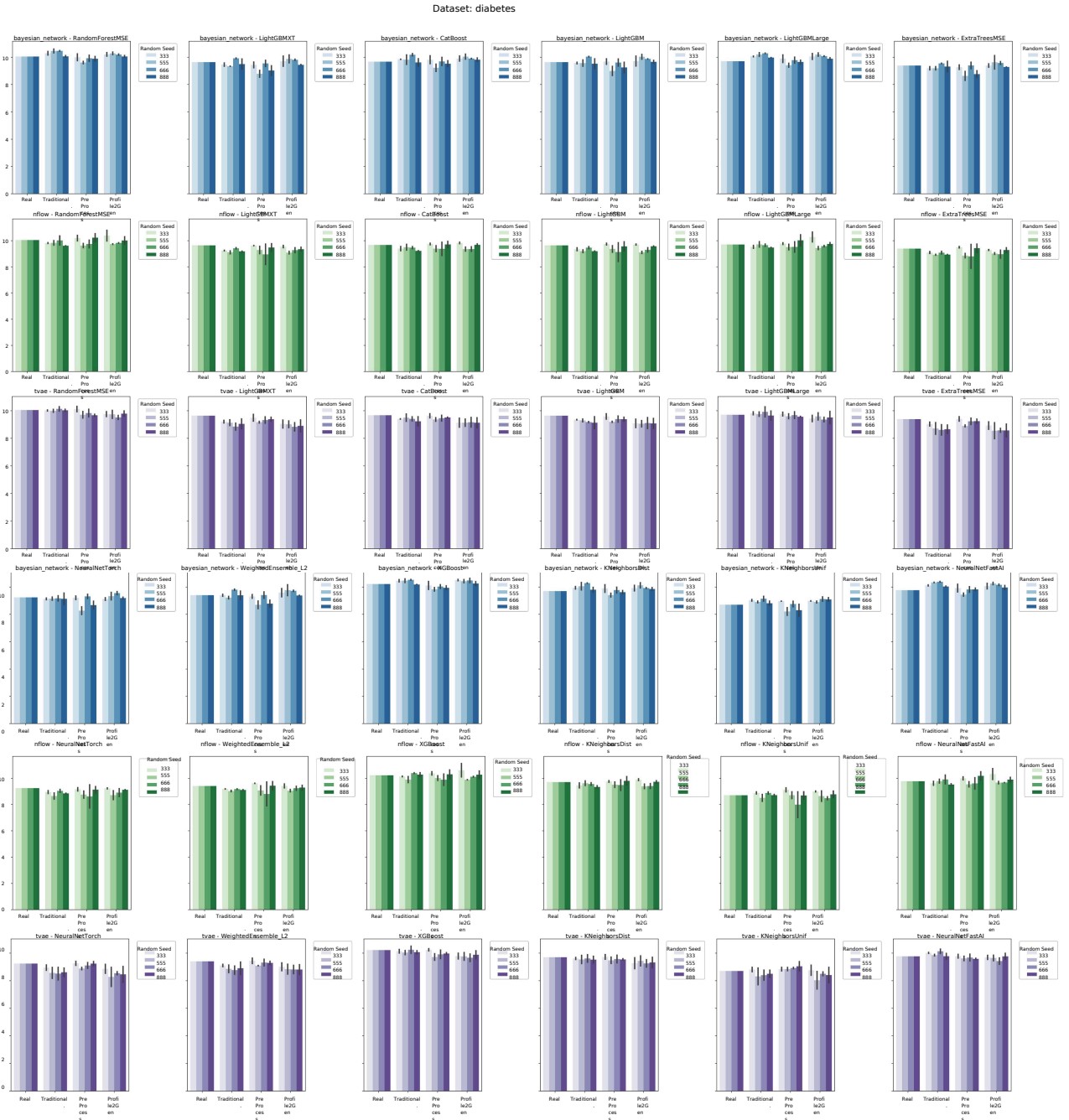

*Figure 29.* Performance of Augmentation Methods at the Evaluator Level.

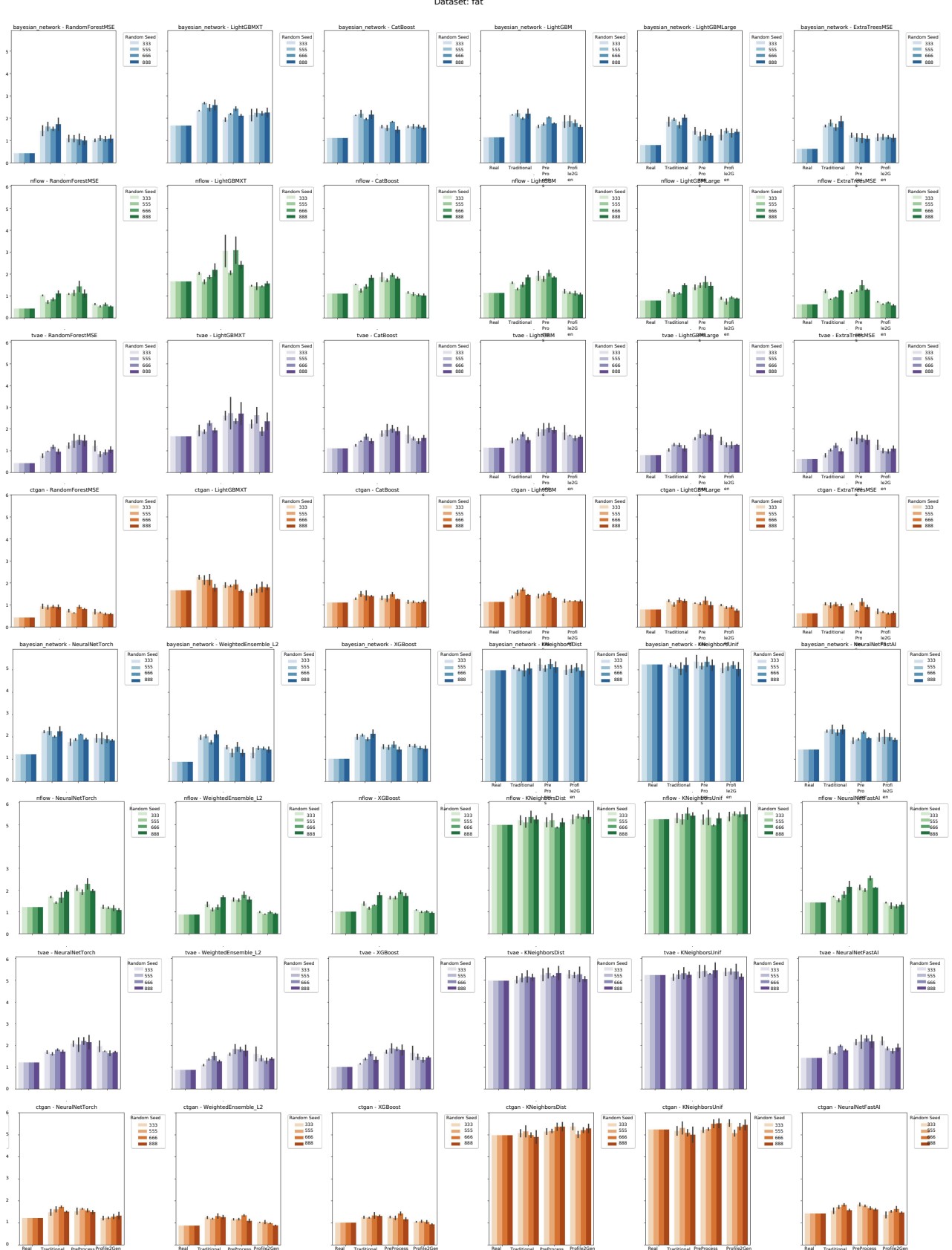

*Figure 30.* Performance of Augmentation Methods at the Evaluator Level.

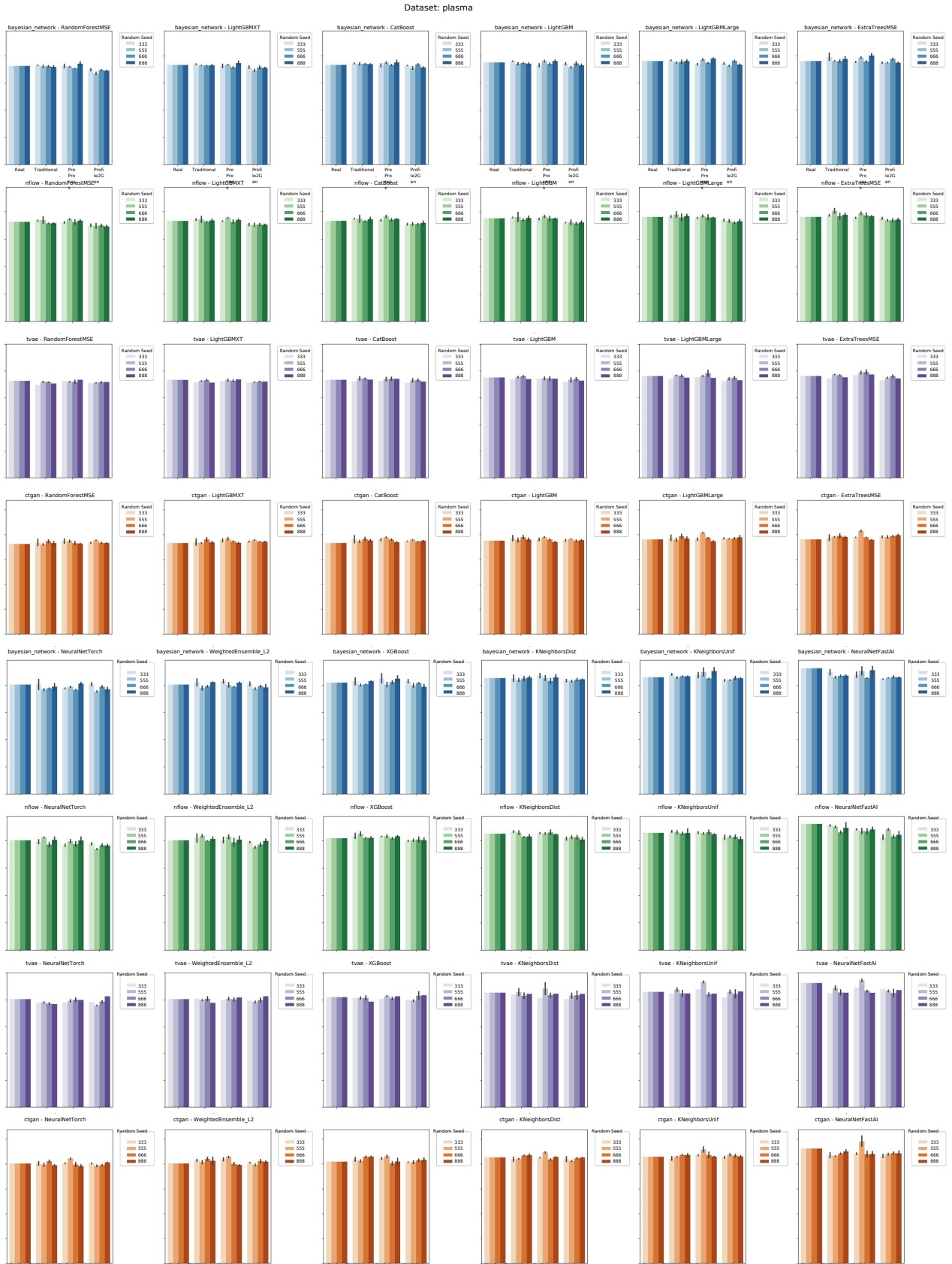

*Figure 31.* Performance of Augmentation Methods at the Evaluator Level.

- PreProcess vs Profile2Gen: p-value . = 0.0146 (Significant)

**Dataset: urinary, Model: ctgan**   Wilcoxon Results: - Real vs Traditional: p-value = 0.0029 (Significant)

- Real vs PreProcess: p-value = 0.0059 (Significant)

- Real vs Profile2Gen: p-value = 0.0029 (Significant)

- Traditional vs PreProcess: p-value = 0.0029 (Significant)

- Traditional vs Profile2Gen: p-value = 0.0410 (Significant)

- PreProcess vs Profile2Gen: p-value = 1.0000 (Not Significant)

**Dataset: Parkinson, Model: bayesian_network**   Wilcoxon Results: - Real vs PreProcess: p-value = 0.0015 (Significant)

- Real vs Profile2Gen: p-value = 0.0015 (Significant)

- PreProcess vs Profile2Gen: p-value = 0.0015 (Significant)

**Dataset: Parkinson, Model: NFlow**   Wilcoxon Results: - Real vs Traditional: p-value = 0.0029 (Significant)

- Real vs PreProcess: p-value = 0.0029 (Significant)

- Real vs Profile2Gen: p-value = 0.0029 (Significant)

- Traditional vs PreProcess: p-value = 0.2051 (Not Significant

- Traditional vs Profile2Gen: p-value = 0.0029 (Significant)

- PreProcess vs Profile2Gen: p-value = 0.0029 (Significant)

**Dataset: Parkinson, Model: tvae**   Wilcoxon Results: - Real vs Traditional: p-value = 0.0029 (Significant)

- Real vs PreProcess: p-value = 0.0029 (Significant)

- Real vs Profile2Gen: p-value = 0.0029 (Significant)

- Traditional vs PreProcess: p-value = 0.0029 (Significant)

- Traditional vs Profile2Gen: p-value = 0.0029 (Significant)

- PreProcess vs Profile2Gen: p-value = 0.6592 (Not Significant)

**Dataset: Parkinson, Model: ctgan**   Wilcoxon Results: - Real vs Traditional: p-value = 0.0029 (Significant)

- Real vs PreProcess: p-value = 0.0029 (Significant)

- Real vs Profile2Gen: p-value = 0.0029 (Significant)

- Traditional vs PreProcess: p-value = 0.0029 (Significant)

- Traditional vs Profile2Gen: p-value = 0.0029 (Significant)

- PreProcess vs Profile2Gen: p-value = 0.2549 (Not Significant)

**Dataset: cholesterol, Model: bayesian_network**   Wilcoxon Results: - Real vs Traditional: p-value = 0.0029 (Significant)

- Real vs PreProcess: p-value = 0.0029 (Significant)

- Real vs Profile2Gen: p-value = 1.0000 (Not Significant)

- Traditional vs PreProcess: p-value = 1.0000 (Not Significant

- Traditional vs Profile2Gen: p-value = 0.0029 (Significant)

- PreProcess vs Profile2Gen: p-value = 0.0029 (Significant)

**Dataset: cholesterol, Model: NFlow**     Wilcoxon Results: - Real vs Traditional: p-value = 0.0029 (Significant)

- Real vs PreProcess: p-value = 0.0059 (Significant)

- Real vs Profile2Gen: p-value = 0.0029 (Significant)

- Traditional vs PreProcess: p-value = 0.0029 (Significant)

- Traditional vs Profile2Gen: p-value = 0.6592 (Not Significant)

- PreProcess vs Profile2Gen: p-value = 0.0205 (Significant)

**Dataset: cholesterol, Model: tvae**     Wilcoxon Results: - Real vs Traditional: p-value = 0.0029 (Significant)

- Real vs PreProcess: p-value = 0.0029 (Significant)

- Real vs Profile2Gen: p-value = 0.0029 (Significant)

- Traditional vs PreProcess: p-value = 0.0029 (Significant)

- Traditional vs Profile2Gen: p-value = 0.4629 (Not Significant)

- PreProcess vs Profile2Gen: p-value = 0.0029 (Significant)

**Dataset: cholesterol, Model: ctgan**     Wilcoxon Results: - Real vs Traditional: p-value = 0.0029 (Significant)

- Real vs PreProcess: p-value = 0.0029 (Significant)

- Real vs Profile2Gen: p-value = 0.0029 (Significant)

- Traditional vs PreProcess: p-value = 0.3135 (Not Significant)

- Traditional vs Profile2Gen: p-value = 0.0410 (Significant)

- PreProcess vs Profile2Gen: p-value = 0.4629 (Not Significant)

**Dataset: diabetes, Model: bayesian_network**     Wilcoxon Results: - Real vs Traditional: p-value = 0.0557 (Not Significant

- Real vs PreProcess: p-value = 0.0088 (Significant)

- Real vs Profile2Gen: p-value = 0.0029 (Significant)

- Traditional vs PreProcess: p-value = 0.0029 (Significant)

- Traditional vs Profile2Gen: p-value = 1.0000 (Not Significant)

- PreProcess vs Profile2Gen: p-value = 0.0029 (Significant)

**Dataset: diabetes, Model: NFlow**     Wilcoxon Results: - Real vs Traditional: p-value = 0.0029 (Significant)

- Real vs PreProcess: p-value = 0.2051 (Not Significant

- Real vs Profile2Gen: p-value = 0.5537 (Not Significant)

- Traditional vs PreProcess: p-value = 0.0029 (Significant)

- Traditional vs Profile2Gen: p-value = 0.0029 (Significant)

- PreProcess vs Profile2Gen: p-value = 1.0000 (Not Significant)

**Dataset: diabetes, Model: tvae**     Wilcoxon Results: - Real vs Traditional: p-value = 0.0205 (Significant)

- Real vs PreProcess: p-value = 0.0029 (Significant)

- Real vs Profile2Gen: p-value = 0.0029 (Significant)

- Traditional vs PreProcess: p-value = 0.0029 (Significant)

- Traditional vs Profile2Gen: p-value = 0.0029 (Significant)

- PreProcess vs Profile2Gen: p-value = 0.0059 (Significant)

**Dataset: fat, Model: bayesian_network**   Wilcoxon Results: - Real vs Traditional: p-value = 0.0029 (Significant)

- Real vs PreProcess: p-value = 0.0029 (Significant)

- Real vs Profile2Gen: p-value = 0.0029 (Significant)

- Traditional vs PreProcess: p-value = 0.0029 (Significant)

- Traditional vs Profile2Gen: p-value = 0.0029 (Significant)

- PreProcess vs Profile2Gen: p-value = 1.0000 (Not Significant)

**Dataset: fat, Model: NFlow**   Wilcoxon Results: - Real vs Traditional: p-value = 0.0029 (Significant)

- Real vs PreProcess: p-value = 0.0029 (Significant)

- Real vs Profile2Gen: p-value = 0.0029 (Significant)

- Traditional vs PreProcess: p-value = 0.0410 (Significant)

- Traditional vs Profile2Gen: p-value = 0.0029 (Significant)

- PreProcess vs Profile2Gen: p-value = 0.0029 (Significant)

**Dataset: fat, Model: tvae**   Wilcoxon Results: - Real vs Traditional: p-value = 0.0029 (Significant)

- Real vs PreProcess: p-value = 0.0029 (Significant)

- Real vs Profile2Gen: p-value = 0.0029 (Significant)

- Traditional vs PreProcess: p-value = 0.0146 (Significant)

- Traditional vs Profile2Gen: p-value = 0.0029 (Significant)

- PreProcess vs Profile2Gen: p-value = 0.0146 (Significant)

**Dataset: fat, Model: ctgan**   Wilcoxon Results: - Real vs Traditional: p-value = 0.0029 (Significant)

- Real vs PreProcess: p-value = 0.0029 (Significant)

- Real vs Profile2Gen: p-value = 0.0029 (Significant)

- Traditional vs PreProcess: p-value = 0.0293 (Significant)

- Traditional vs Profile2Gen: p-value = 0.6592 (Not Significant)

- PreProcess vs Profile2Gen: p-value = 0.0029 (Significant)

**Dataset: plasma, Model: NFlow**   Wilcoxon Results: - Real vs Traditional: p-value = 0.0029 (Significant)

- Real vs PreProcess: p-value = 0.0059 (Significant)

- Real vs Profile2Gen: p-value = 0.0029 (Significant)

- Traditional vs PreProcess: p-value = 0.0029 (Significant)

- Traditional vs Profile2Gen: p-value = 1.0000 (Not Significant)

- PreProcess vs Profile2Gen: p-value = 0.0029 (Significant)

**Dataset: plasma, Model: tvae**   Wilcoxon Results: - Real vs Traditional: p-value = 0.0029 (Significant)

- Real vs PreProcess: p-value = 0.0029 (Significant)

- Real vs Profile2Gen: p-value = 0.0029 (Significant)

- Traditional vs PreProcess: p-value = 1.0000 (Not Significant

- Traditional vs Profile2Gen: p-value = 1.0000 (Not Significant)

- PreProcess vs Profile2Gen: p-value = 1.0000 (Not Significant)

**Dataset: plasma, Model: ctgan** Wilcoxon Results:

- Real vs Traditional: p-value = 0.0029 (Significant)

- Real vs PreProcess: p-value = 0.0029 (Significant)

- Real vs Profile2Gen: p-value = 0.0410 (Significant)

- Traditional vs PreProcess: p-value = 0.2549 (Not Significant

- Traditional vs Profile2Gen: p-value = 0.0029 (Significant)

- PreProcess vs Profile2Gen: p-value = 1.0000 (Not Significant)

**P.2. Augmentation**

These results present post hoc tests (Wilcoxon tests ) following Kruskal-Wallis tests, comparing 'Real' data to 'Traditional', 'PreProcess', and 'Profile2Gen' synthetic data generation methods across different datasets, models, and proportions. Our analysis of augmentation tasks revealed that the 'Profile2Gen' method effectively generated augmented data that closely resembled the 'Real' data, particularly at higher proportions. For instance, in the urinary dataset with the Bayesian Network, the 'Real vs. Profile2Gen' comparison was non-significant at a proportion of 0.682, suggesting statistical similarity. This indicates that 'Profile2Gen' augments data in a way that preserves the real data's underlying structure better than 'Traditional' or 'PreProcess' methods.

**Dataset: Urinary, Model: Bayesian_network, Proportion: 0.227** Post hoc Results: - Real vs Traditional:p-value = 0.0000 (Significant) - Real vs PreProcess:p-value = 0.0000 (Significant) - Real vs Profile2Gen:p-value = 0.0379 (Significant) - Traditional vs PreProcess:p-value = 0.0000 (Significant) - Traditional vs Profile2Gen:p-value = 0.0000 (Significant) - PreProcess vs Profile2Gen:p-value = 0.3006 (Not Significant)

**Dataset: Urinary, Model: Bayesian_network, Proportion: 0.454** Post hoc Results: - Real vs Traditional:p-value = 0.0000 (Significant) - Real vs PreProcess:p-value = 0.0000 (Significant) - Real vs Profile2Gen:p-value = 0.0010 (Significant) - Traditional vs PreProcess:p-value = 0.0000 (Significant) - Traditional vs Profile2Gen:p-value = 0.0000 (Significant) - PreProcess vs Profile2Gen:p-value = 0.0007 (Significant)

**Dataset: Urinary, Model: Bayesian_network, Proportion: 0.682** Post hoc Results: - Real vs Traditional:p-value = 0.0000 (Significant) - Real vs PreProcess:p-value = 0.0000 (Significant) - Real vs Profile2Gen:p-value = 1.0000 (Not Significant) - Traditional vs PreProcess:p-value = 0.0000 (Significant) - Traditional vs Profile2Gen:p-value = 0.0000 (Significant) - PreProcess vs Profile2Gen:p-value = 0.0000 (Significant)

**Dataset: Urinary, Model: Bayesian_network, Proportion: 0.909** Post hoc Results: - Real vs Traditional:p-value = 0.0000 (Significant) - Real vs PreProcess:p-value = 0.0000 (Significant) - Real vs Profile2Gen:p-value = 0.0004 (Significant) - Traditional vs PreProcess:p-value = 0.0000 (Significant) - Traditional vs Profile2Gen:p-value = 0.0000 (Significant) - PreProcess vs Profile2Gen:p-value = 0.0000 (Significant)

**Dataset: Urinary, Model: NFlow, Proportion: 0.227** Post hoc Results: - Real vs Traditional:p-value = 0.0000 (Significant) - Real vs PreProcess:p-value = 0.0000 (Significant) - Real vs Profile2Gen:p-value = 1.0000 (Not Significant) - Traditional vs PreProcess:p-value = 0.1255 (Not Significant) - Traditional vs Profile2Gen:p-value = 0.0000 (Significant) - PreProcess vs Profile2Gen:p-value = 0.0000 (Significant)

**Dataset: Urinary, Model: NFlow, Proportion: 0.454** Post hoc Results: - Real vs Traditional:p-value = 0.0000 (Significant) - Real vs PreProcess:p-value = 0.0000 (Significant) - Real vs Profile2Gen:p-value = 1.0000 (Not Significant) -

Traditional vs PreProcess:p-value = 0.3649 (Not Significant) - Traditional vs Profile2Gen:p-value = 0.0000 (Significant) - PreProcess vs Profile2Gen:p-value = 0.0000 (Significant)

**Dataset: Urinary, Model: NFlow, Proportion: 0.682** Post hoc Results: - Real vs Traditional:p-value = 0.0000 (Significant) - Real vs PreProcess:p-value = 0.0000 (Significant) - Real vs Profile2Gen:p-value = 1.0000 (Not Significant) - Traditional vs PreProcess:p-value = 0.0243 (Significant) - Traditional vs Profile2Gen:p-value = 0.0000 (Significant) - PreProcess vs Profile2Gen:p-value = 0.0000 (Significant)

**Dataset: Urinary, Model: NFlow, Proportion: 0.909** Post hoc Results: - Real vs Traditional:p-value = 0.0000 (Significant) - Real vs PreProcess:p-value = 0.0000 (Significant) - Real vs Profile2Gen:p-value = 1.0000 (Not Significant) - Traditional vs PreProcess:p-value = 1.0000 (Not Significant) - Traditional vs Profile2Gen:p-value = 0.0000 (Significant) - PreProcess vs Profile2Gen:p-value = 0.0000 (Significant)

**Dataset: Urinary, Model: Tvae, Proportion: 0.227** Post hoc Results: - Real vs Traditional:p-value = 0.0000 (Significant) - Real vs PreProcess:p-value = 0.0000 (Significant) - Real vs Profile2Gen:p-value = 0.0000 (Significant) - Traditional vs PreProcess:p-value = 0.0000 (Significant) - Traditional vs Profile2Gen:p-value = 0.0000 (Significant) - PreProcess vs Profile2Gen:p-value = 0.0001 (Significant)

**Dataset: Urinary, Model: Tvae, Proportion: 0.454** Post hoc Results: - Real vs Traditional:p-value = 0.0000 (Significant) - Real vs PreProcess:p-value = 0.0000 (Significant) - Real vs Profile2Gen:p-value = 0.0000 (Significant) - Traditional vs PreProcess:p-value = 0.0000 (Significant) - Traditional vs Profile2Gen:p-value = 0.0000 (Significant) - PreProcess vs Profile2Gen:p-value = 0.0000 (Significant)

**Dataset: Urinary, Model: Tvae, Proportion: 0.682** Post hoc Results: - Real vs Traditional:p-value = 0.0000 (Significant) - Real vs PreProcess:p-value = 0.0000 (Significant) - Real vs Profile2Gen:p-value = 1.0000 (Not Significant) - Traditional vs PreProcess:p-value = 0.0000 (Significant) - Traditional vs Profile2Gen:p-value = 0.0000 (Significant) - PreProcess vs Profile2Gen:p-value = 0.0000 (Significant)

**Dataset: Urinary, Model: Tvae, Proportion: 0.909** Post hoc Results: - Real vs Traditional:p-value = 0.0000 (Significant) - Real vs PreProcess:p-value = 0.0000 (Significant) - Real vs Profile2Gen:p-value = 0.1797 (Not Significant) - Traditional vs PreProcess:p-value = 0.0000 (Significant) - Traditional vs Profile2Gen:p-value = 0.0000 (Significant) - PreProcess vs Profile2Gen:p-value = 0.0000 (Significant)

**Dataset: Urinary, Model: Ctgan, Proportion: 0.227** Post hoc Results: - Real vs Traditional:p-value = 0.0000 (Significant) - Real vs PreProcess:p-value = 0.1797 (Not Significant) - Real vs Profile2Gen:p-value = 1.0000 (Not Significant) - Traditional vs PreProcess:p-value = 0.0000 (Significant) - Traditional vs Profile2Gen:p-value = 0.0000 (Significant) - PreProcess vs Profile2Gen:p-value = 0.1485 (Not Significant)

**Dataset: Urinary, Model: Ctgan, Proportion: 0.454** Post hoc Results: - Real vs Traditional:p-value = 0.0000 (Significant) - Real vs PreProcess:p-value = 1.0000 (Not Significant) - Real vs Profile2Gen:p-value = 1.0000 (Not Significant) - Traditional vs PreProcess:p-value = 0.0000 (Significant) - Traditional vs Profile2Gen:p-value = 0.0000 (Significant) - PreProcess vs Profile2Gen:p-value = 0.1947 (Not Significant)

**Dataset: Urinary, Model: Ctgan, Proportion: 0.682** Post hoc Results: - Real vs Traditional:p-value = 0.0000 (Significant) - Real vs PreProcess:p-value = 0.2524 (Not Significant) - Real vs Profile2Gen:p-value = 1.0000 (Not Significant) - Traditional vs PreProcess:p-value = 0.0000 (Significant) - Traditional vs Profile2Gen:p-value = 0.0000 (Significant) - PreProcess vs Profile2Gen:p-value = 0.0000 (Significant)

**Dataset: Urinary, Model: Ctgan, Proportion: 0.909** Post hoc Results: - Real vs Traditional:p-value = 0.0000 (Significant) - Real vs PreProcess:p-value = 0.4604 (Not Significant) - Real vs Profile2Gen:p-value = 1.0000 (Not Significant) - Traditional vs PreProcess:p-value = 0.0000 (Significant) - Traditional vs Profile2Gen:p-value = 0.0000 (Significant) - PreProcess vs Profile2Gen:p-value = 0.0000 (Significant)

**Dataset: Parkinson, Model: Bayesian_network, Proportion: 0.227** Post hoc Results: - Real vs Traditional:p-value = 1.0000 (Not Significant) - Real vs PreProcess:p-value = 0.0001 (Significant) - Real vs Profile2Gen:p-value = 1.0000 (Not Significant) - Traditional vs PreProcess:p-value = 0.9776 (Not Significant) - Traditional vs Profile2Gen:p-value = 0.0000 (Significant) - PreProcess vs Profile2Gen:p-value = 0.1532 (Not Significant)

**Dataset: Parkinson, Model: Bayesian_network, Proportion: 0.454** Post hoc Results: - Real vs Traditional:p-value = 0.1254 (Not Significant) - Real vs PreProcess:p-value = 0.0000 (Significant) - Real vs Profile2Gen:p-value = 1.0000 (Not Significant) - Traditional vs PreProcess:p-value = 0.2528 (Not Significant) - Traditional vs Profile2Gen:p-value = 0.0000 (Significant) - PreProcess vs Profile2Gen:p-value = 0.0468 (Significant)

**Dataset: Parkinson, Model: Bayesian_network, Proportion: 0.682** Post hoc Results: - Real vs Traditional:p-value = 0.0239 (Significant) - Real vs PreProcess:p-value = 0.0000 (Significant) - Real vs Profile2Gen:p-value = 1.0000 (Not Significant) - Traditional vs PreProcess:p-value = 0.0570 (Not Significant) - Traditional vs Profile2Gen:p-value = 0.0000 (Significant) - PreProcess vs Profile2Gen:p-value = 0.0445 (Significant)

**Dataset: Parkinson, Model: Bayesian_network, Proportion: 0.909** Post hoc Results: - Real vs Traditional:p-value = 0.0043 (Significant) - Real vs PreProcess:p-value = 0.0000 (Significant) - Real vs Profile2Gen:p-value = 1.0000 (Not Significant) - Traditional vs PreProcess:p-value = 0.0691 (Not Significant) - Traditional vs Profile2Gen:p-value = 0.0000 (Significant) - PreProcess vs Profile2Gen:p-value = 0.0354 (Significant)

**Dataset: Parkinson, Model: NFlow, Proportion: 0.227** Post hoc Results: - Real vs Traditional:p-value = 0.0000 (Significant) - Real vs PreProcess:p-value = 0.0000 (Significant) - Real vs Profile2Gen:p-value = 0.0000 (Significant) - Traditional vs PreProcess:p-value = 1.0000 (Not Significant) - Traditional vs Profile2Gen:p-value = 0.0000 (Significant) - PreProcess vs Profile2Gen:p-value = 0.0193 (Significant)

**Dataset: Parkinson, Model: NFlow, Proportion: 0.454** Post hoc Results: - Real vs Traditional:p-value = 0.0000 (Significant) - Real vs PreProcess:p-value = 0.0000 (Significant) - Real vs Profile2Gen:p-value = 0.0000 (Significant) - Traditional vs PreProcess:p-value = 1.0000 (Not Significant) - Traditional vs Profile2Gen:p-value = 0.0000 (Significant) - PreProcess vs Profile2Gen:p-value = 0.0760 (Not Significant)

**Dataset: Parkinson, Model: NFlow, Proportion: 0.682** Post hoc Results: - Real vs Traditional:p-value = 0.0000 (Significant) - Real vs PreProcess:p-value = 0.0000 (Significant) - Real vs Profile2Gen:p-value = 0.0000 (Significant) - Traditional vs PreProcess:p-value = 0.8619 (Not Significant) - Traditional vs Profile2Gen:p-value = 0.0000 (Significant) - PreProcess vs Profile2Gen:p-value = 0.8758 (Not Significant)

**Dataset: Parkinson, Model: NFlow, Proportion: 0.909** Post hoc Results: - Real vs Traditional:p-value = 0.0000 (Significant) - Real vs PreProcess:p-value = 0.0000 (Significant) - Real vs Profile2Gen:p-value = 0.0000 (Significant) - Traditional vs PreProcess:p-value = 1.0000 (Not Significant) - Traditional vs Profile2Gen:p-value = 0.0000 (Significant) - PreProcess vs Profile2Gen:p-value = 0.0000 (Significant)

**Dataset: Parkinson, Model: Tvae, Proportion: 0.227** Post hoc Results: - Real vs Traditional:p-value = 0.0004 (Significant) - Real vs PreProcess:p-value = 0.0000 (Significant) - Real vs Profile2Gen:p-value = 0.0192 (Significant) - Traditional vs PreProcess:p-value = 0.3665 (Not Significant) - Traditional vs Profile2Gen:p-value = 0.0000 (Significant) - PreProcess vs Profile2Gen:p-value = 0.0006 (Significant)

**Dataset: Parkinson, Model: Tvae, Proportion: 0.454** Post hoc Results: - Real vs Traditional:p-value = 0.0000 (Significant) - Real vs PreProcess:p-value = 0.0000 (Significant) - Real vs Profile2Gen:p-value = 0.0005 (Significant) - Traditional vs PreProcess:p-value = 1.0000 (Not Significant) - Traditional vs Profile2Gen:p-value = 0.0000 (Significant) - PreProcess vs Profile2Gen:p-value = 0.0005 (Significant)

**Dataset: Parkinson, Model: Tvae, Proportion: 0.682** Post hoc Results: - Real vs Traditional:p-value = 0.0000 (Significant) - Real vs PreProcess:p-value = 0.0000 (Significant) - Real vs Profile2Gen:p-value = 0.0000 (Significant) - Traditional vs PreProcess:p-value = 1.0000 (Not Significant) - Traditional vs Profile2Gen:p-value = 0.0000 (Significant) - PreProcess vs Profile2Gen:p-value = 0.0010 (Significant)

**Dataset: Parkinson, Model: Tvae, Proportion: 0.909** Post hoc Results: - Real vs Traditional:p-value = 0.0000 (Significant) - Real vs PreProcess:p-value = 0.0000 (Significant) - Real vs Profile2Gen:p-value = 0.0001 (Significant) - Traditional vs PreProcess:p-value = 1.0000 (Not Significant) - Traditional vs Profile2Gen:p-value = 0.0000 (Significant) - PreProcess vs Profile2Gen:p-value = 0.0000 (Significant)

**Dataset: Parkinson, Model: Ctgan, Proportion: 0.227** Post hoc Results: - Real vs Traditional:p-value = 0.0000 (Significant) - Real vs PreProcess:p-value = 0.0000 (Significant) - Real vs Profile2Gen:p-value = 0.6071 (Not Significant) - Traditional vs PreProcess:p-value = 1.0000 (Not Significant) - Traditional vs Profile2Gen:p-value = 0.0000 (Significant) - PreProcess vs Profile2Gen:p-value = 0.0002 (Significant)

**Dataset: Parkinson, Model: Ctgan, Proportion: 0.454** Post hoc Results: - Real vs Traditional:p-value = 0.0000 (Significant) - Real vs PreProcess:p-value = 0.0000 (Significant) - Real vs Profile2Gen:p-value = 0.0124 (Significant) - Traditional vs PreProcess:p-value = 0.4101 (Not Significant) - Traditional vs Profile2Gen:p-value = 0.0000 (Significant) - PreProcess vs Profile2Gen:p-value = 0.0001 (Significant)

**Dataset: Parkinson, Model: Ctgan, Proportion: 0.682** Post hoc Results: - Real vs Traditional:p-value = 0.0000 (Significant) - Real vs PreProcess:p-value = 0.0000 (Significant) - Real vs Profile2Gen:p-value = 0.0009 (Significant) - Traditional vs PreProcess:p-value = 0.1150 (Not Significant) - Traditional vs Profile2Gen:p-value = 0.0000 (Significant) - PreProcess vs Profile2Gen:p-value = 0.0001 (Significant)

**Dataset: Parkinson, Model: Ctgan, Proportion: 0.909** Post hoc Results: - Real vs Traditional:p-value = 0.0000 (Significant) - Real vs PreProcess:p-value = 0.0000 (Significant) - Real vs Profile2Gen:p-value = 0.0007 (Significant) - Traditional vs PreProcess:p-value = 0.0383 (Significant) - Traditional vs Profile2Gen:p-value = 0.0000 (Significant) - PreProcess vs Profile2Gen:p-value = 0.0000 (Significant)

**Dataset: Cholesterol, Model: Bayesian_network, Proportion: 0.909** Post hoc Results: - Real vs Traditional:p-value = 0.0036 (Significant)

**Dataset: Cholesterol, Model: NFlow, Proportion: 0.909** Post hoc Results: - Real vs Traditional:p-value = 0.0004 (Significant)

**Dataset: Cholesterol, Model: Tvae, Proportion: 0.909** Post hoc Results: - Real vs Traditional:p-value = 0.0009 (Significant)

**Dataset: Cholesterol, Model: Ctgan, Proportion: 0.682** Post hoc Results: - Real vs Traditional:p-value = 0.0000 (Significant)

**Dataset: Cholesterol, Model: Ctgan, Proportion: 0.909** Post hoc Results: - Real vs Traditional:p-value = 0.0000 (Significant)

**Dataset: Diabetes, Model: Bayesian_network, Proportion: 0.227** Post hoc Results: - Real vs Traditional:p-value = 0.0000 (Significant) - Real vs PreProcess:p-value = 1.0000 (Not Significant) - Real vs Profile2Gen:p-value = 0.0000 (Significant) - Traditional vs PreProcess:p-value = 0.0002 (Significant) - Traditional vs Profile2Gen:p-value = 1.0000 (Not Significant) - PreProcess vs Profile2Gen:p-value = 0.0000 (Significant)

**Dataset: Diabetes, Model: Bayesian_network, Proportion: 0.454** Post hoc Results: - Real vs Traditional:p-value = 0.4817 (Not Significant) - Real vs PreProcess:p-value = 0.0000 (Significant) - Real vs Profile2Gen:p-value = 0.0343 (Significant) - Traditional vs PreProcess:p-value = 0.0000 (Significant) - Traditional vs Profile2Gen:p-value = 1.0000 (Not Significant) - PreProcess vs Profile2Gen:p-value = 0.0000 (Significant)

**Dataset: Diabetes, Model: Bayesian_network, Proportion: 0.682** Post hoc Results: - Real vs Traditional:p-value = 0.0000 (Significant) - Real vs PreProcess:p-value = 0.8360 (Not Significant) - Real vs Profile2Gen:p-value = 0.0000 (Significant) - Traditional vs PreProcess:p-value = 0.0000 (Significant) - Traditional vs Profile2Gen:p-value = 1.0000 (Not Significant) - PreProcess vs Profile2Gen:p-value = 0.0000 (Significant)

**Dataset: Diabetes, Model: Bayesian_network, Proportion: 0.909**   Post hoc Results: - Real vs Traditional:p-value = 1.0000 (Not Significant) - Real vs PreProcess:p-value = 0.0000 (Significant) - Real vs Profile2Gen:p-value = 0.0000 (Significant) - Traditional vs PreProcess:p-value = 0.0000 (Significant) - Traditional vs Profile2Gen:p-value = 0.0000 (Significant) - PreProcess vs Profile2Gen:p-value = 0.0000 (Significant)

**Dataset: Diabetes, Model: NFlow, Proportion: 0.227**   Post hoc Results: - Real vs Traditional:p-value = 0.0007 (Significant) - Real vs PreProcess:p-value = 0.0001 (Significant) - Real vs Profile2Gen:p-value = 0.0006 (Significant) - Traditional vs PreProcess:p-value = 0.0000 (Significant) - Traditional vs Profile2Gen:p-value = 1.0000 (Not Significant) - PreProcess vs Profile2Gen:p-value = 0.0000 (Significant)

**Dataset: Diabetes, Model: NFlow, Proportion: 0.454**   Post hoc Results: - Real vs Traditional:p-value = 0.0000 (Significant) - Real vs PreProcess:p-value = 0.0000 (Significant) - Real vs Profile2Gen:p-value = 0.2337 (Not Significant) - Traditional vs PreProcess:p-value = 1.0000 (Not Significant) - Traditional vs Profile2Gen:p-value = 0.0197 (Significant) - PreProcess vs Profile2Gen:p-value = 0.0000 (Significant)

**Dataset: Diabetes, Model: NFlow, Proportion: 0.682**   Post hoc Results: - Real vs Traditional:p-value = 0.0000 (Significant) - Real vs PreProcess:p-value = 0.5749 (Not Significant) - Real vs Profile2Gen:p-value = 1.0000 (Not Significant) - Traditional vs PreProcess:p-value = 0.4104 (Not Significant) - Traditional vs Profile2Gen:p-value = 0.0094 (Significant) - PreProcess vs Profile2Gen:p-value = 1.0000 (Not Significant)

**Dataset: Diabetes, Model: NFlow, Proportion: 0.909**   Post hoc Results: - Real vs Traditional:p-value = 0.0000 (Significant) - Real vs PreProcess:p-value = 0.0010 (Significant) - Real vs Profile2Gen:p-value = 1.0000 (Not Significant) - Traditional vs PreProcess:p-value = 1.0000 (Not Significant) - Traditional vs Profile2Gen:p-value = 0.0000 (Significant) - PreProcess vs Profile2Gen:p-value = 0.0003 (Significant)

**Dataset: Diabetes, Model: Tvae, Proportion: 0.454**   Post hoc Results: - Real vs Traditional:p-value = 0.0000 (Significant) - Real vs PreProcess:p-value = 0.0000 (Significant) - Real vs Profile2Gen:p-value = 0.0000 (Significant) - Traditional vs PreProcess:p-value = 0.0132 (Significant) - Traditional vs Profile2Gen:p-value = 0.0000 (Significant) - PreProcess vs Profile2Gen:p-value = 0.0000 (Significant)

**Dataset: Diabetes, Model: Tvae, Proportion: 0.682**   Post hoc Results: - Real vs Traditional:p-value = 0.0002 (Significant) - Real vs PreProcess:p-value = 0.1896 (Not Significant) - Real vs Profile2Gen:p-value = 0.0000 (Significant) - Traditional vs PreProcess:p-value = 0.0000 (Significant) - Traditional vs Profile2Gen:p-value = 0.0000 (Significant) - PreProcess vs Profile2Gen:p-value = 0.0021 (Significant)

**Dataset: Diabetes, Model: Tvae, Proportion: 0.909**   Post hoc Results: - Real vs Traditional:p-value = 0.0000 (Significant) - Real vs PreProcess:p-value = 0.0000 (Significant) - Real vs Profile2Gen:p-value = 0.0000 (Significant) - Traditional vs PreProcess:p-value = 0.2398 (Not Significant) - Traditional vs Profile2Gen:p-value = 0.0000 (Significant) - PreProcess vs Profile2Gen:p-value = 0.0000 (Significant)

**Dataset: Fat, Model: Bayesian_network, Proportion: 0.227**   Post hoc Results: - Real vs Traditional:p-value = 0.0000 (Significant) - Real vs PreProcess:p-value = 0.0000 (Significant) - Real vs Profile2Gen:p-value = 0.0000 (Significant) - Traditional vs PreProcess:p-value = 0.0000 (Significant) - Traditional vs Profile2Gen:p-value = 0.0000 (Significant) - PreProcess vs Profile2Gen:p-value = 0.7107 (Not Significant)

**Dataset: Fat, Model: Bayesian_network, Proportion: 0.454**   Post hoc Results: - Real vs Traditional:p-value = 0.0000 (Significant) - Real vs PreProcess:p-value = 0.0000 (Significant) - Real vs Profile2Gen:p-value = 0.0000 (Significant) - Traditional vs PreProcess:p-value = 0.0000 (Significant) - Traditional vs Profile2Gen:p-value = 0.0000 (Significant) - PreProcess vs Profile2Gen:p-value = 1.0000 (Not Significant)

**Dataset: Fat, Model: Bayesian_network, Proportion: 0.682**   Post hoc Results: - Real vs Traditional:p-value = 0.0000 (Significant) - Real vs PreProcess:p-value = 0.0000 (Significant) - Real vs Profile2Gen:p-value = 0.0000 (Significant) - Traditional vs PreProcess:p-value = 0.0000 (Significant) - Traditional vs Profile2Gen:p-value = 0.0000 (Significant) - PreProcess vs Profile2Gen:p-value = 1.0000 (Not Significant)

**Dataset: Fat, Model: Bayesian_network, Proportion: 0.909** Post hoc Results: - Real vs Traditional:p-value = 0.0000 (Significant) - Real vs PreProcess:p-value = 0.0000 (Significant) - Real vs Profile2Gen:p-value = 0.0000 (Significant) - Traditional vs PreProcess:p-value = 0.0015 (Significant) - Traditional vs Profile2Gen:p-value = 0.0000 (Significant) - PreProcess vs Profile2Gen:p-value = 0.0018 (Significant)

**Dataset: Fat, Model: NFlow, Proportion: 0.227** Post hoc Results: - Real vs Traditional:p-value = 0.0000 (Significant) - Real vs PreProcess:p-value = 0.0000 (Significant) - Real vs Profile2Gen:p-value = 0.0243 (Significant) - Traditional vs PreProcess:p-value = 0.0000 (Significant) - Traditional vs Profile2Gen:p-value = 0.0000 (Significant) - PreProcess vs Profile2Gen:p-value = 0.0000 (Significant)

**Dataset: Fat, Model: NFlow, Proportion: 0.454** Post hoc Results: - Real vs Traditional:p-value = 0.0000 (Significant) - Real vs PreProcess:p-value = 0.0000 (Significant) - Real vs Profile2Gen:p-value = 1.0000 (Not Significant) - Traditional vs PreProcess:p-value = 0.0002 (Significant) - Traditional vs Profile2Gen:p-value = 0.0000 (Significant) - PreProcess vs Profile2Gen:p-value = 0.0000 (Significant)

**Dataset: Fat, Model: NFlow, Proportion: 0.682** Post hoc Results: - Real vs Traditional:p-value = 0.0000 (Significant) - Real vs PreProcess:p-value = 0.0000 (Significant) - Real vs Profile2Gen:p-value = 1.0000 (Not Significant) - Traditional vs PreProcess:p-value = 0.0000 (Significant) - Traditional vs Profile2Gen:p-value = 0.0000 (Significant) - PreProcess vs Profile2Gen:p-value = 0.0000 (Significant)

**Dataset: Fat, Model: NFlow, Proportion: 0.909** Post hoc Results: - Real vs Traditional:p-value = 0.0000 (Significant) - Real vs PreProcess:p-value = 0.0000 (Significant) - Real vs Profile2Gen:p-value = 0.0015 (Significant) - Traditional vs PreProcess:p-value = 0.0000 (Significant) - Traditional vs Profile2Gen:p-value = 0.0000 (Significant) - PreProcess vs Profile2Gen:p-value = 0.0000 (Significant)

**Dataset: Fat, Model: Tvae, Proportion: 0.227** Post hoc Results: - Real vs Traditional:p-value = 0.0000 (Significant) - Real vs PreProcess:p-value = 0.0000 (Significant) - Real vs Profile2Gen:p-value = 0.0000 (Significant) - Traditional vs PreProcess:p-value = 0.0000 (Significant) - Traditional vs Profile2Gen:p-value = 0.0785 (Not Significant) - PreProcess vs Profile2Gen:p-value = 0.0000 (Significant)

**Dataset: Fat, Model: Tvae, Proportion: 0.454** Post hoc Results: - Real vs Traditional:p-value = 0.0000 (Significant) - Real vs PreProcess:p-value = 0.0000 (Significant) - Real vs Profile2Gen:p-value = 0.0000 (Significant) - Traditional vs PreProcess:p-value = 0.0000 (Significant) - Traditional vs Profile2Gen:p-value = 0.1220 (Not Significant) - PreProcess vs Profile2Gen:p-value = 0.0000 (Significant)

**Dataset: Fat, Model: Tvae, Proportion: 0.682** Post hoc Results: - Real vs Traditional:p-value = 0.0000 (Significant) - Real vs PreProcess:p-value = 0.0000 (Significant) - Real vs Profile2Gen:p-value = 0.0000 (Significant) - Traditional vs PreProcess:p-value = 0.0000 (Significant) - Traditional vs Profile2Gen:p-value = 0.0011 (Significant) - PreProcess vs Profile2Gen:p-value = 0.0000 (Significant)

**Dataset: Fat, Model: Tvae, Proportion: 0.909** Post hoc Results: - Real vs Traditional:p-value = 0.0000 (Significant) - Real vs PreProcess:p-value = 0.0000 (Significant) - Real vs Profile2Gen:p-value = 0.0000 (Significant) - Traditional vs PreProcess:p-value = 0.0000 (Significant) - Traditional vs Profile2Gen:p-value = 1.0000 (Not Significant) - PreProcess vs Profile2Gen:p-value = 0.0000 (Significant)

**Dataset: Fat, Model: Ctgan, Proportion: 0.227** Post hoc Results: - Real vs Traditional:p-value = 0.0000 (Significant) - Real vs PreProcess:p-value = 0.0000 (Significant) - Real vs Profile2Gen:p-value = 0.0000 (Significant) - Traditional vs PreProcess:p-value = 1.0000 (Not Significant) - Traditional vs Profile2Gen:p-value = 0.0007 (Significant) - PreProcess vs Profile2Gen:p-value = 0.0000 (Significant)

**Dataset: Fat, Model: Ctgan, Proportion: 0.454** Post hoc Results: - Real vs Traditional:p-value = 0.0000 (Significant) - Real vs PreProcess:p-value = 0.0000 (Significant) - Real vs Profile2Gen:p-value = 0.0177 (Significant) - Traditional vs PreProcess:p-value = 0.0634 (Not Significant) - Traditional vs Profile2Gen:p-value = 0.0000 (Significant) - PreProcess vs Profile2Gen:p-value = 0.0000 (Significant)

**Dataset: Fat, Model: Ctgan, Proportion: 0.682**  Post hoc Results: - Real vs Traditional:p-value = 0.0000 (Significant) - Real vs PreProcess:p-value = 0.0000 (Significant) - Real vs Profile2Gen:p-value = 0.0000 (Significant) - Traditional vs PreProcess:p-value = 0.0049 (Significant) - Traditional vs Profile2Gen:p-value = 0.0000 (Significant) - PreProcess vs Profile2Gen:p-value = 0.0000 (Significant)

**Dataset: Fat, Model: Ctgan, Proportion: 0.909**  Post hoc Results: - Real vs Traditional:p-value = 0.0000 (Significant) - Real vs PreProcess:p-value = 0.0000 (Significant) - Real vs Profile2Gen:p-value = 0.0000 (Significant) - Traditional vs PreProcess:p-value = 0.0197 (Significant) - Traditional vs Profile2Gen:p-value = 0.0000 (Significant) - PreProcess vs Profile2Gen:p-value = 0.0000 (Significant)

**Dataset: Plasma, Model: Bayesian_network, Proportion: 0.682**  Post hoc Results: - Real vs Traditional:p-value = 1.0000 (Not Significant) - Real vs PreProcess:p-value = 0.0033 (Significant) - Real vs Profile2Gen:p-value = 0.0000 (Significant) - Traditional vs PreProcess:p-value = 0.1703 (Not Significant) - Traditional vs Profile2Gen:p-value = 0.0000 (Significant) - PreProcess vs Profile2Gen:p-value = 0.0000 (Significant)

**Dataset: Plasma, Model: Bayesian_network, Proportion: 0.909**  Post hoc Results: - Real vs Traditional:p-value = 0.1947 (Not Significant) - Real vs PreProcess:p-value = 0.0000 (Significant) - Real vs Profile2Gen:p-value = 0.0000 (Significant) - Traditional vs PreProcess:p-value = 0.0000 (Significant) - Traditional vs Profile2Gen:p-value = 0.0000 (Significant) - PreProcess vs Profile2Gen:p-value = 0.0000 (Significant)

**Dataset: Plasma, Model: NFlow, Proportion: 0.227**  Post hoc Results: - Real vs Traditional:p-value = 1.0000 (Not Significant) - Real vs PreProcess:p-value = 1.0000 (Not Significant) - Real vs Profile2Gen:p-value = 0.0000 (Significant) - Traditional vs PreProcess:p-value = 1.0000 (Not Significant) - Traditional vs Profile2Gen:p-value = 0.0000 (Significant) - PreProcess vs Profile2Gen:p-value = 0.0000 (Significant)

**Dataset: Plasma, Model: NFlow, Proportion: 0.454**  Post hoc Results: - Real vs Traditional:p-value = 0.1657 (Not Significant) - Real vs PreProcess:p-value = 1.0000 (Not Significant) - Real vs Profile2Gen:p-value = 0.0000 (Significant) - Traditional vs PreProcess:p-value = 0.1947 (Not Significant) - Traditional vs Profile2Gen:p-value = 0.0000 (Significant) - PreProcess vs Profile2Gen:p-value = 0.0000 (Significant)

**Dataset: Plasma, Model: NFlow, Proportion: 0.909**  Post hoc Results: - Real vs Traditional:p-value = 1.0000 (Not Significant) - Real vs PreProcess:p-value = 1.0000 (Not Significant) - Real vs Profile2Gen:p-value = 0.0000 (Significant) - Traditional vs PreProcess:p-value = 1.0000 (Not Significant) - Traditional vs Profile2Gen:p-value = 0.0000 (Significant) - PreProcess vs Profile2Gen:p-value = 0.0000 (Significant)

## Q. Fairness

In this section, we present complementary figures to the main text related to the discussion of fairness.

## R. Model-Specific Performance

In machine learning, there is no silver bullet for solving problems. This is especially true for generative models, whose diversity reflects different trade-offs and applications. Each model has unique characteristics that make it more suitable for specific scenarios, and our experiments highlight this variability. Our primary goal was to analyze whether, despite these differences, our approach remains effective. Variability is an inherent characteristic of generative models due to their distinct underlying mechanisms. CTGAN, for instance, relies on deep neural networks, which are highly sensitive to training conditions, hyperparameters, and data distribution. This sensitivity can lead to fluctuations in performance (Xu et al., 2019b). In contrast, Bayesian Networks explicitly model dependencies between variables, making them less prone to such fluctuations (Heckerman, 2022). Our results confirm this pattern: lower RMSE values were observed in Bayesian Networks and CTGAN, particularly in TSTR, when trained with smaller synthetic datasets. This suggests that these models can generalize well to real data, as their generated distributions align closely with real distributions. However, the observed variation in performance across datasets indicates that factors such as dataset complexity, feature dependencies, and data impurity also play a role in model stability. Both models perform better with structured and cleaner data, as their ability to capture distributions improves in less noisy scenarios. Additionally, as detailed in Appendix D, we selected a

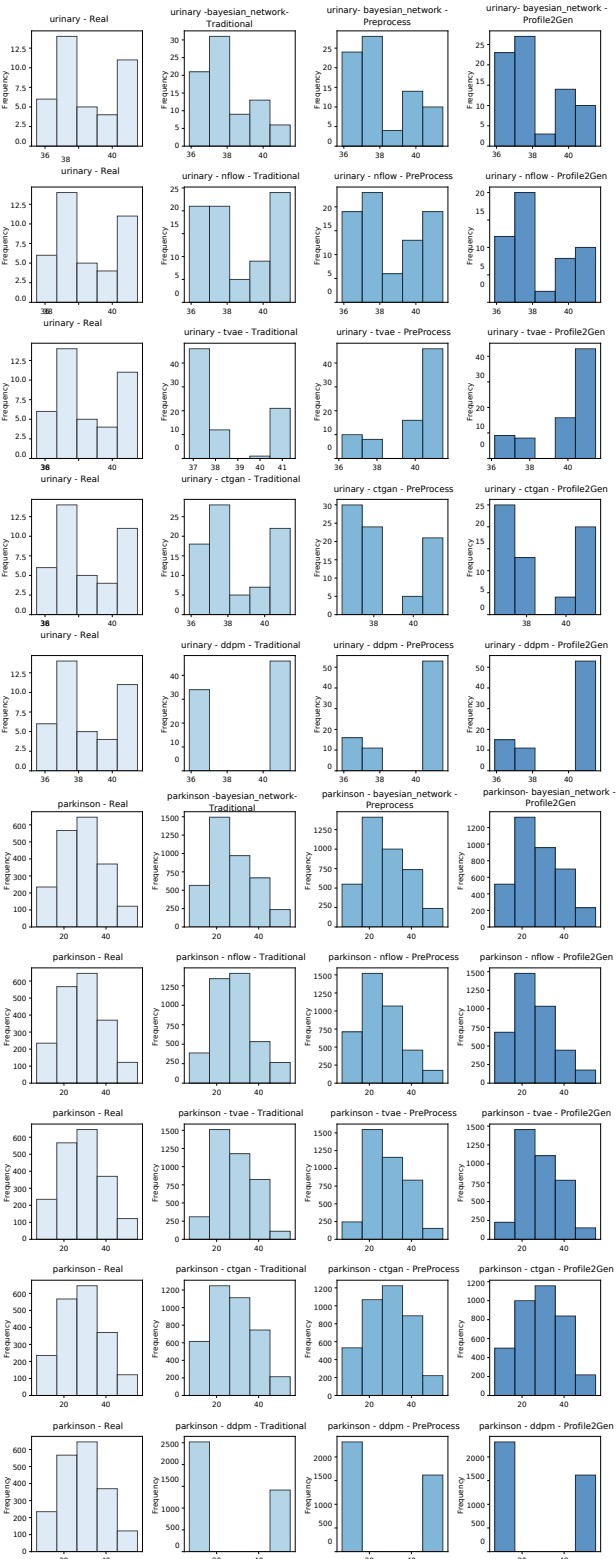

*Figure 32.* Histograms of target values for the various datasets, comparing the distribution of real data with synthetic data generated by different models ( ddpm, tabformer, and great) and techniques (Traditional, Preprocessing, Profile2Gen). This comparison highlights the impact of different models and techniques on the representation of minority classes (values in the tails of the distribution).

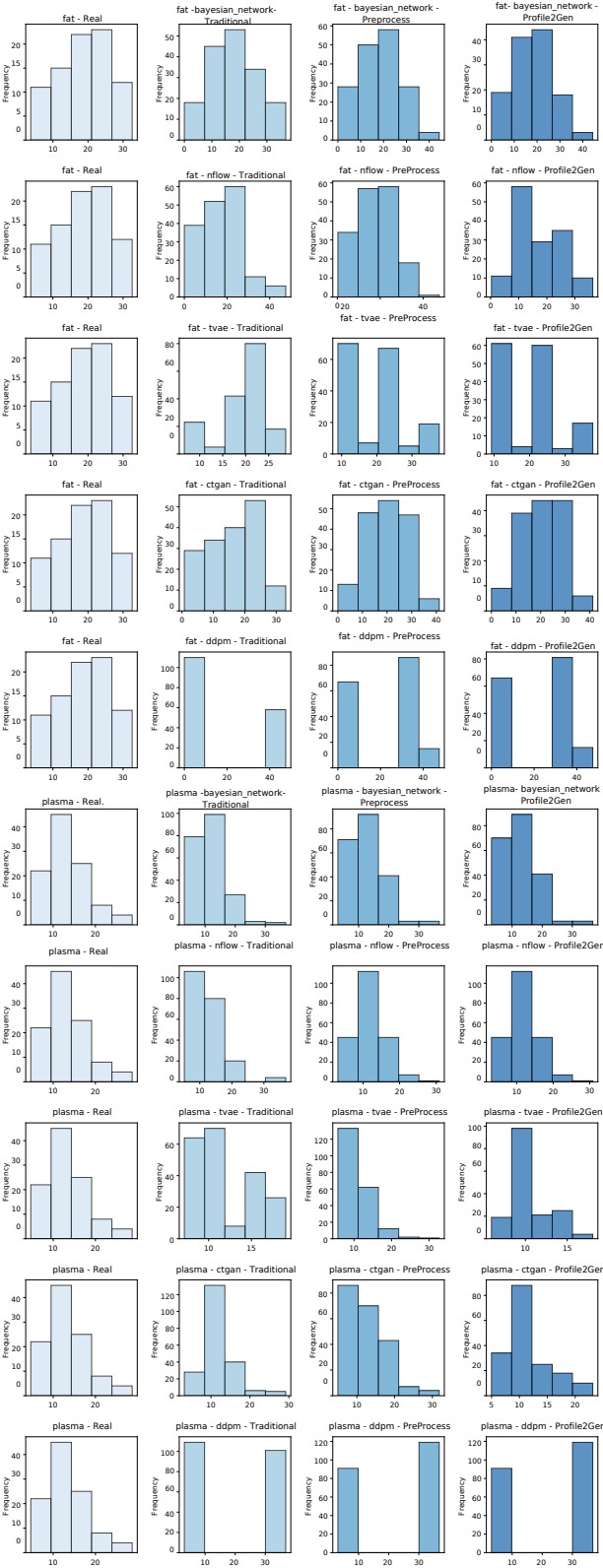

*Figure 33.* Histograms of target values for the various datasets, comparing the distribution of real data with synthetic data generated by different models (ddpm, tabformer, and great) and techniques (Traditional, Preprocessing, Profile2Gen). This comparison highlights the impact of different models and techniques on the representation of minority classes (values in the tails of the distribution).

medium-sized dataset for parameter tuning, considering dataset lengths and computational feasibility. We conducted an optimized parameter search using Optuna to minimize errors. However, this tuning was not applied to every dataset due to the large number of experiments and to avoid overfitting. Therefore, the choice of hyperparameters may have also influenced model variability. Variability in performance is an expected outcome in machine learning models, where small parameter changes can lead to significant differences in results. Our study highlights both the most variable and the most stable models. Additionally, when evaluating model performance, we observed greater consistency in tree-based models such as Random Forest and Extra Trees, which split nodes based on impurity indices (James et al., 2013). This characteristic may explain their robustness, particularly in the Profile2Gen experiments.

To corroborate our hypothesis that the nature of the model and dataset characteristics contribute to variability, we conducted an additional experiment in which fine-tuning was performed on the same dataset used for training. This reduces external influences and allows us to analyze whether hyperparameter optimization is the main factor affecting performance. We randomly selected two datasets (Fat and Cholesterol) and evaluated CTGAN in two scenarios:

1. Fine-tuning performed on the same dataset used for training.

2. Fine-tuning performed on an intermediary dataset - the same one used in the main paper.

| RMSE (Tuning on the same dataset) | Dataset | Approach | RMSE (Tuning on intermediary dataset) |
|---|---|---|---|
| 2.801107 | Fat | Profile2Gen | 3.260037 |
| 1.227212 | Cholesterol | Profie2Gen | 1.234250 |
| 3.749975 | Fat | Traditional | 3.303178 |
| 1.032838 | Cholesterol | Traditional | 1.182572 |
| 4.167451 | Fat | Preprocess | 4.2000461 |
| 1.211957 | Cholesterol | Preprocess | 1.104390 |

Table R shows that in Profile2Gen, variability was lower when the tuning was performed on the same dataset than using an intermediary dataset. However, for the other approaches, the variability increased, suggesting that although hyperparameter tuning has an impact, it is not the primary factor driving performance fluctuations. Instead, our results indicate that the model's intrinsic nature and dataset complexity contribute more significantly to variability.

## S. Dataset-variation analysis

In general, Profile2Gen outperforms traditional preprocessing methods. When analyzing the protocols separately, we observe that in the TSTR scenario, Profile2Gen achieves the best performance in approximately 80% of the cases. However, the datasets where preprocessing outperformed Profile2Gen the most were Cholesterol and Diabetes, which also happen to be the two smallest datasets. Conversely, in the Parkinson dataset—the largest one—Profile2Gen demonstrated significantly better performance than preprocessing. This suggests that Profile2Gen handles larger datasets more effectively than preprocessing in a TSTR scenario. A similar pattern was observed in the augmentation scenario.

