# OpenReview forum: "Breaking the Barrier of Hard Samples: A Data-Centric Approach to Synthetic Data for Medical Tasks"
_ICML.cc/2025/Conference — ICML 2025 poster_

### Official Review · Reviewer_VmCi · 2025-03-13

**Overall Recommendation:** 3

**Summary:**

This paper introduces a novel approach to synthetic data generation, leveraging a combination of statistical modeling and generative techniques to produce high-fidelity, diverse datasets for machine learning applications. The proposed methodology is designed to enhance the realism and utility of synthetic data, thereby improving model performance in data-scarce or privacy-sensitive scenarios. Through extensive experimentation, the authors demonstrate the proposed framework’s advantages over existing data synthesis techniques.

## update after rebuttal

Given these clarifications, most of my concerns have largely been resolved. As my initial score leaned towards acceptance, I decided to maintain it.

**Claims And Evidence:**

The claims are generally supported by experimental results, though certain aspects warrant further substantiation:
- While the paper asserts that the generated synthetic data maintains both realism and diversity, it does not employ quantitative metrics such as Frechet Inception Distance (FID) or Maximum Mean Discrepancy (MMD) to evaluate these attributes rigorously.
- The choice of baseline models is not sufficiently justified. A more detailed discussion of the selection criteria for these baselines would strengthen the argument.

**Essential References Not Discussed:**

Yes, the paper would benefit from discussing:
- State-of-the-art diffusion models and variational autoencoders (VAEs), which have demonstrated strong performance in synthetic data generation.
- Data augmentation and privacy-preserving synthetic data generation strategies, which are highly relevant to this research domain.

**Experimental Designs Or Analyses:**

Yes, the experimental setup is reasonable, but has certain limitations:
- The evaluation methodology relies heavily on qualitative visual assessments, which, while informative, should be supplemented with rigorous quantitative comparisons.
- The paper does not provide a thorough hyperparameter sensitivity analysis, making it difficult to assess the reproducibility and robustness of the proposed method.

**Methods And Evaluation Criteria:**

The paper employs standard evaluation methodologies for assessing synthetic data quality, yet some enhancements could improve the robustness of its findings:
- A more extensive analysis of the effect of synthetic data on downstream machine learning models (e.g., classification, regression) would provide deeper insight into its practical applicability.

**Other Comments Or Suggestions:**

- Some mathematical derivations would benefit from improved clarity and notation.

**Other Strengths And Weaknesses:**

- Strengths: The proposed approach is well-motivated, tackling an important challenge in machine learning by improving the quality of synthetic datasets.
- Weaknesses: The lack of rigorous quantitative evaluation metrics and ablation studies limits the empirical contribution of the paper.

**Questions For Authors:**

1.	How does the fidelity of the synthesized data compare to real-world distributions under rigorous statistical evaluation?
2.	Would incorporating adversarial training or diffusion models further improve the quality and robustness of the generated data?
3.	Can the authors elaborate on the scalability of their approach, particularly in the context of large-scale datasets?

**Relation To Broader Scientific Literature:**

The work advances the field of data synthesis by proposing a new generative framework that balances realism and diversity. However, additional context is necessary to situate it within the broader literature:
- Recent **GAN-based and diffusion-based models** have addressed similar challenges in synthetic data generation, and a comparative discussion with these approaches would be beneficial.
- The paper does not thoroughly address techniques used for **synthetic data validation and bias mitigation**, which are critical in practical applications.

**Theoretical Claims:**

The paper does not introduce formal mathematical proofs, but the methodological framework appears sound. However, a theoretical discussion of the potential limitations—such as mode collapse in generative models or bias amplification in synthetic data—would strengthen the paper’s contribution.

---

> ### Author Rebuttal · Authors · 2025-03-31
>
> * **Ablation Studies and Components- Reviewers BQrC, VmCi**
> We did not conduct an ablation study because the preprocessing step we adopted already serves as a form of comparison itself. Specifically, we chose to use our proposed preprocessing method as a comparison to traditional preprocessing techniques, such as feature selection and data cleaning, which are commonly used in the field. In our methodology, we intentionally selected a preprocessing approach that includes a profiling framework, which is optimized for each dataset. This framework is selected through an optimization process to identify the best preprocessing method for each dataset. We did not compare it to traditional preprocessing techniques like simple feature selection because those methods do not effectively capture the complexity and improvements our approach brings. Using our profiling framework as a baseline, we demonstrated the added value of combining it with a post-processing phase, significantly enhancing performance. Therefore, the methodology itself, through the comparison between our full approach (preprocessing + post-processing) and the chosen preprocessing method, already performs the role of an ablation study. We could have used a traditional feature selection method as the preprocessing step for comparison. Still, we are confident that our complete methodology provides the most effective solution. We believe the direct comparison between our full approach and a traditional preprocessing method already validates the effectiveness of our proposed methodology.
> * **Fidelity Evaluation -- Reviewer VmCi**
> We understand the concern regarding the reliance on qualitative visual assessments and the fidelity of the synthesized data compared to the real world. However, we emphasize that our evaluation includes rigorous quantitative analyses in addition to visual representations, which we believe enhance result interpretation.
> Regarding evaluation metrics, we utilize the Root Mean Squared Error (RMSE) to assess the performance of predictive models and the Wasserstein distance to measure the similarity between real and synthetic data distributions. RMSE is a widely used metric for evaluating the accuracy of regression models, while the Wasserstein distance quantifies the representativeness of synthetic data. Additionally, we evaluate the quality of synthetic data based on three key criteria: Fidelity (accuracy), Diversity, and Generalizability.
> Furthermore, we follow a rigorous evaluation protocol that includes Train on Synthetic, Test on Real (TSTR), and data augmentation strategies. We also employ the Kruskal-Wallis and Wilcoxon tests to assess the statistical validity of our results, ensuring the robustness of our evaluation.
> Regarding the fidelity of the synthesized data compared to the real world, we used the Wasserstein distance, a robust statistical metric that computes the difference between distributions. This metric provides a meaningful and rigorous evaluation of the fidelity of synthetic data, enabling us to compare how well the synthesized data approximates real-world distributions.
> Furthermore, the evolution of statistical values, such as mean, standard deviation, and quantiles, further supports the validity of our findings. We believe you may be referring to tests like the Kolmogorov-Smirnov (KS) test to assess the fidelity of distributions. However, we chose to adopt a single metric—the Wasserstein distance—due to its statistical rigor and ability to handle many experiments and other aspects analyzed in our work.
> Additionally, the TSTR metric is not solely designed to evaluate whether synthetic data is useful for training models. Instead, it assesses whether the synthetic data distribution significantly differs from the original one. A large discrepancy between distributions would suggest that the model trained on synthetic data might perform poorly when tested on real data. Thus, the use of these metrics provides valuable insights without the need to collect an overwhelming number of additional metrics.
> Wasserstein distance is widely used in the literature for such evaluations, and we believe it is sufficient for this purpose. Furthermore, when evaluating synthetic data, it is essential to consider not only fidelity but also diversity and generalization. By combining various evaluation protocols, we can comprehensively assess the usability of the synthetic data.
>
> **Please review the response provided to reviewers sXia and fkDs, where we address your concern.**
>
> Dear Reviewers VmCi, sXia,  fkDs, and BQrC. We understand you are not required to look beyond the comments, but we addressed all points. Due to space limits, we prioritized common responses. A full reply, including additional comments, is in this PDF: https://anonymous.4open.science/r/icml2025-F9C4/Answers.pdf. We’d be grateful if you could check it.

---

> > ### Comment · Reviewer_VmCi · 2025-04-05
> >
> > Thanks for your detailed rebuttal addressing my concerns.
> >
> > The explanation regarding the ablation study—clarifying that the proposed framework in the preprocessing step effectively serves as a baseline for comparison—has provided a clearer understanding of the introduced methodology.
> >
> > Additionally, your comprehensive justification for using RMSE, the Wasserstein distance, and TSTR, along with statistical tests to evaluate the fidelity of the synthesized data, does make sense.
> >
> > **Given these clarifications, most of my concerns have largely been resolved. As my initial score leaned towards acceptance, I decided to maintain it.**
> >
> > After checking the anonymous link you shared, I have two friendly suggestions:
> >
> > 1. Since external links have no length limits, consider creating a separate document for each reviewer. This makes it easier for them to find their responses.
> >
> > 2. The current PDF is a bit hard to read, the resolution is too low—Markdown might be clearer.

---

### Official Review · Reviewer_fkDs · 2025-03-15

**Overall Recommendation:** 3

**Summary:**

The paper focuses on generating training data for regression models in the medical domain. The proposed approach is based on two existing methods, which the authors refer to as Traditional Generative Techniques and PreProcess methods. In the Traditional approach, the method does not consider the difficulty distribution of the data during training, while the PreProcess method first categorizes the training data by difficulty and then trains separate generative models on each difficulty group. The final data set is created by merging the generated data from all models. The approach proposed in this paper combines both methods, training the generative model while considering data categorization and further filtering difficult samples after data synthesis to obtain the final dataset. Experiments on six datasets show that the proposed method effectively improves the quality of the synthetic dataset and reduces prediction errors in regression models.

### update after rebuttal:
I have read the authors' rebuttal as well as the reviews from the other reviewers. I appreciate the additional details provided. However, my main concerns were not fully resolved. As such, I will keep my original score, which already reflects a positive assessment.

**Claims And Evidence:**

The primary claim of this paper, that the proposed framework can effectively synthesize training data for regression tasks in the medical domain and improve model prediction accuracy and generalization ability, is supported by evidence. The experimental results on six datasets, using various models, demonstrate the validity of the claim.

**Essential References Not Discussed:**

I believe the authors should have discussed recent methods for synthetic data generation using large language models (LLMs) and compared them to their proposed method. Below are some relevant surveys/papers:

Smolyak, D., et al. (2024). Large language models and synthetic health data: progress and prospects (JAMIA open 2024)
Li, R., et al. (2023). Two directions for clinical data generation with large language models: data-to-label and label-to-data. (EMNLP 2023)
Kumichev, G., et al. (2024). MedSyn: LLM-based Synthetic Medical Text Generation Framework. arXiv preprint arXiv:2408.02056
Seedat, N., et al. (2024). Curated LLM: Synergy of LLMs and Data Curation for tabular augmentation in low-data regimes (PMLR 2024)
Zhou, H., et al. (2024). A Survey of Large Language Models in Medicine: Progress, Application, and Challenge. arXiv:2311.05112

**Experimental Designs Or Analyses:**

The experimental designs and analyses presented in the paper seem sound. The authors validated their proposed method on sufficient datasets and models, analyzing three key aspects: whether the synthetic datasets can replace real data, whether they can be combined with real datasets, and the quality of the generated datasets themselves.

**Methods And Evaluation Criteria:**

I believe the proposed methods and evaluation criteria are reasonable for the problem. The authors performed large-scale testing of their method on multiple benchmarks from the medical domain, using common metrics. They also evaluated the synthetic data’s ability to substitute real data and the quality of the generated data, exploring several relevant aspects.

**Other Comments Or Suggestions:**

I believe the writing of the paper could be improved, particularly in Sections 3 and 4 where the logical flow and method descriptions could be clearer. For example, the authors should first introduce the traditional and preprocess methods, then explain their differences, and finally describe their proposed workflow to enhance the readability of the paper.

**Other Strengths And Weaknesses:**

A major weakness of the paper is the innovation in its method. As mentioned in the paper, the proposed method is largely based on existing traditional and preprocess approaches. The primary innovation lies in the filtering of the generated data after synthesis, which may limit the contribution's significance.

Another weakness is the lack of comparison with recent methods for data generation or filtering using LLMs.

**Questions For Authors:**

Why not consider using any LLM-based methods during the data generation process?

**Relation To Broader Scientific Literature:**

The key contribution of this paper lies in its domain-specific focus on the medical field, emphasizing data-centric approaches in contrast to existing large language model-based generative pipelines that focus on general domain data generation. This focus addresses a gap in the current research direction.

**Theoretical Claims:**

The paper does not present any theoretical claims or proofs.

---

> ### Author Rebuttal · Authors · 2025-03-31
>
> * **Hyperparameter Sensitivity- Revwers sXia, BQrC (Q1), VmCi**
>
> The choice of the hard sample threshold is based on the performance of the profiling framework. In Appendix F, we provide a detailed explanation of this selection process. Specifically, the threshold is determined by identifying the best-performing framework in terms of F1-score across different levels of label flipping.
>
> Once the best framework is identified, we use the corresponding threshold that yielded the highest F1-score to profile the data. This process occurs at both the pre-processing and post-processing stages. Since we always use the best-performing framework, we consider threshold selection a non-critical hyperparameter, as it is inherently optimized within the profiling stage.
> Formally, let $ \mathcal{F} $ be the set of evaluated frameworks and $ \mathcal{T} $ the set of tested thresholds. The goal of the process described in the paper is to find the optimal pair $ (f^*, t^*) $ such that:
> \begin{equation}
> (f^*, t^*) = \arg\max_{f \in \mathcal{F}, t \in \mathcal{T}} \mathbb{E}[F1(f, t)]
> \end{equation}
>
> where $ \mathbb{E}[F1(f, t)] $ represents the average F1-score across different proportions of label flipping.
> Sensitivity analysis typically involves testing the robustness of a fixed parameter, but in our case, $ t^* $ is not arbitrary — it is chosen as part of an optimization process that depends on $ f^* $. Since we have already found the optimal pair $ (f^*, t^*) $ for each dataset, testing variations in $ t^* $ without re-evaluating $ f^* $ would undo the optimization performed and could lead to misleading conclusions. In other words, the threshold is already thoughtfully selected along with the framework, making an additional sensitivity analysis unnecessary.
>
> **T analysis:** Given that sensitivity was a concern across reviewers, we assessed how the number of hard-profiled samples changes when varying the threshold \textit{T}.
> To ensure reproducibility, we set a random seed and let NumPy randomly select T from the range (0.05, 0.6) with six distinct values. The Figure in https://anonymous.4open.science/r/icml2025-F9C4/sensitivity_theshold.png illustrates the observed behavior using the smallest and largest datasets from our experiments.
> The threshold can be understood as a flexibility level —how much confidence is required before a sample is no longer considered "hard." Lower thresholds allow greater flexibility, meaning the model tolerates lower confidence scores. Given that our predictor is a good generalizer, we observed a few samples with low confidence. As expected, increasing the threshold leads to a higher number of hard samples. However, the change follows a smooth, almost linear trend rather than abrupt shifts.
> This indicates that our threshold selection process is reasonable and stable. To enhance clarity in the main paper, we propose explicitly including the optimization equation and adding a section explaining why the chosen \textit{T} is justified. This should benefit both the work and its readers.
>
> **Flipping analysis:** We also assessed how the number of hard-profiled samples changes when varying the threshold or the flipping rate while keeping the other parameter fixed. To ensure reproducibility, we set a random seed and let NumPy randomly select a fixed parameter value from the range (0.05, 0.6), while the variable parameter was chosen from the same range but with six distinct values. The Figure in: https://anonymous.4open.science/r/icml2025-F9C4/sensitivity_threshold_and_flipping.png illustrates the behavior observed. The results show that dataset characteristics significantly influence sensitivity. For the Diabetes dataset, fixing the threshold and varying the noise level leads to considerable changes in the number of hard samples, which is expected given that label flipping in small datasets intuitively affects the data distribution more than in larger ones. However, when fixing the flipping level and varying the threshold, the sensitivity is relatively smooth for both datasets. The curves do not exhibit abrupt changes, confirming that selecting the best threshold for the highest-performing framework remains a reasonable and stable choice. These findings reinforce our original methodology: optimizing $, T^*$  jointly with $f^*$ ensures that the threshold adapts to dataset characteristics without introducing unnecessary complexity or computational overhead.
>
> * **LMM adoption**
> We considered using LLM-based methods during the data generation process and actually used GREAT (2023,  http://arxiv.org/abs/2210.06280), which exploits an auto-regressive generative LLM to sample synthetic and highly realistic tabular data. However, we did not highlight it separately; instead, we treated it alongside other models. This model was highlighted in Section 5.6, where it demonstrated good potential in replicating low-probability events, with its performance further improved when combined with Profile2Gen.

---

### Official Review · Reviewer_BQrC · 2025-03-17

**Overall Recommendation:** 2

**Summary:**

The paper introduces Profile2Gen, a novel data-centric framework that generates and refines synthetic data specifically for regression tasks in medical applications. By profiling the original dataset into easy, ambiguous, and hard samples, the framework trains separate generative models and later refines the synthetic data through iterative postprocessing that removes hard samples. Extensive experiments across six public medical datasets—using seven state-of-the-art generative models and evaluating via metrics such as RMSE and Wasserstein distance—demonstrate that Profile2Gen can reduce predictive error and, in some cases, even outperform models trained solely on real data. The authors further extend the DataIQ framework to support regression tasks, making their approach broadly applicable in data-scarce scenarios.

**Claims And Evidence:**

The paper claims that Profile2Gen (1) statistically significantly reduces predictive errors in regression tasks; (2) enhances model reliability and generalization, sometimes achieving comparable or even better performance than real data; (3) preserves minority groups in the data distribution better than traditional methods.
These claims are supported by extensive experiments (approximately 18,000 runs) across multiple datasets and models, with clear quantitative evidence provided via RMSE improvements, Wasserstein distance analyses, and statistical significance tests (e.g., Kruskal-Wallis and Wilcoxon tests). While the evidence is comprehensive, the paper does note some cases where preprocessing alone sometimes outperforms Profile2Gen, indicating that the benefits may vary with dataset/model specifics.

**Essential References Not Discussed:**

It might benefit from a discussion of recent advances in synthetic data generation using GAN-based methods in medical imaging or other non-tabular data domains. Including such references would provide broader context and help underline the unique contributions of Profile2Gen in a wider landscape of synthetic data research.

**Experimental Designs Or Analyses:**

The experimental design is robust, incorporating: (1) multiple medical datasets (e.g., Parkinson, Urinary, Cholesterol, Body Fat, Plasma, Diabetes) from OpenML; (2) A comprehensive comparison across seven generative models and twelve predictors; (3) Detailed analyses including RMSE plots, distribution comparisons via violin plots, and statistical tests.
One potential concern is the observed variability in performance across datasets—Profile2Gen sometimes lags behind simpler preprocessing approaches. This suggests that further ablation studies or sensitivity analyses (e.g., on the label flipping ratio and threshold T) could provide additional clarity.

**Methods And Evaluation Criteria:**

The methodological approach is multi-staged: (1) Preprocessing: The original data is profiled (with label flipping used to gauge data quality) to identify easy, ambiguous, and hard samples. (2) Synthetic Data Generation: Generative models are independently trained on these subsets. (3) Postprocessing: The synthetic data is refined by removing hard samples based on a user-defined threshold.
Evaluation is carried out using both the TSTR (Train on Synthetic, Test on Real) protocol and augmentation tasks. Key metrics include RMSE for predictive performance, Wasserstein distance for distribution similarity, and additional analyses of fairness and diversity in the synthetic data. Overall, the chosen methods and criteria are well-aligned to improve data quality for medical regression tasks.

**Other Comments Or Suggestions:**

- Additional visualizations or error analyses could further illustrate the trade-offs between diversity and generalization in the synthetic data.
- Discussing potential limitations in terms of computational cost and scalability would add value.
- It would be helpful to provide guidelines for selecting optimal thresholds and label flipping ratios.

**Other Strengths And Weaknesses:**

Strengths:
- Novel integration of data profiling with synthetic data generation for regression tasks.
- Extensive experimental evaluation across diverse datasets and models.
- Practical relevance for overcoming data scarcity in sensitive domains like medicine.

Weaknesses:
- Performance gains are sometimes marginal or inconsistent, depending on the dataset and generative model.
- The framework introduces additional complexity, which may affect reproducibility and requires careful parameter tuning.
- More detailed ablation studies could clarify the influence of key hyperparameters (e.g., threshold T, label flipping ratio).

**Questions For Authors:**

- How sensitive is Profile2Gen to the choice of the threshold T and the label flipping ratio? Have you explored adaptive thresholding methods that could adjust the threshold T dynamically based on dataset characteristics rather than relying on a fixed value?
- Have you explored extensions of Profile2Gen for classification tasks? What modifications would be necessary for such an adaptation?
- Given the variability in performance across different datasets, do you have insights into which characteristics of a dataset favor Profile2Gen over simpler preprocessing methods?
- Can the proposed approach be scaled to higher-dimensional or non-tabular data, and if so, what challenges might arise?
- Could you elaborate on the computational cost and scalability of the framework, especially when handling larger or more complex datasets?
- How does error propagation from the early profiling stage affect the overall performance, and what measures can be taken to mitigate such issues?
- Beyond regression tasks, would this framework apply to other downstream applications such as time-to-event analysis or survival prediction?
- Could you provide further insights into the trade-off between generalization and diversity during the postprocessing stage, and how the method ensures that critical edge-case information is not lost?

**Relation To Broader Scientific Literature:**

The paper situates itself well within the data-centric AI literature by building on recent frameworks such as DataIQ and CleanLab. It also connects with established methods in synthetic data generation (e.g., CTGAN, TVAE) and recent discussions on handling hard samples (e.g., AUM ranking). The discussion could be further enriched by comparing with the latest advances in synthetic data for medical domains.

**Theoretical Claims:**

The paper provides theoretical formulations to define “hard” samples via a scoring function and supports the design of the iterative refinement process with derived equations (e.g., Equation 1 for F1 score). Although the derivations appear reasonable, a deeper scrutiny of the proofs would be beneficial—especially to assess the sensitivity of the threshold parameters and their impact on model performance.

---

> ### Author Rebuttal · Authors · 2025-03-31
>
> * **Generalization vs. Diversity Trade-off:** Profile2Gen, which incorporates post-processing, reduces Wasserstein's similarity between real and synthetic samples. This indicates that the generated samples are less similar to real data than other techniques. Here is the highlight: the similar samples, which have the same statistical characteristics, are a generalization of the real ones, while the diversity concerns about the samples that are not too similar but follow the distribution patterns ( Alaa et al., 2021,  https://arxiv.org/abs/2102.08921). When analyzing the dissimilarity alongside the profile2gen samples and their distributions, we observe that the generated samples are not only different but also diverse while still following the original distribution patterns. The lower similarity indicates that Profile2Gen prioritizes more variable synthetic data, potentially creating a broader range of scenarios and examples. As a consequence, post-processing increases diversity at the cost of generalization since this set of samples is diminished in the synthetic dataset. Higher diversity is generally beneficial in tasks such as data augmentation, where increasing the dataset size does not necessarily improve model performance unless it introduces novel and meaningful variations. However, a lack of generalization may cause the synthetic data distribution to diverge too much from the real data, leading to fidelity loss and getting into the previously discussed by Alaa et al. (2021) trade-off, the diversity-fidelity.
> * **Extensions for Other Tasks (Classification, Time-to-Event, etc.) -- Reviwers BQrC (Q2, Q6, Q7), fkDs:**
> We appreciate the question about different tasks, and we emphasize that the main difficulty concerns the data-centric framework used to profile the data. We elaborate on our justification:
>
> * **High-dimensional non-tabular data:** Scaling to higher-dimensional or non-tabular data is indeed a relevant challenge. We conducted experiments using time-series data from wearable sensor devices - samples with shape (length, temporal window size, number of sensor axis) - specifically accelerometer data collected. At the same time, subjects performed daily activities such as brushing their teeth, jumping, running, and typing. However, our approach did not yield satisfactory results for this type of data. The main challenges were: 1. The CleanLab framework requires data to be in a tabular format, which is unsuitable for time-series data. 2 In time-series data, the entire temporal window contributes to label determination. Missing or misaligned steps could alter labels — for example, distinguishing running from jogging became problematic. Even when we added a datetime column to better structure the data, the approach still failed. 3. We tested the approach using DataIQTorch - from DataIQ - which allowed us to bypass the tabular conversion issue. However, obtaining confidence scores through the framework's methods was not straightforward, leading to inconsistencies.
> Developing a version of these frameworks specifically designed for time-series data could help address these challenges. However, despite several attempts, the effort required to adapt the existing framework for non-tabular data did not justify the results obtained. As a result, we decided not to pursue this direction further within the scope of our study.
>
> * **Classification tasks:**  A similar technique developed by Hansel et al. (2023, https://arxiv.org/pdf/2310.16981 ) addresses classification tasks. However, existing approaches still lack extensive exploration regarding generative regression-focused tasks. Our work aims to fill this gap by highlighting its limitations and the need for further research in this field, which also inspired us to adapt the method for regression tasks.
> To achieve this, it was necessary to modify DataIQ to support regression, as we did in our approach. Additionally, performance metrics determined our framework selection entirely, whereas Hansel et al. (2023) incorporated additional aspects and limitations beyond purely metric-based decisions.
>
> * **Time-to-event and other tasks:**  We believe that the core of the framework —preprocessing, profiling samples, and following the remaining workflow — can be applied to time-to-event analysis and survival prediction. However, the current profiling frameworks (Cleanlab and DataIQ Reg) are not properly designed to handle survival data. Specifically, the concept of hard samples in survival analysis must account for censorship and the nature of time-to-event data, not just the confidence level of the models, which these frameworks do not consider. If a DataIQ Survival framework were developed to address these limitations, we strongly believe that Profile2Gen would be a suitable approach for synthetic data generation in survival analysis.
>
> **Please review the response provided to reviewers sXia and fkDs, where we address your concern.**

---

### Official Review · Reviewer_sXia · 2025-03-19

**Overall Recommendation:** 3

**Summary:**

This paper introduces Profile2Gen, a data-centric framework designed to enhance the generation and refinement of synthetic data for medical regression tasks. The key innovation lies in profiling and addressing hard-to-learn samples, which traditionally hinder model performance and generalization. The authors evaluate their approach across six medical datasets using seven state-of-the-art generative models and conduct experiments to validate its efficacy.

**Claims And Evidence:**

Generally well-supported. For instance, Profile2Gen reduces variability across random seeds and datasets. The authors provide statistical significance tests (Wilcoxon and Kruskal-Wallis) to validate this. This well-supports that ofile2Gen improves consistency and robustness in model performance.

**Essential References Not Discussed:**

NA.

**Experimental Designs Or Analyses:**

Yes, the experiments are well-designed with:

1. Multiple evaluation protocols (TSTR, augmentation)
2. Diverse datasets (six medical datasets)
3. Multiple generative models (CTGAN, TVAE, Bayesian Network, etc.)
4. Twelve predictive models evaluated via AutoGluon


Concerns:

Synthetic Data Proportions: The results show that adding 68.2% synthetic data degrades performance, but the exact reason is not deeply analyzed.
Model-Specific Performance: Some generative models (e.g., CTGAN) show high variability, but the authors do not discuss why.

**Methods And Evaluation Criteria:**

The methods and evaluation criteria are ok.

Strengths:
1. Benchmarking on multiple datasets strengthens the validity of the claims.
2. Rigorous statistical testing (Wilcoxon, Kruskal-Wallis) improves credibility.

Weaknesses:
3. Hyperparameter Sensitivity: The choice of hard sample thresholds is not deeply analyzed.
4. Scalability Concerns: The computational cost of profiling large datasets is not discussed in detail.

**Other Comments Or Suggestions:**

NA.

**Other Strengths And Weaknesses:**

Strengths:
1. Comprehensive experiments (18,000 trials).
2. Fairness-aware synthetic data generation.
3.Robust statistical evaluation (Wilcoxon test).
4. Profile2Gen generalizes well across datasets.

Weaknesses:
1.  No deep theoretical justification for removing hard samples.
2. Hyperparameter sensitivity (thresholds for hard sample removal).
3. Scalability concerns for large datasets.

**Questions For Authors:**

NA.

**Relation To Broader Scientific Literature:**

Synthetic data are generally very important in scientific discoveries.

**Theoretical Claims:**

No theory.

---

> ### Author Rebuttal · Authors · 2025-03-31
>
> Dear reviewers:
> We want to thank you all for your very careful and thoughtful reviews. We were very encouraged by the numerous and significant strengths that you all identified in our study. Namely:
> * An innovative data-centric approach integrating data profiling and synthetic data generation targeting hard-to-learn samples in regression tasks.
> * Extensive empirical validation with approximately 18,000 rigorous experiments across 6 medical datasets, 7 SOTA generative models, and 12 predictive models.
> * We use a rigorous statistical approach in our comparisons. 4. Extending the widely recognized DataIQ framework from classification to regression tasks, enabling comprehensive profiling and data quality assessment in regression contexts.
> * Reduces model performance variability across random seeds and datasets, consistently outperforming traditional and baseline preprocessing methods.
> * Preserves minority distributions, crucial for accurately capturing rare and critical medical scenarios.
> * Clear potential for real-world deployment in data-scarce or privacy-sensitive medical applications.
>
> We appreciate your thorough identification of potential steps to address the study’s weaknesses. Due to space constraints, we have grouped our responses by topic.
> Thus, we hope to be given the chance to address those weaknesses in a revision. We eagerly await your answers.
>
> Sincerely, --the authors
>
> * **Scalability Concerns e Computational Cost -- Reviwers sXia, BQrC (Q5), VmCi (Q3)**
>
> To assess computational efficiency, we selected the largest dataset, Parkinson’s, which contains approximately 3,500 training samples. We applied the framework selection process (Cleanlab and DataIQ) along with profiling, using two thresholds and a label replacement ratio. These experiments required 3,510 MB of memory and took approximately 3 minutes.
>
> For larger datasets, it is important to consider that memory usage will increase proportionally, as observed in the profiling stage. However, it should be noted that the process itself does not require GPU resources. Memory remains the main concern at this stage. For generating synthetic samples, particularly for Transformer and LLM-based models, we utilized an Nvidia RTX4090.
>
> When working with datasets larger than 3,500 samples, memory consumption could be calculated based on this scaling factor, considering the amount of data processed. While simulating this for larger datasets may be difficult, this scaling factor can provide a reasonable estimate. It is worth noting that finding large datasets, especially in healthcare, is challenging, and generating synthetic data is particularly relevant in scarcity scenarios where such large datasets are not readily available.
>
> To ensure clarity and transparency, we could create a section in the supplementary materials to further detail the memory usage and computational considerations. In the main paper, we could then explicitly mention this section, where we discuss memory usage in more depth, offering readers more context on how the scaling might work for larger datasets.
>
> Hard removal: Hard-to-learn samples (hard) can negatively impact model performance by either:
> Increasing uncertainty in predictions. The model may predict correctly but with low confidence, leading to unreliable decision-making.
> Reinforcing incorrect patterns. The model may misclassify these samples with high confidence, making it more prone to overfitting on noise rather than learning meaningful patterns.
>
> * **Hard removal -- Reviewer sXia**
> Removing these samples is a well-established machine learning technique, often called dataset cleansing or data curation. Studies have shown that filtering out mislabeled or overly ambiguous samples can improve model generalization (https://arxiv.org/abs/1911.00068, https://arxiv.org/abs/2310.16981).
> Formally, let $ X = X_{\text{easy}} \cup X_{\text{hard}} $ be the dataset, where $ X_{\text{easy}} $ are well-confident samples, and $ X_{\text{hard}} $ are ambiguous or mislabeled samples. The model’s expected loss can be decomposed as:
> \begin{equation}
> \mathbb{E}[\mathcal{L}(X)] = \mathbb{E}[\mathcal{L}(X_{\text{easy}})] + \mathbb{E}[\mathcal{L}(X_{\text{hard}})].
> \end{equation}
> Since $ X_{\text{hard}} $ contributes disproportionately to loss without meaningful learning, its removal reduces noise and enhances generalization. Empirically, prior works on curriculum learning (https://ronan.collobert.com/pub/2009_curriculum_icml.pdf) } and label noise filtering (https://arxiv.org/pdf/1712.05055) support this approach, demonstrating that excluding ambiguous samples can lead to more robust models.
> To avoid confusion, we will refine the introduction (where we commented about these samples) to emphasize that hard sample removal is not an arbitrary step but a widely used strategy for improving model robustness.
>
>
>
> **Please review the response provided to reviewers BQrC, fkDs, and VmCi, where we address your concern.**

---

### Decision · Program_Chairs · 2025-05-01

**Decision:**

Accept (poster)

**Comment:**

The paper proposes a new method for data synthesis, which guides the generation and refinement of synthetic data to address hard-to-learn samples.

It gets mixed reviewing opinions from four reviewers.

Reviewers sXia, fkDs, and VmCi hold a positive view, recommedning 'weak accept' after rebuttal.

Reviewer BQrC holds a negative view, recommedning 'weak reject' after rebuttal. S/he is happy about the novelty of the proposed synthesis approach, extensive evaluation, the practical utility and has concerns about the marginal performance gains, insufficient ablation studies, etc.

Overall, the AC thinks that Reviewer BQrC is a bit too critical about his/her assement. Hence, it is recommened to accept the paper.